# FAST AND STABLE RIEMANNIAN METRICS ON SPD MANIFOLDS VIA CHOLESKY PRODUCT GEOMETRY

**Ziheng Chen**[1], **Yue Song**[2,3], **Xiao-Jun Wu**[4] **& Nicu Sebe**[1]
[1] University of Trento, Trento, Italy
[2] Shanghai Qi Zhi Institute, Shanghai, China
[3] College of AI, Tsinghua University, Beijing, China
[4] Jiangnan University, Wuxi, China

## ABSTRACT

Recent advances in Symmetric Positive Definite (SPD) matrix learning show that Riemannian metrics are fundamental to effective SPD neural networks. Motivated by this, we revisit the geometry of the Cholesky factors and uncover a simple product structure that enables convenient metric design. Building on this insight, we propose two fast and stable SPD metrics, Power–Cholesky Metric (PCM) and Bures–Wasserstein–Cholesky Metric (BWCM), derived via Cholesky decomposition. Compared with existing SPD metrics, the proposed metrics provide closed-form operators, computational efficiency, and improved numerical stability. We further apply our metrics to construct Riemannian Multinomial Logistic Regression (MLR) classifiers and residual blocks for SPD neural networks. Experiments on SPD deep learning, numerical stability analyses, and tensor interpolation demonstrate the effectiveness, efficiency, and robustness of our metrics. The code is available at `https://github.com/GitZH-Chen/PCM_BWCM`.

## 1 INTRODUCTION

Symmetric Positive Definite (SPD) matrices play a central role in various machine learning applications, including medical imaging (Chakraborty et al., 2020), electroencephalography (EEG) analysis (Li et al., 2025), brain functional connectivity (Dan et al., 2024), signal processing (Brooks et al., 2019), and computer vision (Huang and Van Gool, 2017; Wang et al., 2023; Chen et al., 2025a; Kang et al., 2025). As SPD matrices form a non-Euclidean manifold (Arsigny et al., 2005), traditional Euclidean methods are inadequate. To close this gap, several Riemannian metrics have been developed, including Affine-Invariant Metric (AIM) (Pennec et al., 2006), Log-Euclidean Metric (LEM) (Arsigny et al., 2005), Power-Euclidean Metric (PEM) (Dryden et al., 2010), Log-Cholesky Metric (LCM) (Lin, 2019), Bures–Wasserstein Metric (BWM) (Bhatia et al., 2019), and Generalized Bures–Wasserstein Metric (GBWM) (Han et al., 2023). These metrics not only provide mathematically principled tools, but also enable the extension of core deep learning components such as Multinomial Logistics Regression (MLR) (Nguyen and Yang, 2023; Chen et al., 2024a;c), Fully Connected (FC) and convolutional layers (Chakraborty et al., 2020), normalization (Brooks et al., 2019; Chakraborty, 2020; Kobler et al., 2022; Chen et al., 2024b; 2025b; Wang et al., 2025b;c), attention (Pan et al., 2022; Wang et al., 2025a), residual blocks (Katsman et al., 2024), and pooling (Wang et al., 2020; Song et al., 2022; Chen et al., 2025a; Gu et al., 2025), as well as emerging modules including graph & recurrent neural networks (Chakraborty et al., 2018; Cruceru et al., 2021) and generative models (Li et al., 2024; Chen and Lipman, 2024). Therefore, designing Riemannian metrics that are *theoretically convenient*, *computationally efficient*, and *numerically stable* is crucial to advance SPD matrix learning.

Due to the fast and stable computation of the Cholesky decomposition, LCM enjoys several advantages over the other metrics (Lin, 2019), including simple closed-form Riemannian operators, efficiency, and numerical stability. LCM is induced by the Cholesky decomposition from a Riemannian metric on the Cholesky manifold (Lin, 2019), which is the space of lower triangular matrices with positive diagonal entries. We refer to this Cholesky metric as the diagonal log metric. We reveal a simple *product structure* underlying the diagonal log metric: a Euclidean metric on the strictly lower triangular part together with $n$ copies of a Riemannian metric on $\mathbb{R}_{++}$ for the diagonal part. This

Figure 1: Overview of our theoretical framework. Here, $L$ is a Cholesky matrix.

observation opens up a principled design space, as any metric on $\mathbb{R}_{++}$ can induce a metric on the Cholesky manifold, and further yield a corresponding metric on the SPD manifold via the Cholesky decomposition.

Building on this product structure, we introduce two Cholesky metrics, the diagonal power metric and the diagonal Bures–Wasserstein metric, which induce two SPD metrics via the Cholesky decomposition: Power-Cholesky Metric (PCM) and Bures–Wasserstein–Cholesky Metric (BWCM). Unlike the diagonal logarithm in LCM, our metrics rely on diagonal powers, improving numerical stability by avoiding exponentiation and logarithms. We further define in the Cholesky factors of SPD matrices a diagonal power deformation, which continuously connects existing and new metrics. As power $\theta \to 0$, the deformed metric converges to LCM, while at $\theta = 1$ it recovers our proposed metrics, thereby offering a tunable trade-off. All proposed SPD metrics admit closed-form Riemannian operators, including geodesics, logarithmic and exponential maps, parallel transport, as well as gyrovector operators (Ungar, 2022), which extend vector addition and scalar multiplication into manifolds. These operators make our metrics directly applicable to SPD neural networks. In particular, by substituting these operators into prior formulations of Riemannian MLR classifiers (Chen et al., 2024c) and residual blocks (Katsman et al., 2024), we directly obtain SPD MLR classifiers and residual blocks under our metrics. Fig. 1 summarizes the overall framework. We validate our metric with experiments on SPD neural networks, numerical stability analyses, and tensor interpolation, showing the effectiveness, efficiency, and robustness of our metrics. In summary, our main **contributions** are:

1. Revealing the underlying simple product structure in the Cholesky manifold;
2. Proposing two Cholesky metrics and their SPD counterparts, PCM and BWCM, which admit fast and stable closed-form operators;
3. Developing SPD classifiers and residual blocks based on our metrics for SPD neural networks.

## 2 PRELIMINARIES

This section reviews the necessary background. For readability, Sec. B provides a summary of notation, while Sec. C recalls gyrovector spaces and properties of Riemannian isometries.

### 2.1 PULLBACK METRICS

**Definition 2.1** (Pullback metrics). Let $\mathcal{M}, \mathcal{N}$ be smooth manifolds, $g$ a Riemannian metric on $\mathcal{N}$, and $f : \mathcal{M} \to \mathcal{N}$ a diffeomorphism. The pullback of $g$ by $f$ is defined pointwise as $(f^*g)_P(V_1, V_2) = g_{f(P)}(f_{*,P}(V_1), f_{*,P}(V_2))$, where $P \in \mathcal{M}$, $V_1, V_2 \in T_P\mathcal{M}$, and $f_{*,P}(\cdot)$ denotes the differential of $f$ at $P$. The metric $f^*g$ is called the *pullback metric* by $f$, and $f$ is a Riemannian isometry.

This paper only considers pullbacks by diffeomorphisms. Riemannian isometries generalize bijections in set theory to manifolds while preserving their Riemannian structure. On the SPD manifold, pullback metrics are also known as *deformed metrics* (Thanwerdas and Pennec, 2022, Sec. 3).

### 2.2 SPD AND CHOLESKY MATRIX GEOMETRIES

**SPD geometries.** We denote $n \times n$ SPD matrices as $\mathcal{S}_{++}^n$ and $n \times n$ real symmetric matrices as $\mathcal{S}^n$. $\mathcal{S}_{++}^n$ is an open submanifold of the Euclidean space $\mathcal{S}^n$, known as the SPD manifold (Arsigny et al., 2005). As mentioned in Sec. 1, there are six popular Riemannian metrics on the SPD manifold: AIM (Pennec et al., 2006), LEM (Arsigny et al., 2005), PEM (Dryden et al., 2010), LCM (Lin, 2019), BWM (Bhatia et al., 2019), and GBWM (Han et al., 2023). Note that PEM is the pullback metric of

the Euclidean Metric (EM) by matrix power $\mathrm{Pow}_\theta(S) = S^\theta$ and scaled by $1/\theta^2$ $(\theta \neq 0)$ (Thanwerdas and Pennec, 2022), while GBWM is the pullback metric of BWM by $\pi(S) = M^{-\frac{1}{2}} S M^{-\frac{1}{2}}$ with $S, M \in \mathcal{S}_{++}^n$ (Han et al., 2023). For clarity, we denote PEM and GBWM as $\theta$-EM and $M$-BWM, respectively. We denote the metric tensors of AIM, LEM, $\theta$-EM, LCM, BWM, and $M$-BWM as $g^{\mathrm{AI}}$, $g^{\mathrm{LE}}$, $g^{\theta\text{-E}}$, $g^{\mathrm{LC}}$, $g^{\mathrm{BW}}$, and $g^{M\text{-BW}}$, respectively. Given SPD matrices $P, Q, M \in \mathcal{S}_{++}^n$, and tangent vectors $V, W \in T_P \mathcal{S}_{++}^n$ in the tangent space at $P$, the metric tensors are defined as

$$g_P^{\mathrm{AI}}(V, W) = \langle P^{-1}V, WP^{-1}\rangle, \qquad\qquad g_P^{\mathrm{LE}}(V, W) = \langle \mathrm{mlog}_{*,P}(V), \mathrm{mlog}_{*,P}(W)\rangle,$$
$$g_P^{\theta\text{-E}}(V, W) = \frac{1}{\theta^2}\langle \mathrm{Pow}_{\theta*,P}\, V, \mathrm{Pow}_{\theta*,P}\, W\rangle, \quad g_P^{\mathrm{LC}}(V, W) = \langle \lfloor X \rfloor, \lfloor Y \rfloor \rangle + \langle \mathbb{L}^{-1}\mathbb{X}, \mathbb{L}^{-1}\mathbb{Y}\rangle, \tag{1}$$

$$g_P^{M\text{-BW}}(V, W) = \frac{1}{2}\langle \mathcal{L}_{P,M}(V), W\rangle. \tag{2}$$

When $M = I$ is the identity matrix, $M$-BWM becomes BWM. The following explains the notation.

- $\langle \cdot, \cdot \rangle$ is the standard Frobenius inner product.
- Chol, mlog, and $\mathrm{Pow}_\theta$ represent the Cholesky decomposition, matrix logarithm, and matrix power. Their differential maps at $P$ are $\mathrm{Chol}_{*,P}$, $\mathrm{Pow}_{\theta*,P}$, and $\mathrm{mlog}_{*,P}$, respectively.
- $X = \mathrm{Chol}_{*,P}(V)$, $Y = \mathrm{Chol}_{*,P}(W)$, and $L = \mathrm{Chol}(P)$.
- $\lfloor \cdot \rfloor$ is the strictly lower part of a square matrix.
- $\mathbb{X}$, $\mathbb{Y}$, and $\mathbb{L}$ are diagonal matrices with diagonal elements of $X$, $Y$, and $L$.
- $\mathcal{L}_{P,M}(V)$ is the generalized Lyapunov operator: $M\mathcal{L}_{P,M}(V)P + P\mathcal{L}_{P,M}(V)M = V$. When $M = I$, $\mathcal{L}_{P,I}[V]$ is reduced to the Lyapunov operator, denoted $\mathcal{L}_P(V)$.

**Cholesky geometries.** The Euclidean space of $n \times n$ lower triangular matrices is denoted $\mathcal{L}^n$. Its open subset, whose diagonal elements are all positive, is denoted by $\mathcal{L}_{++}^n$. The Cholesky space $\mathcal{L}_{++}^n$ forms a submanifold of $\mathcal{L}^n$ (Lin, 2019). For a Cholesky matrix $L \in \mathcal{L}_{++}^n$ and tangent vectors $X, Y \in T_L \mathcal{L}_{++}^n$, the Riemannian metric on the Cholesky manifold, referred to as the diagonal log metric, is

$$g_L^{\mathrm{DL}}(X, Y) = \langle \lfloor X \rfloor, \lfloor Y \rfloor \rangle + \langle \mathbb{L}^{-1}\mathbb{X}, \mathbb{L}^{-1}\mathbb{Y}\rangle. \tag{3}$$

LCM is the pullback metric of $g^{\mathrm{DL}}$ by the Cholesky decomposition. As shown by Chen et al. (2024d, Thm. III.1.), the diagonal log metric is the pullback metric, by the diagonal log map, of the Euclidean metric over $\mathcal{L}^n$, which rationalizes our nomenclature.

## 3 PRODUCT GEOMETRIES ON THE CHOLESKY

We first unveil the product structure beneath the existing diagonal log metric on the Cholesky manifold. Based on this, we propose two novel Cholesky metrics.

### 3.1 DISENTANGLING THE CHOLESKY GEOMETRY

We denote $n \times n$ diagonal matrices with positive diagonal elements as $\mathbb{D}_{++}^n$, and $n \times n$ strictly lower triangular matrices as $\mathcal{SL}^n$. Then, $\mathcal{SL}^n$ is a Euclidean space, and $\mathbb{D}_{++}^n \cong \mathbb{R}_{++}^n$ is an open submanifold of $\mathbb{R}^n$. Recalling Eq. (3), it is defined separately on $\mathcal{SL}^n$ and $\mathbb{D}_{++}^n$. Besides, $\mathbb{D}_{++}^n$ can be identified as the product of $n$ copies of $\mathbb{R}_{++}$. The above discussions imply a product structure. We denote the standard Euclidean metric over $\mathcal{SL}^n$ as $g^{\mathrm{E}}$, and define the Riemannian metric on $\mathbb{R}_{++}$ as

$$g_p^{\mathbb{R}++}(v, w) = p^{-2}vw, \quad \forall p \in \mathbb{R}_{++} \text{ and } v, w \in T_p \mathbb{R}_{++}. \tag{4}$$

Then, $\mathcal{L}_{++}^n$ is the product manifold of $\mathcal{SL}^n$ and $n$ copies of $\mathbb{R}_{++}$:

$$\{\mathcal{L}_{++}^n, g^{\mathrm{DL}}\} = \{\mathcal{SL}^n, g^{\mathrm{E}}\} \times \overbrace{\{\mathbb{R}_{++}, g^{\mathbb{R}++}\} \times \cdots \times \{\mathbb{R}_{++}, g^{\mathbb{R}++}\}}^{n}. \tag{5}$$

### 3.2 PRODUCT GEOMETRIES ON THE CHOLESKY

The following definition characterizes the underlying product structure in Eq. (5).

**Definition 3.1** (Product geometries). Supposing $g^{\mathcal{SL}}$ is a Euclidean inner product on $\mathcal{SL}^n$ and $\{g^i\}_{i=1}^n$ are Riemannian metrics on $\mathbb{R}_{++}$, then the weighted product metric $g$ on $\mathcal{L}_{++}^n$ is defined as $g_L(X,Y) = g^{\mathcal{SL}}(\lfloor X \rfloor, \lfloor Y \rfloor) + \sum_{i=1}^n \alpha_i g_{L_{ii}}^i(X_{ii}, Y_{ii})$, with $L \in \mathcal{L}_{++}^n$, $X, Y \in T_L\mathcal{L}_{++}^n$ and $\alpha_i > 0$. Here, $\lfloor X \rfloor$ and $\lfloor Y \rfloor$ are strictly lower triangular parts of $X$ and $Y$, while $L_{ii}$, $X_{ii}$ and $Y_{ii}$ are the $i$-th diagonal elements.

For simplicity, we focus on the case where $g^{\mathcal{SL}} = g^{\mathrm{E}}$ is the standard Euclidean metric, all $\alpha_i$ are equal to 1, and all $g^i$ are identical. Since $\mathbb{R}_{++}$ can be viewed as a 1-dimensional SPD manifold $\mathcal{S}_{++}^1$, the existing Riemannian metrics on the SPD manifold can be immediately used to build Riemannian metrics on the Cholesky manifold. Simple computations show that AIM, LEM, and LCM coincide with Eq. (4) on $\mathcal{S}_{++}^1$. Therefore, the six metrics introduced in Sec. 2.2 are reduced to three classes on $\mathcal{S}_{++}^1$: (1) LEM, LCM, or AIM; (2) $\theta$-EM; (3) GBWM and BWM. When the metric on $\mathbb{R}_{++}$ is AIM (LEM or LCM), the resulting product metric on the Cholesky manifold is the diagonal log metric, and the pullback SPD metric via the Cholesky decomposition is exactly LCM.

Inspired by the above analysis, we obtain two new metrics on the Cholesky manifold by setting each $g^i$ in Def. 3.1 as $\theta$-EM and GBWM, termed Diagonal Power Metric ($\theta$-DPM) and Diagonal Bures-Wasserstein Metric ($\mathbb{M}$-DBWM with $\mathbb{M} \in \mathbb{D}_{++}^n$), respectively. By product geometries (Lee, 2018), we can obtain the closed-form expressions of their Riemannian operators, such as geodesic, logarithm, parallel transport, and weighted Fréchet mean. These operators are significantly important in building concrete learning algorithms (Yuan et al., 2012; Lezcano Casado, 2019; Brooks et al., 2019; López et al., 2021).

**Theorem 3.2** ($\theta$-DPM). [↓] *Supposing* $L, K \in \mathcal{L}_{++}^n$, $X, Y \in T_L\mathcal{L}_{++}^n$, *and* $\{L_i \in \mathcal{L}_{++}^n\}_{i=1}^N$ *with weights* $\{w_i\}_{i=1}^N$ *satisfying* $w_i > 0$ *for all* $i$ *and* $\sum_{i=1}^N w_i = 1$, *the Riemannian operators under* $\theta$-*DPM with* $\theta \neq 0$ *are*

$$g_L^{\theta\text{-}DE}(X,Y) = \langle \lfloor X \rfloor, \lfloor Y \rfloor \rangle + \langle \mathbb{L}^{\theta-1}\mathbb{X}, \mathbb{L}^{\theta-1}\mathbb{Y} \rangle, \quad \gamma_{(L,X)}(t) = \lfloor L \rfloor + t\lfloor X \rfloor + \mathbb{L}\left(I + t\theta\mathbb{L}^{-1}\mathbb{X}\right)^{\frac{1}{\theta}},$$

$$\mathrm{Log}_L(K) = \lfloor K \rfloor - \lfloor L \rfloor + \tfrac{1}{\theta}\mathbb{L}\left[\left(\mathbb{L}^{-1}\mathbb{K}\right)^\theta - I\right], \quad \mathrm{PT}_{L\to K}(X) = \lfloor X \rfloor + \left(\mathbb{L}^{-1}\mathbb{K}\right)^{1-\theta}\mathbb{X},$$

$$\mathrm{d}^2(L,K) = \|\lfloor K \rfloor - \lfloor L \rfloor\|_{\mathrm{F}}^2 + \tfrac{1}{\theta^2}\left\|\mathbb{K}^\theta - \mathbb{L}^\theta\right\|_{\mathrm{F}}^2, \quad \mathrm{WFM}(\{w_i\}, \{L_i\}) = \sum_i w_i\lfloor L_i \rfloor + \left(\sum_i w_i\mathbb{L}_i^\theta\right)^{\frac{1}{\theta}},$$

*where* $\|\cdot\|_{\mathrm{F}}$ *is the Frobenius norm.* $\mathbb{X}, \mathbb{Y}, \mathbb{L}, \mathbb{K}$, *and* $\mathbb{L}_i$ *are diagonal matrices with diagonal elements from* $X, Y, L, K$, *and* $L_i$. $\gamma_{(L,X)}(t)$ *denotes the geodesic starting at* $L$ *with initial velocity* $X$. $\mathrm{PT}_{L\to K}(\cdot)$ *is the parallel transportation along the geodesic connecting* $L$ *and* $K$. Log, d, WFM *are the Riemannian logarithm, geodesic distance, and weighted Fréchet mean. Note that* $\gamma_{(L,X)}(t)$ *is locally defined in* $\{t \in \mathbb{R}|\mathbb{L} + t\theta\mathbb{X} \in \mathbb{D}_{++}^n\}$.

**Theorem 3.3** ($\mathbb{M}$-DBWM). [↓] *Following the notation in Thm. 3.2, the Riemannian operators under* $\mathbb{M}$-*DBWM with* $\mathbb{M} \in \mathbb{D}_{++}^n$ *are*

$$g_L^{\mathbb{M}\text{-}DBW}(X,Y) = \langle \lfloor X \rfloor, \lfloor Y \rfloor \rangle + \tfrac{1}{4}\langle \mathbb{L}^{-1}\mathbb{X}, \mathbb{M}^{-1}\mathbb{Y} \rangle, \quad \gamma_{(L,X)}(t) = \lfloor L \rfloor + t\lfloor X \rfloor + \mathbb{L}\left(I + t\tfrac{1}{2}\mathbb{L}^{-1}\mathbb{X}\right)^2,$$

$$\mathrm{Log}_L(K) = \lfloor K \rfloor - \lfloor L \rfloor + 2\mathbb{L}\left[\left(\mathbb{L}^{-1}\mathbb{K}\right)^{\frac{1}{2}} - I\right], \quad \mathrm{PT}_{L\to K}(X) = \lfloor X \rfloor + \left(\mathbb{L}^{-1}\mathbb{K}\right)^{\frac{1}{2}}\mathbb{X},$$

$$\mathrm{d}^2(L,K) = \|\lfloor K \rfloor - \lfloor L \rfloor\|_{\mathrm{F}}^2 + \left\|\mathbb{M}^{-\frac{1}{2}}\left(\mathbb{K}^{\frac{1}{2}} - \mathbb{L}^{\frac{1}{2}}\right)\right\|_{\mathrm{F}}^2, \quad \mathrm{WFM}(\{w_i\}, \{L_i\}) = \sum_i w_i\lfloor L_i \rfloor + \left(\sum_i w_i\mathbb{L}_i^{\frac{1}{2}}\right)^2,$$

*where the geodesic* $\gamma_{(L,X)}(t)$ *is locally defined in* $\{t \in \mathbb{R}|\mathbb{L} + \tfrac{t}{2}\mathbb{X} \in \mathbb{D}_{++}^n\}$. *When* $\mathbb{M} = I$ *in* $\mathbb{M}$-*DBWM, the resulting metric is denoted as DBWM.*

On the SPD manifold, GBWM is locally AIM (Han et al., 2023). Similarly, on the Cholesky manifold, our $\mathbb{M}$-DBWM is locally the diagonal log metric at $L \in \mathcal{L}_{++}^n$: $g_L^{\mathbb{L}\text{-}DBW}(X,Y) = \langle \lfloor X \rfloor, \lfloor Y \rfloor \rangle + \tfrac{1}{4}\langle \mathbb{L}^{-1}\mathbb{X}, \mathbb{L}^{-1}\mathbb{Y} \rangle$. Besides, GBWM on the SPD manifold generally has no closed-form expression for the Fréchet mean (Bhatia et al., 2019). Besides, the closed-form expression of parallel transportation under BWM is known only if two SPD matrices commute (Thanwerdas and Pennec, 2023). In contrast, all these operators have closed-form expressions under $\mathbb{M}$-DBWM on $\mathcal{L}_{++}^n$.

### 3.3 DEFORMED CHOLESKY METRICS

On SPD manifolds, the deformed metrics by the matrix power can interpolate between a given metric and an LEM-like metric (Thanwerdas and Pennec, 2022, Sec. 3.1). Inspired by this, we define a diagonal power deformation on the Cholesky manifold. We denote the diagonal power as

$\text{DPow}_\theta : \mathbb{D}^n_{++} \ni \mathbb{P} \longmapsto \mathbb{P}^\theta \in \mathbb{D}^n_{++}$. We will show how our proposed metric is connected to the existing diagonal log metric by $\text{DPow}_\theta$.

**Definition 3.4.** Let $\{\mathcal{L}^n_{++}, g\} = \{\mathcal{SL}^n, g^{\text{E}}\} \times \{\mathbb{D}^n_{++}, \tilde{g}\}$ be a product metric. We defined the diagonal-power-deformed metrics of $g$ as $\{\mathcal{L}^n_{++}, g^\theta\} = \{\mathcal{SL}^n, g^{\text{E}}\} \times \{\mathbb{D}^n_{++}, \frac{1}{\theta^2} \text{DPow}^*_\theta \tilde{g}\}$.

The following lemma shows that $g^\theta$ in Def. 3.4 tends to be a diagonal-log-like metric with $\theta \to 0$.

**Lemma 3.5.** [↓] *Given $L \in \mathcal{L}^n_{++}$ and $X, Y \in T_L \mathcal{L}^n_{++}$, $g^\theta$ in Def. 3.4 satisfies*

$$g^\theta_L(X, Y) = \langle \lfloor X \rfloor, \lfloor Y \rfloor \rangle + \tilde{g}_{\mathbb{L}^\theta} \left( \mathbb{L}^{\theta-1} \mathbb{X}, \mathbb{L}^{\theta-1} \mathbb{Y} \right) \xrightarrow[\theta \to 0]{} \langle \lfloor X \rfloor, \lfloor Y \rfloor \rangle + \tilde{g}_I (\mathbb{L}^{-1} \mathbb{X}, \mathbb{L}^{-1} \mathbb{Y}). \quad (6)$$

Now, we discuss the deformation on diagonal log metric, $\theta$-DPM, and $\mathbb{M}$-DBWM. Firstly, Eq. (6) indicates that the diagonal-power-deformed metric of diagonal log metric is itself. Secondly, $\theta$-DPM is the diagonal-power-deformed metric of the Euclidean Metric on the Cholesky manifold. Besides, $\theta$-DPM interpolates between the diagonal log metric ($\theta \to 0$) and Euclidean Metric ($\theta = 1$). Thirdly, the diagonal-power-deformed metric of $\mathbb{M}$-DBWM, referred to as $(\theta, \mathbb{M})$-DBWM, is $g^{(\theta, \mathbb{M})\text{-DBW}}_L(X, Y) = \langle \lfloor X \rfloor, \lfloor Y \rfloor \rangle + \frac{1}{4} \langle \mathbb{L}^{\theta-2} \mathbb{X}, \mathbb{M}^{-1} \mathbb{Y} \rangle$. When $\mathbb{M} = I$, the deformed metric of DBWM, *i.e.*, $\theta$-DBWM, tends to be scaled diagonal log metric as $\theta \to 0$: $g^{\theta\text{-DBW}}_L(X, Y) = \langle \lfloor X \rfloor, \lfloor Y \rfloor \rangle + \frac{1}{4} \langle \mathbb{L}^{-1} \mathbb{X}, \mathbb{L}^{-1} \mathbb{Y} \rangle$. As $(\theta, \mathbb{M})$-DBWM is the pullback metric by diagonal power and scaled by a constant, the Riemannian operators also have closed-form expressions, which are discussed in Sec. D.

## 3.4 ALGEBRAIC STRUCTURES

The gyrovector space forms the algebraic basis for hyperbolic geometry (Ungar, 2005) as the vector space for the Euclidean space. It has recently been extended to matrix manifolds (Nguyen, 2022). Given $P$ and $Q$ in the Riemannian manifold $\mathcal{M}$, the gyroaddition and gyromultiplication (Nguyen and Yang, 2023, Eqs.1 and 2) are defined as

$$\text{Binary operation: } P \oplus Q = \text{Exp}_P \left( \text{PT}_{E \to P}(\text{Log}_E(Q)) \right), \quad (7)$$

$$\text{Gyromultiplication: } t \odot P = \text{Exp}_E(t \, \text{Log}_E(P)), \quad (8)$$

where $E$ is the origin of $\mathcal{M}$. For more details on gyrovector spaces, please refer to Sec. C.1. The gyro operations under diagonal log metric have been studied by Nguyen (2022). This subsection studies the gyro-structures over $\theta$-DPM and $(\theta, \mathbb{M})$-DBWM.

Let the identity matrix be the origin and $\mathcal{C}$ be $\theta$-DPM or $(\theta, \mathbb{M})$-DBWM. We have the following.

**Lemma 3.6** (Gyro-structures). [↓] *For $L, K \in \mathcal{L}^n_{++}$ and $t \in \mathbb{R}$, the gyro operations are*

$$L \oplus^\mathcal{C} K = \lfloor L \rfloor + \lfloor K \rfloor + \left( \mathbb{L}^\beta + \mathbb{K}^\beta - I \right)^{\frac{1}{\beta}}, \quad (9)$$

$$t \odot^\mathcal{C} L = t\lfloor L \rfloor + \left( t\mathbb{L}^\beta + (1-t)I \right)^{\frac{1}{\beta}}, \quad (10)$$

*where $\beta = \theta$ for $\theta$-DPM, and $\beta = \theta/2$ for $(\theta, \mathbb{M})$-DBWM. $\oplus^\mathcal{C}$ requires $L$ and $K$ to satisfy $\mathbb{L}^\beta + \mathbb{K}^\beta - I \in \mathbb{D}^n_{++}$, while $\odot^\mathcal{C}$ requires $(1-t)I + t\mathbb{L}^\beta \in \mathbb{D}^n_{++}$.*

Note that gyro operations can only be defined under the assumptions in Lem. 3.6, which comes from the locally defined Riemannian exp. In the following, we make these assumptions implicitly.

**Theorem 3.7.** [↓] *$\{\mathcal{L}^n_{++}, \oplus^\mathcal{C}\}$ conforms with all the axioms of gyrocommutative gyrogroups (Defs. C.1 and C.2). $\{\mathcal{L}^n_{++}, \oplus^\mathcal{C}, \odot^\mathcal{C}\}$ conforms with all the axioms of gyrovector spaces (Def. C.3).*

**Corollary 3.8.** *The identity element of $\{\mathcal{L}^n_{++}, \oplus^\mathcal{C}\}$ is the identity matrix, i.e., $\forall L \in \mathcal{L}^n_{++}, I \oplus^\mathcal{C} L = L$. The inverses are $\ominus^\mathcal{C} L = -1 \odot^\mathcal{C} L = -\lfloor L \rfloor + \left( 2I - \mathbb{L}^\beta \right)^{\frac{1}{\beta}}$, for $L \in \{L \in \mathcal{L}^n_{++} | 2I - \mathbb{L}^\beta \in \mathbb{D}^n_{++}\}$.*

*Remark* 3.9. The gyro-structures on the Grassmann manifold have shown success in building Riemannian algorithms (Nguyen, 2022; Nguyen and Yang, 2023). Like our gyro-structure, the gyro-structures of the Grassmann manifold also require some assumptions for well-definedness (Nguyen, 2022, Sec. 3.2). Therefore, we also discuss the gyro-structures under our metrics for completeness. In practice, such positivity constraints can be easily remedied by numeric tricks. Taking $2I - \mathbb{L}^\beta \in \mathbb{D}^n_{++}$ as an example, one can use $d_i \leftarrow \max(d_i, \varepsilon)$ for each diagonal element $d_i$ with a small constant $\varepsilon > 0$.

Table 1: Riemannian and gyro operators of different metrics on the Cholesky manifold. For diagonal log metric, $\log(\cdot)$ and $\exp(\cdot)$ are diagonal logarithm and exponentiation.

| Operators | Diagonal Log Metric | $\theta$-DPM | $(\theta, \mathbb{M})$-DBWM |
|---|---|---|---|
| $g_L(X,Y)$ | $\lfloor X \rfloor + \lfloor Y \rfloor + \langle \mathbb{L}^{-1}\mathbb{X}, \mathbb{L}^{-1}\mathbb{Y} \rangle$ | $\langle \lfloor X \rfloor, \lfloor Y \rfloor \rangle + \langle \mathbb{L}^{\theta-1}\mathbb{X}, \mathbb{L}^{\theta-1}\mathbb{Y} \rangle$ | $\langle \lfloor X \rfloor, \lfloor Y \rfloor \rangle + \frac{1}{4}\langle \mathbb{L}^{\theta-2}\mathbb{X}, \mathbb{M}^{-1}\mathbb{Y} \rangle$ |
| $\gamma_{(L,X)}(t)$ | $\lfloor L \rfloor + t\lfloor X \rfloor + \mathbb{L}\exp(t\mathbb{L}^{-1}\mathbb{X})$ | $\lfloor L \rfloor + t\lfloor X \rfloor + \mathbb{L}\left(I + t\theta\mathbb{L}^{-1}\mathbb{X}\right)^{\frac{1}{\theta}}$ | $\lfloor L \rfloor + t\lfloor X \rfloor + \mathbb{L}\left(I + t\frac{\theta}{2}\mathbb{L}^{-1}\mathbb{X}\right)^{\frac{2}{\theta}}$ |
| $\mathrm{Log}_L(K)$ | $\lfloor K \rfloor - \lfloor L \rfloor + \mathbb{L}\log(\mathbb{L}^{-1}\mathbb{K})$ | $\lfloor K \rfloor - \lfloor L \rfloor + \frac{1}{\theta}\mathbb{L}\left[\left(\mathbb{L}^{-1}\mathbb{K}\right)^{\theta} - I\right]$ | $\lfloor K \rfloor - \lfloor L \rfloor + \frac{2}{\theta}\mathbb{L}\left[\left(\mathbb{L}^{-1}\mathbb{K}\right)^{\frac{\theta}{2}} - I\right]$ |
| $\mathrm{PT}_{L\to K}(X)$ | $\lfloor X \rfloor + (\mathbb{L}^{-1}\mathbb{K})\mathbb{X}$ | $\lfloor X \rfloor + (\mathbb{L}^{-1}\mathbb{K})^{1-\theta}\mathbb{X}$ | $\lfloor X \rfloor + (\mathbb{L}^{-1}\mathbb{K})^{1-\frac{\theta}{2}}\mathbb{X}$ |
| $\mathrm{d}^2(L,K)$ | $\|\lfloor K \rfloor - \lfloor L \rfloor\|_F^2 + \|\log(\mathbb{K}) - \log(\mathbb{L})\|_F^2$ | $\|\lfloor K \rfloor - \lfloor L \rfloor\|_F^2 + \frac{1}{\theta^2}\|\mathbb{K}^{\theta} - \mathbb{L}^{\theta}\|_F^2$ | $\|\lfloor K \rfloor - \lfloor L \rfloor\|_F^2 + \frac{1}{\theta^2}\left\|\mathbb{M}^{-\frac{1}{2}}\left(\mathbb{K}^{\frac{\theta}{2}} - \mathbb{L}^{\frac{\theta}{2}}\right)\right\|_F^2$ |
| $\mathrm{WFM}(\{w_i\}, \{L_i\})$ | $\sum_i w_i\lfloor L_i \rfloor + \exp\left(\sum_i w_i \log(\mathbb{L}_i)\right)$ | $\sum_i w_i\lfloor L_i \rfloor + \left(\sum_i w_i\mathbb{L}_i^{\theta}\right)^{\frac{1}{\theta}}$ | $\sum_i w_i\lfloor L_i \rfloor + \left(\sum_i w_i\mathbb{L}_i^{\frac{\theta}{2}}\right)^{\frac{2}{\theta}}$ |
| $L \oplus K$ | $\lfloor L \rfloor + \lfloor K \rfloor + \mathbb{L}\mathbb{K}$ | $\lfloor L \rfloor + \lfloor K \rfloor + \left(\mathbb{L}^{\theta} + \mathbb{K}^{\theta} - I\right)^{\frac{1}{\theta}}$ | $\lfloor L \rfloor + \lfloor K \rfloor + \left(\mathbb{L}^{\frac{\theta}{2}} + \mathbb{K}^{\frac{\theta}{2}} - I\right)^{\frac{2}{\theta}}$ |
| $t \odot L$ | $t\lfloor L \rfloor + \mathbb{L}^t$ | $t\lfloor L \rfloor + \left(t\mathbb{L}^{\theta} + (1-t)I\right)^{\frac{1}{\theta}}$ | $t\lfloor L \rfloor + \left(t\mathbb{L}^{\frac{\theta}{2}} + (1-t)I\right)^{\frac{2}{\theta}}$ |

## 3.5 Numerical advantages over diagonal log metric

Tab. 1 summarizes all the Riemannian and gyro operators. The Riemannian operators under $(\theta, \mathbb{M})$-DBWM and $\theta$-DPM are mostly calculated by *the diagonal power function*, while the ones under diagonal log metric are computed by *the diagonal logarithm or exponentiation*. This indicates that our $\theta$-DPM and $(\theta, \mathbb{M})$-DBWM could have better numerical stability than the existing diagonal log metric, as logarithm or exponentiation might overly stretch the diagonal elements compared with the power function. The gyro operations under our $\theta$-DPM and $(\theta, \mathbb{M})$-DBWM also have numerical advantages over the ones under the diagonal log metric. The former is based on linear operations combined with power and its inverse, causing relatively minor changes to the input magnitude, while the gyro operations under diagonal log metric are based on product or power, resulting in more noticeable alterations to the input magnitude.

## 4 Geometries on the SPD manifold

This section discusses the Riemannian metrics on the SPD manifold via the Cholesky decomposition. We first review some basic properties of the Cholesky decomposition, followed by the SPD metrics.

The Cholesky decomposition, denoted as $\mathrm{Chol}(\cdot) : \mathcal{S}_{++}^n \to \mathcal{L}_{++}^n$, is a diffeomorphism (Lin, 2019). Therefore, it can pull back the Riemannian and gyro-structures from the Cholesky manifold $\mathcal{L}_{++}^n$ to the SPD manifold $\mathcal{S}_{++}^n$. We denote the pullback metrics on $\mathcal{S}_{++}^n$ of $\theta$-DPM and $(\theta, \mathbb{M})$-DBWM on $\mathcal{L}_{++}^n$ by the Cholesky decomposition as Power Cholesky Metric ($\theta$-PCM) and Bures-Wasserstein Cholesky Metric ($(\theta, \mathbb{M})$-BWCM). Then, the Riemannian operators under $\theta$-PCM and $(\theta, \mathbb{M})$-BWCM can be obtained by the properties of Riemannian isometry (Sec. C.2).

**Differentials.** For $P \in \mathcal{S}_{++}^n$ with Cholesky decomposition $P = LL^\top$, the differential map (Lin, 2019) at $P$ is $\mathrm{Chol}_{*,P}(\cdot) : T_P\mathcal{S}_{++}^n \ni V \mapsto L\left(L^{-1}VL^{-\top}\right)_{\frac{1}{2}} \in T_L\mathcal{L}_{++}^n$, with $(V)_{\frac{1}{2}} = \lfloor V \rfloor + \frac{1}{2}\mathbb{V}$.

**Operators.** Let $\mathcal{C} \in \{\theta\text{-DPM}, (\theta, \mathbb{M})\text{-DBWM}\}$ and $\mathcal{S} \in \{\theta\text{-PCM}, (\theta, \mathbb{M})\text{-BWCM}\}$. We denote $\mathrm{Log}^{\mathcal{S}}, \mathrm{Exp}^{\mathcal{S}}, \gamma^{\mathcal{S}}, \mathrm{PT}^{\mathcal{S}}, \mathrm{d}^{\mathcal{S}}(\cdot, \cdot), \mathrm{WFM}^{\mathcal{S}}, \oplus^{\mathcal{S}}$, and $\odot^{\mathcal{S}}$ as the Riemannian logarithm, exponentiation, geodesic, parallel transportation along the geodesic, geodesic distance, and weighted Fréchet mean on $\{\mathcal{S}_{++}^n, g^{\mathcal{S}}\}$, while $\mathrm{Log}^{\mathcal{C}}, \mathrm{Exp}^{\mathcal{C}}, \gamma^{\mathcal{C}}, \mathrm{PT}^{\mathcal{C}}, \mathrm{d}^{\mathcal{C}}(\cdot, \cdot), \mathrm{WFM}^{\mathcal{C}}, \oplus^{\mathcal{C}}$, and $\odot^{\mathcal{C}}$ are the counterparts on $\{\mathcal{L}_{++}^n, g^{\mathcal{C}}\}$. For $P, Q \in \mathcal{S}_{++}^n$, $V, W \in T_P\mathcal{M}$, and $\{P_i \in \mathcal{M}\}_{i=1}^N$ with weights $\{w_i\}_{i=1}^N$ satisfying $w_i > 0$ for all $i$ and $\sum_{i=1}^N w_i = 1$, we have the following w.r.t. the Riemannian and gyro operators:

$$
\begin{aligned}
\gamma_{(P,V)}^{\mathcal{S}}(t) &= \mathrm{Chol}^{-1}\left(\gamma_{(L,\widetilde{V})}^{\mathcal{C}}(t)\right), & \mathrm{Log}_P^{\mathcal{S}}(Q) &= (\mathrm{Chol}_{*,P})^{-1}\left(\mathrm{Log}_L^{\mathcal{C}}(K)\right), \\
\mathrm{Exp}_P^{\mathcal{S}}(V) &= \mathrm{Chol}^{-1}\left(\mathrm{Exp}_L^{\mathcal{C}}\left(\widetilde{V}\right)\right), & \mathrm{PT}_{P\to Q}^{\mathcal{S}}(V) &= (\mathrm{Chol}_{*,Q})^{-1}\left(\mathrm{PT}_{L\to K}^{\mathcal{C}}\left(\widetilde{V}\right)\right), \\
\mathrm{d}^{\mathcal{S}}(P,Q) &= \mathrm{d}^{\mathcal{C}}(L,K), & \mathrm{WFM}^{\mathcal{S}}(\{P_i\}, \{w_i\}) &= \mathrm{Chol}^{-1}\left(\mathrm{WFM}^{\mathcal{C}}\left(\{L_i\}, \{w_i\}\right)\right), \\
P \oplus^{\mathcal{S}} Q &= \mathrm{Chol}^{-1}(L\widetilde{\oplus}^{\mathcal{C}}K), & t \odot^{\mathcal{S}} P &= \mathrm{Chol}^{-1}(t\widetilde{\odot}^{\mathcal{C}}L),
\end{aligned}
\tag{11}
$$

where $P = LL^\top$, $Q = KK^\top$, and $P_i = L_i L_i^\top$ are Cholesky decompositions. Here, $\widetilde{V} = \mathrm{Chol}_{*,P}(V)$ and $\mathrm{Chol}^{-1}(L) = LL^\top$. Besides, when $\mathcal{C}$ is the diagonal log metric, the above recovers the Riemannian and gyro-structures under LCM.

The above gyro operations, when well-defined, also conform with gyrovector space axioms.

**Theorem 4.1.** [↓] $\{\mathcal{S}_{++}^n, \oplus^{\mathcal{S}}, \odot^{\mathcal{S}}\}$ *conform with all the axioms of gyrovector spaces.*

*Remark* 4.2. As discussed in Sec. 3.5, the Cholesky metric $\theta$-DPM and $(\theta, \mathbb{M})$-DBWM are more numerically stable than the existing diagonal log metric. As the pullback metrics by the Cholesky decomposition, our $\theta$-PCM and $(\theta, \mathbb{M})$-BWCM, therefore, preserve the advantage of numeric stability over the existing LCM. Besides, all the Riemannian operators have closed-form expressions and are easy to use, as the differential maps of the Cholesky decomposition can be easily calculated.

## 5 APPLICATIONS TO SPD NEURAL NETWORKS

As discussed in Sec. 1, Riemannian metrics enable the construction of basic building blocks in SPD neural networks. In this section, we apply our proposed $\theta$-DPM and $(\theta, \mathbb{M})$-DBWM to build Multinomial Logistics Regression (MLR) classifiers and residual blocks on the SPD manifold.

**MLR.** A typical classification layer in Euclidean networks is MLR, $\mathrm{Softmax}(Ax + b)$, which computes the multinomial probability of each class. Given $C$ classes and input $x \in \mathbb{R}^n$, MLR can be written by the point-to-hyperplane distance (Lebanon and Lafferty, 2004, Sec. 5):

$$\forall k \in \{1, \ldots, C\}, \quad p(y = k \mid x) \propto \exp\left(\mathrm{sign}(\langle a_k, x - p_k\rangle)\|a_k\| \, \mathrm{d}(x, H_{a_k, p_k})\right) \quad (12)$$

where $a_k, p_k \in \mathbb{R}^n$, and $H_{a_k, p_k} = \{x \in \mathbb{R}^n \mid \langle a_k, x - p_k\rangle = 0\}$ is a hyperplane with $\mathrm{d}(x, H_{a_k, p_k})$ as the point-to-hyperplane distance. Chen et al. (2024c, Eqs. 3-4) extended Eq. (12) into manifolds by Riemannian reinterpretation. Given input $X \in \mathcal{M}$, the Riemannian MLR is

$$p(y = k \mid X) \propto \exp\left(\mathrm{sign}(\langle A_k, \mathrm{Log}_{P_k}(X)\rangle_{P_k})\|A_k\|_{P_k} \, \mathrm{d}(X, H_{A_k, P_k})\right), \quad \forall X \in \mathcal{M} \quad (13)$$

where $P_k \in \mathcal{M}$ and $\tilde{A}_k \in T_{P_k}\mathcal{M}$ are parameters. Following Chen et al. (2024c, Thm. 3.3), we can obtain the SPD MLR under our metrics.

**Theorem 5.1.** [↓] *Given an input SPD matrix $S \in \mathcal{S}_{++}^n$, the $C$-class SPD MLRs under $\theta$-PCM and $(\theta, \mathbb{M})$-BWCM are*

$$\theta\text{-PCM} : p(y = k \mid S \in \mathcal{S}_{++}^n) \propto \exp\left[\langle \lfloor K \rfloor - \lfloor L_k \rfloor, \lfloor A_k \rfloor\rangle + \frac{1}{2\theta}\langle \mathbb{K}^\theta - \mathbb{L}_k^\theta, \mathbb{A}_k\rangle\right], \quad (14)$$

$$(\theta, \mathbb{M})\text{-BWCM} : p(y = k \mid S \in \mathcal{S}_{++}^n) \propto \exp\left[\langle \lfloor K \rfloor - \lfloor L_k \rfloor, \lfloor A_k \rfloor\rangle + \frac{1}{4\theta}\langle \mathbb{K}^{\frac{\theta}{2}} - \mathbb{L}_k^{\frac{\theta}{2}}, \mathbb{M}^{-1}\mathbb{A}_k\rangle\right], \quad (15)$$

*where $S = KK^\top$ and $P_k = L_k L_k^\top$ are Cholesky decompositions. The parameters are $P_k \in \mathcal{S}_{++}^n$ and $A_k \in \mathcal{L}^n$ for each class $k = 1, \cdots, C$.*

**Residual blocks.** Euclidean residual blocks can be written as $x^{(i)} = x^{(i-1)} + n_i\left(x^{(i-1)}\right)$, where $n_i$ is a network. Katsman et al. (2024) generalized this to manifolds by replacing addition with the Riemannian exponential map $x^{(i)} = \mathrm{Exp}_{x^{(i-1)}}\left(\ell_i(x^{(i-1)})\right)$, where $\ell_i : \mathcal{M} \to T\mathcal{M}$ outputs a vector field parameterized by the neural network. On the SPD manifold, the vector field is generated by the eigenvalues. Specifically, the SPD residual block (Katsman et al., 2024, Eqs. 22-23) is

$$Y = \mathrm{Exp}_X\left(Q \, \mathrm{diag}\left(f(\mathrm{spec}(X))\right) Q^T\right), \quad (16)$$

where $X \in \mathcal{S}_{++}^n$, $\mathrm{spec}(X)$ contains all the eigenvalues, $f : \mathbb{R}^n \to \mathbb{R}^n$ is a neural network, and $Q$ is an orthogonal parameter. Here, $\mathrm{diag}(\cdot)$ returns a diagonal matrix from the input vector. The only component that varies across different metrics is the Riemannian exponentiation. Therefore, we can readily build the residual blocks under our metrics.

## 6 EXPERIMENTS

### 6.1 EXPERIMENTS ON SPD NEURAL NETWORKS

We compare our metrics against the popular AIM, LEM, and LCM on building SPD MLR and residual blocks. Following previous work (Huang and Van Gool, 2017; Chen et al., 2024c), we adopt

Table 2: SPD MLRs under different metrics on the SPDNet backbone. The best two results are highlighted in **red** and **blue**.

(a) Radar

| Metric | Acc | Time |
|---|---|---|
| AIM | **94.53 ± 0.95** | 0.80 |
| LEM | 93.55 ± 1.21 | 0.76 |
| LCM | 93.49 ± 1.25 | 0.72 |
| $\theta$-PCM | **95.79 ± 0.38** | 0.72 |
| $\theta$-BWCM | 93.93 ± 0.79 | 0.71 |

(a) HDM05

| Metric | 1-Block | | 2-Block | | 3-Block | |
|---|---|---|---|---|---|---|
| | Acc | Time | Acc | Time | Acc | Time |
| AIM | 58.07 ± 0.64 | 17.32 | 60.72 ± 0.62 | 18.75 | 61.14 ± 0.94 | 19.23 |
| LEM | 56.97 ± 0.61 | 2.21 | 60.69 ± 1.02 | 2.92 | 60.28 ± 0.91 | 3.50 |
| LCM | 60.69 ± 1.89 | 1.83 | 62.61 ± 1.46 | 2.40 | 62.33 ± 2.15 | 2.90 |
| $\theta$-PCM | **62.51 ± 1.65** | 1.58 | **63.66 ± 1.30** | 2.29 | **65.75 ± 2.86** | 2.76 |
| $\theta$-BWCM | **62.71 ± 0.88** | 1.64 | **64.52 ± 0.56** | 2.27 | **67.40 ± 0.90** | 2.87 |

(c) FPHA

| Metric | Acc | Time |
|---|---|---|
| AIM | 85.57 ± 0.50 | 7.14 |
| LEM | 85.90 ± 0.47 | 0.98 |
| LCM | **86.37 ± 0.59** | 0.74 |
| $\theta$-PCM | **89.40 ± 0.13** | 0.69 |
| $\theta$-BWCM | 86.27 ± 0.60 | 0.70 |

the Radar dataset (Brooks et al., 2019) for radar signal classification, and HDM05 (Müller et al., 2007) & FPHA (Garcia-Hernando et al., 2018) datasets for human action recognition. For more details on datasets and implementation, please refer to Sec. E.1.

### 6.1.1 RIEMANNIAN CLASSIFIERS

We compare the SPD MLRs under our metrics with those under AIM, LEM, and LCM (Chen et al., 2024c, Thm.4.2). Following Chen et al. (2024c); Nguyen and Yang (2023), we adopt SPDNet (Huang et al., 2017) and GyroSPD (Nguyen and Yang, 2023) as two backbones, both mimicking feedforward neural networks. For simplicity, we set $\mathbb{M}$ in $(\theta, \mathbb{M})$-BWCM to the identity matrix.

**SPDNet.** On HDM05, we further evaluate architectures with up to three transformation blocks. Tab. 2 reports the five-fold accuracy and training time per epoch, from which we draw the following observations.

- *Effectiveness.* Our metrics generally yield higher accuracy than their counterparts. Notably, they outperform LCM, although both originate from the Cholesky product structure. This improvement is attributed to the fact that the diagonal logarithm and exponentiation in LCM tend to overly stretch the diagonal entries, *i.e.*, eigenvalues of the Cholesky factors, whereas our diagonal power transformation achieves a more balanced scaling.
- *Efficiency.* Our metrics substantially reduce computational cost compared to AIM, remain faster than LEM, and achieve efficiency comparable to LCM. Together with their superior accuracy, these results highlight the dual advantages of our approach in both effectiveness and efficiency.

**GyroSPD.** Tab. 3 reports the results on the GyroSPD backbone, which consists of a single gyrotranslation layer followed by an SPD MLR. Similar to the SPDNet results, our $\theta$-PCM and $\theta$-BWCM consistently outperform LCM across all datasets while maintaining comparable efficiency. They also deliver higher accuracy with lower runtime than AIM and LEM, particularly AIM.

Table 3: SPD MLRs on the GyroSPD backbone.

| Metric | Radar | | HDM05 | | FPHA | |
|---|---|---|---|---|---|---|
| | Acc | Time | Acc | Time | Acc | Time |
| AIM | **96.80 ± 0.59** | 1.23 | 66.05 ± 1.80 | 21.65 | 85.77 ± 0.52 | 11.48 |
| LEM | 96.58 ± 0.27 | 1.18 | 66.42 ± 0.47 | 2.02 | 85.87 ± 0.79 | 1.22 |
| LCM | 96.29 ± 0.53 | 1.12 | 68.37 ± 0.66 | 1.66 | 89.83 ± 0.28 | 0.98 |
| $\theta$-PCM | **97.04 ± 0.64** | 1.18 | **71.93 ± 1.21** | 1.51 | **91.17 ± 0.30** | 1.00 |
| $\theta$-BWCM | 96.21 ± 0.25 | 1.05 | **72.74 ± 0.43** | 1.58 | **91.00 ± 0.11** | 0.96 |

**Deformation.** In Sec. E.3, we perform a systematic ablation on the deformation factor $\theta$ in $\theta$-PCM and $\theta$-BWCM, sweeping $\theta$ from $-2$ to $1.5$ on both SPDNet and GyroSPD. The results show that, across all datasets, suitably chosen $\theta$ consistently improves accuracy over the default $\theta = 1$, confirming the effectiveness of our diagonal deformation. The gains are modest on Radar and FPHA but much larger on HDM05. Sec. E.4 further attributes this dataset dependence to the Cholesky diagonal statistics of each dataset. Radar and FPHA have relatively balanced Cholesky diagonals, whereas HDM05 has highly imbalanced diagonal entries. In this case, adjusting $\theta$ more effectively activates these under-represented diagonal entries on HDM05.

**Scalability.** Sec. E.5 evaluates the scalability of our metrics through SPD MLR. The synthetic experiments show that our metrics are the most efficient ones. They perform on par with LCM at small and medium dimensions and become clearly faster in high-dimensional settings, while SVD-based metrics such as AIM and BWM become prohibitively slow.

Table 5: Failure probabilities (%) of geodesics under different metrics with small eigenvalues in $L \in \mathcal{L}_{++}^n$. An output matrix containing any INF or NAN is considered a failure. Here, DLM denotes the diagonal log metric, while DPM and DBWM denote $\theta$-DPM and $\theta$-DBWM, respectively.

| | | $3 \times 3$ for small matrices | | | | | | | $256 \times 256$ for large matrices | | | | | |
| | | $\theta = 1.5$ | | $\theta = 0.5$ | | $\theta = 0.15$ | | | $\theta = 1.5$ | | $\theta = 0.5$ | | $\theta = 0.15$ | |
| $\epsilon$ | DLM | DPM | DBWM | DPM | DBWM | DPM | DBWM | DLM | DPM | DBWM | DPM | DBWM | DPM | DBWM |
|---|---|---|---|---|---|---|---|---|---|---|---|---|---|---|
| $1e^{-1}$ | 0.62 | 0 | 0 | 0 | 0 | 0 | 0 | 14.29 | 0 | 0 | 0 | 0 | 0 | 0 |
| $1e^{-2}$ | 5.70 | 0 | 0 | 0 | 0 | 0 | 0 | 18.48 | 0 | 0 | 0 | 0 | 0 | 0 |
| $1e^{-3}$ | 51.32 | 0 | 0 | 0 | 0 | 0 | 0 | 58.35 | 0 | 0 | 0 | 0 | 0 | 0 |
| $1e^{-4}$ | 94.34 | 0 | 0 | 0 | 0 | 0 | 0 | 95.02 | 0 | 0 | 0 | 0 | 0 | 0 |
| $1e^{-5}$ | 99.39 | 0 | 0 | 0 | 0 | 0 | 0 | 99.47 | 0 | 0 | 0 | 0 | 0 | 0 |
| $1e^{-10}$ | 100 | 0 | 0 | 0 | 0 | 0 | 0 | 100 | 0 | 0 | 0 | 0 | 0 | 0 |
| $1e^{-15}$ | 100 | 0 | 0 | 0 | 0 | 0 | 0 | 100 | 0 | 0 | 0 | 0 | 0 | 0 |
| $1e^{-20}$ | 100 | 0 | 0 | 0 | 0 | 0 | 0.002 | 100 | 0 | 0 | 0 | 0 | 0 | 0.02 |
| $1e^{-21}$ | 100 | 0 | 0 | 0 | 0 | 0 | 0.03 | 100 | 0 | 0 | 0 | 0 | 0 | 0.01 |
| $1e^{-22}$ | 100 | 0 | 0 | 0 | 0 | 0 | 0.25 | 100 | 0 | 0 | 0 | 0 | 0 | 0.23 |
| $1e^{-23}$ | 100 | 0 | 0 | 0 | 0 | 0 | 2.26 | 100 | 0 | 0 | 0 | 0 | 0 | 2.42 |
| $1e^{-24}$ | 100 | 0 | 0 | 0 | 0 | 0 | 22.98 | 100 | 0 | 0 | 0 | 0 | 0 | 23.13 |
| $1e^{-25}$ | 100 | 0 | 0 | 0 | 0 | 0 | 86.34 | 100 | 0 | 0 | 0 | 0 | 0 | 86.58 |
| $1e^{-30}$ | 100 | 0 | 0 | 0 | 0 | 0 | 100 | 100 | 0 | 0 | 0 | 0 | 0 | 100 |

### 6.1.2 RIEMANNIAN RESIDUAL BLOCKS

The backbone architecture of Riemannian ResNet (RResNet) (Katsman et al., 2024) largely follows SPDNet. The key difference lies in the head: while SPDNet directly applies a classification layer, RResNet inserts a residual block before the classification head. Following Katsman et al. (2024), we adopt $\mathrm{Log}_I$ + FC + Softmax for classification under each metric.

Since the Riemannian exponential is similar for $\theta$-PCM and $(\theta, \mathbb{M})$-BWCM, we focus on $\theta$-PCM and compare it against AIM, LEM, and LCM in constructing RResNet. Tab. 4 reports the best results across three trials, showing that our metric consistently achieves superior accuracy while maintaining comparable efficiency.

Table 4: Residual blocks under different metrics.

| Metric | Radar | | HDM05 | | FPHA | |
| | Acc | Time | Acc | Time | Acc | Time |
|---|---|---|---|---|---|---|
| AIM | 96.4 | 1.02 | 57.01 | 1.14 | 87.33 | 0.72 |
| LEM | 97.07 | 0.81 | 67.52 | 0.52 | 86.17 | 0.32 |
| LCM | 97.07 | 0.85 | 66.27 | 0.63 | 86.83 | 0.49 |
| $\theta$-PCM | **97.87** | 0.85 | **68.05** | 0.63 | **88.33** | 0.48 |

### 6.2 NUMERICAL EXPERIMENTS

**Numerical stability.** As discussed by Lin (2019, p. 16), LCM is more stable than AIM and LEM owing to the numerical advantage of Cholesky decomposition over SVD. Moreover, as highlighted in Rmk. 4.2, the essential distinction between our metrics and LCM lies in the diagonal operations: ours rely on diagonal power, while LCM employs diagonal exponentiation and logarithm. This structural difference grants our metrics stronger numerical stability and robustness compared with LCM, as well as AIM and LEM. To validate this, we evaluate geodesics on the Cholesky manifold. We generate 100,000 synthetic $n \times n$ Cholesky matrices $L$ and tangent vectors $X \in \mathcal{L}^n$, where each entry is uniformly sampled from $[0, 1]$, and we set the smallest eigenvalue (diagonal entry) of $L$ to $\epsilon$. We test two representative sizes: $3 \times 3$ matrices, commonly used in diffusion tensor imaging (Arsigny et al., 2007), and $256 \times 256$ matrices, typical in computer vision (Li et al., 2018; Wang et al., 2023). The deformation parameter $\theta$ is set to 1.5, 0.5, and 0.15. For $(\theta, \mathbb{M})$-DBWM, we set $\mathbb{M} = I$. As shown in Tab. 5, our $\theta$-DPM and $\theta$-DBWM remain highly stable across a wide range of $\epsilon$. For $3 \times 3$ matrices, the diagonal log metric already deteriorates at $\epsilon = 1e^{-3}$, with failure rates increasing rapidly as $\epsilon$ decreases. For $256 \times 256$ matrices, instability emerges even earlier at $\epsilon = 1e^{-1}$. In contrast, our metrics remain stable down to $\epsilon = 1e^{-30}$ in most cases. The only exception occurs when $\theta = 0.15$, where failures appear around $\epsilon = 1e^{-20}$. This behavior is expected since both $\theta$-DPM and $\theta$-DBWM converge to the diagonal log metric as $\theta \to 0$, thereby inheriting its instability in this limit. Overall, these results demonstrate the superior numerical robustness of our metrics.

**Tensor interpolation.** Geodesic interpolation of SPD matrices is widely used in applications such as diffusion tensor imaging (Arsigny et al., 2007). Experiments in Sec. E.2 show that our $\theta$-PCM mitigates the swelling effect in SPD interpolation observed under Power-Euclidean and Bures-Wasserstein metrics, while producing interpolation patterns visually similar to LCM, thereby retaining the practical potential of LCM.

## 7    CONCLUSION

We identify the product structure in the Cholesky manifold and propose two new Cholesky geometries. Through Cholesky decomposition, these yield two fast and stable Riemannian metrics on the SPD manifold, $\theta$-PCM and $(\theta, \mathbb{M})$-BWCM. We provide a comprehensive analysis of their Riemannian and algebraic properties. Like LCM, our metrics admit efficient closed-form formulas for key operators including geodesics, logarithmic and exponential maps, weighted Fréchet mean, parallel transport, and gyro operations. In addition to efficiency, they offer stronger numerical stability than existing approaches. Extensive experiments on Riemannian classifiers, residual networks, geodesic stability, and tensor interpolation confirm their effectiveness. Together, these results establish our metrics as robust and practical alternatives to existing metrics for SPD matrix learning.

## REPRODUCIBILITY STATEMENT

All theoretical results are established under explicit assumptions, with complete proofs in Sec. F. Experimental details are given for SPD neural networks (Sec. E.1), geodesic stability (Sec. 6.2), and tensor interpolation (Sec. E.2). The code will be released upon acceptance.

## ETHICS STATEMENT

This work uses only publicly available benchmark datasets, which contain no personally identifiable or sensitive information. We do not identify any ethical concerns.

## ACKNOWLEDGMENTS

This work was supported by EU Horizon project ELLIOT (No. 101214398) and by the FIS project GUIDANCE (No. FIS2023-03251). We acknowledge CINECA for awarding high-performance computing resources under the ISCRA initiative, and the EuroHPC Joint Undertaking for granting access to Leonardo at CINECA, Italy.

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

APPENDIX CONTENTS

Table 7: Summary of notation.

| Notation | Explanation |
|---|---|
| $(\mathcal{M}, g)$ | Riemannian manifold $\mathcal{M}$ with Riemannian metric $g$ |
| $f_{*,P}(\cdot)$ | Differential map of $f$ at $P$ |
| $f^* g$ | Pullback metric by $f$ from $g$ |
| $\mathrm{Log}_P(\cdot)$ | Riemannian logarithm at $P$ |
| $\mathrm{Exp}_P(\cdot)$ | Riemannian exponentiation at $P$ |
| $\mathrm{PT}_{P \to Q}(\cdot)$ | Parallel transportation along the geodesic connecting $P$ and $Q$ |
| $\gamma_{(P,V)}(t)$ | Geodesic starting at $P$ with initial velocity $V \in T_P\mathcal{M}$ |
| $\mathrm{d}(\cdot, \cdot)$ | Geodesic distance |
| $\mathrm{WFM}(\cdot, \cdot)$ | Weighted Fréchet mean |
| $\mathcal{S}_{++}^n$ | SPD manifold of $n \times n$ SPD matrices |
| $\mathcal{S}^n$ | Euclidean space of $n \times n$ symmetric matrices |
| $\mathcal{L}^n$ | Euclidean space of $n \times n$ lower triangular matrices |
| $\mathcal{SL}^n$ | Euclidean space of $n \times n$ strictly lower triangular matrices |
| $\mathcal{L}_{++}^n$ | Cholesky manifold of $n \times n$ Cholesky matrices |
| $\mathbb{D}_{++}^n$ | Set of $n \times n$ diagonal matrices with positive diagonal |
| $\mathbb{R}$ | Real number field |
| $\mathbb{R}_{++}$ | Space of positive scalars |
| $\mathbb{R}^n$ | $n$-dimensional real vector space |
| $\mathbb{R}_{++}^n$ | Space of $n$-dimensional positive vectors |
| $\oplus$ | Gyroaddition |
| $\odot$ | Scalar gyromultiplication |
| $T_P\mathcal{S}_{++}^n$ | Tangent space at $P \in \mathcal{S}_{++}^n$ |
| $T_L\mathcal{L}_{++}^n$ | Tangent space at $L \in \mathcal{L}_{++}^n$ |
| $T_{\mathbb{P}}\mathbb{D}_{++}^n$ | Tangent space at $\mathbb{P} \in \mathbb{D}_{++}^n$ |
| $\mathrm{Pow}_\theta(\cdot)$ or $(\cdot)^\theta$ | Matrix power |
| $\mathrm{mlog}(\cdot)$ | Matrix logarithm |
| $\mathrm{Chol}(\cdot)$ | Cholesky decomposition |
| $\mathcal{L}_{P,M}(\cdot)$ | Generalized Lyapunov operator |
| $\pi(\cdot)$ | Isometry pulling BWM back to GBWM |
| $\lfloor \cdot \rfloor$ | Strictly lower triangular part of a square matrix |
| $\mathrm{DPow}_\theta(\cdot)$ | Diagonal power function |
| $\log(\cdot)$ | Diagonal logarithm |
| $\exp(\cdot)$ | Diagonal exponentiation |
| $g^{\mathrm{E}}$ or $\langle \cdot, \cdot \rangle$ | Standard Frobenius inner product |
| $g^{\mathrm{LE}}, g^{\mathrm{AI}}, g^{\theta\text{-E}}, g^{\mathrm{LC}}, g^{\mathrm{BW}}, g^{M\text{-BW}}$ | LEM, AIM, PEM, LCM, BWM, GBWM |
| $g^{\mathrm{DL}}, g^{\theta\text{-DE}}, g^{(\theta,\mathbb{M})\text{-DBW}}$ | Diagonal log metric, $\theta$-DPM, $(\theta, \mathbb{M})$-DBWM |
| $g^{\theta\text{-CDE}}, g^{(\theta,\mathbb{M})\text{-CDBW}}$ | $\theta$-PCM, $(\theta, \mathbb{M})$-BWCM |

## LIST OF ACRONYMS

| | |
|---|---|
| FC | Fully Connected 1 |
| MLR | Multinomial Logistics Regression 1 |
| | |
| AIM | Affine-Invariant Metric 1 |
| BWCM | Bures–Wasserstein–Cholesky Metric 2 |
| BWM | Bures–Wasserstein Metric 1 |
| GBWM | Generalized Bures–Wasserstein Metric 1 |
| LCM | Log-Cholesky Metric 1 |
| LEM | Log-Euclidean Metric 1 |
| PCM | Power-Cholesky Metric 2 |
| PEM | Power-Euclidean Metric 1 |
| SPD | Symmetric Positive Definite 1 |

## A  USE OF LARGE LANGUAGE MODELS

Large Language Models (LLMs) were used primarily for language polishing and text editing. In limited cases, they also assisted in translating certain mathematical formulations into PyTorch code.

All generated outputs were carefully reviewed and, where necessary, corrected by the authors. The authors take full responsibility for the final content of this paper.

## B  SUMMARY OF NOTATION

Tab. 7 summarizes the notation used in the main paper.

## C  PRELIMINARIES

### C.1  GYROVECTOR SPACES

The gyrovector space extends the vector space in Euclidean geometry and serves as the algebraic basis for manifolds (Ungar, 2022), which have shown great success in various applications (Ganea et al., 2018; Skopek et al., 2020; Gao et al., 2023; Nguyen and Yang, 2023). We first recap gyrogroups and gyrocommutative gyrogroups (Ungar, 2022), and proceed with gyrovector spaces (Chen et al., 2025c).

**Definition C.1** (Gyrogroups (Ungar, 2022))**.** Given a nonempty set $G$ with a binary operation $\oplus : G \times G \to G$, $\{G, \oplus\}$ forms a gyrogroup if its binary operation satisfies the following axioms for any $a, b, c \in G$ :

(G1) There is at least one element $e \in G$ called a left identity such that $e \oplus a = a$.

(G2) There is an element $\ominus a \in G$ called a left inverse of $a$ such that $\ominus a \oplus a = e$.

(G3) There is an automorphism $\mathrm{gyr}[a, b] : G \to G$ for each $a, b \in G$ such that
$$a \oplus (b \oplus c) = (a \oplus b) \oplus \mathrm{gyr}[a, b]c \quad \text{(Left Gyroassociative Law)}. \tag{17}$$
The automorphism $\mathrm{gyr}[a, b]$ is called the gyroautomorphism, or the gyration of $G$ generated by $a, b$.
(G4) $\mathrm{gyr}[a, b] = \mathrm{gyr}[a \oplus b, b]$ (Left Reduction Property).

**Definition C.2** (Gyrocommutative gyrogroups (Ungar, 2022))**.** A gyrogroup $\{G, \oplus\}$ is gyrocommutative if it satisfies
$$a \oplus b = \mathrm{gyr}[a, b](b \oplus a) \quad \text{(Gyrocommutative Law)}. \tag{18}$$

**Definition C.3** (Gyrovector spaces (Chen et al., 2025c))**.** A gyrocommutative gyrogroup $\{G, \oplus\}$ equipped with a scalar multiplication $\odot : \mathbb{R} \times G \to G$ is called a gyrovector space if it satisfies the following axioms for $s, t \in \mathbb{R}$ and $a, b, c \in G$:

(V1) $1 \odot a = a$.

(V2) $(s + t) \odot a = s \odot a \oplus t \odot a$.

(V3) $(st) \odot a = s \odot (t \odot a)$.

(V4) $\mathrm{gyr}[a, b](t \odot c) = t \odot \mathrm{gyr}[a, b]c$.

(V5) $\mathrm{gyr}[s \odot a, t \odot a] = \mathrm{Id}$, where $\mathrm{Id}$ is the identity map.

*Remark* C.4. Nguyen (2022) presented a similar definition, except that (V1) is defined as $1 \odot x = x, 0 \odot x = t \odot e = e$, and $(-1) \odot x = \ominus x$. However, as discussed by Chen et al. (2025c, Rmk. 5), $0 \odot x = t \odot e = e, (-1) \odot x = \ominus x$ are redundant.

Gyrovector structures were first identified in hyperbolic geometry (Ungar, 2022) and have recently been extended to matrix manifolds, such as SPD and Grassmann manifolds (Nguyen, 2022). Given $P$ and $Q$ in the Riemannian manifold $\mathcal{M}$, the gyro operations (Nguyen and Yang, 2023, Eqs.1 and 2) are defined as
$$\text{Gyroaddition: } P \oplus Q = \mathrm{Exp}_P \left( \mathrm{PT}_{E \to P}(\mathrm{Log}_E(Q)) \right), \tag{19}$$
$$\text{Gyromultiplication: } t \odot P = \mathrm{Exp}_E(t \, \mathrm{Log}_E(P)), \tag{20}$$
where $E$ is the origin of $\mathcal{M}$, $\mathrm{Exp}$ and $\mathrm{Log}$ denotes the Riemannian exponential and logarithmic maps, and $\mathrm{PT}_{E \to P}$ is the parallel transportation along the geodesic connecting $E$ and $P$. If Eqs. (19) and (20) conforms to the axioms of the gyrovector space, $\{\mathcal{M}, \oplus, \odot\}$ forms a gyrovector space. On the SPD manifold, the gyro operations under AIM, LEM, and LCM construct gyrovector spaces (Nguyen, 2022). The gyro-structures on the Grassmannian (Nguyen, 2022) and hyperbolic (Chen et al., 2025c) are also defined by Eqs. (19) and (20).

## C.2 RIEMANNIAN ISOMETRIES

This subsection reviews some basic properties of Riemannian isometries. For a more in-depth discussion, please refer to Do Carmo and Flaherty Francis (1992); Gallier and Quaintance (2020).

Let $\{\mathcal{M}, g\}$ and $\{\mathcal{N}, \tilde{g}\}$ be two Riemannian manifolds, and $\phi : \mathcal{M} \to \mathcal{N}$ be a Riemannian isometry, *i.e.* $g = \phi^* \tilde{g}$. We denote Log, Exp, $\gamma$, PT, $\mathrm{d}(\cdot, \cdot)$, and WFM are the Riemannian logarithm, exponentiation, geodesic, parallel transportation along the geodesic, geodesic distance, and weighted Fréchet mean on $\{\mathcal{M}, g\}$, while $\widetilde{\mathrm{Log}}$, $\widetilde{\mathrm{Exp}}$, $\widetilde{\gamma}$, $\widetilde{\mathrm{PT}}$, $\widetilde{\mathrm{d}}(\cdot, \cdot)$, and $\widetilde{\mathrm{WFM}}$ are the counterparts on $\{\mathcal{N}, \tilde{g}\}$. For $P, Q \in \mathcal{M}, V, W \in T_P \mathcal{M}$, and $\{P_i \in \mathcal{M}\}_{i=1}^N$ with weights $\{w_i\}_{i=1}^N$ satisfying $w_i > 0$ for all $i$ and $\sum_{i=1}^N w_i = 1$, we have the following:

$$\gamma_{(P,V)}(t) = \phi^{-1} \left( \tilde{\gamma}_{(\phi(P), \phi_{*,P}(V))}(t) \right), \tag{21}$$

$$\mathrm{Log}_P(Q) = (\phi_{*,P})^{-1} \left( \widetilde{\mathrm{Log}}_{\phi(P)} (\phi(Q)) \right), \tag{22}$$

$$\mathrm{Exp}_P(V) = \phi^{-1} \left( \widetilde{\mathrm{Exp}}_{\phi(P)} (\phi_{*,P}(V)) \right), \tag{23}$$

$$\mathrm{PT}_{P \to Q}(V) = (\phi_{*,Q})^{-1} \left( \widetilde{\mathrm{PT}}_{\phi(P) \to \phi(Q)} (\phi_{*,P}(V)) \right), \tag{24}$$

$$\mathrm{d}(P, Q) = \widetilde{\mathrm{d}}(\phi(P), \phi(Q)), \tag{25}$$

$$\mathrm{WFM}(\{P_i\}, \{w_i\}) = \phi^{-1} \left( \widetilde{\mathrm{WFM}} (\{\phi(P_i)\}, \{w_i\}) \right), \tag{26}$$

where $\gamma_{(P,V)}(t)$ is the geodesic starting at $P$ with initial velocity $V \in T_P \mathcal{M}$, and $\phi_{*,P} : T_P \mathcal{M} \to T_{\phi(P)} \mathcal{N}$ is the differential map of $\phi$ at $P$. Eq. (26) is the direct corollary of Eq. (25).

# D  RIEMANNIAN OPERATORS UNDER $(\theta, \mathbb{M})$-DBWM

We first define a map $\phi_\theta : \mathcal{L}_{++}^n \to \mathcal{L}_{++}^n$ as

$$\phi_\theta(L) = \lfloor L \rfloor + \mathbb{L}^\theta, \forall L \in \mathcal{L}_{++}^n. \tag{27}$$

Its differential at $L \in \mathcal{L}_{++}^n$ is given as

$$\phi_{\theta *, L}(X) = \lfloor X \rfloor + \theta \mathbb{L}^{\theta-1} \mathbb{X}, \forall X \in T_L \mathcal{L}_{++}^n. \tag{28}$$

Let $g^{\mathbb{M}\text{-DBW}}$ and $g^{(\theta, \mathbb{M})\text{-DBW}}$ be $\mathbb{M}$-DBWM and $(\theta, \mathbb{M})$-DBWM, respectively. Since constant scaling of a Riemannian metric preserves the Christoffel symbols, the Riemannian operators such as Riemannian logarithm, exponentiation, and parallel transportation under $g^{(\theta, \mathbb{M})\text{-DBW}}$ is the same as the pullback metric $\phi_\theta^* g^{\mathbb{M}\text{-DBW}}$. Following Sec. C.2, these Riemannian operators under $(\theta, \mathbb{M})$-DBWM can be obtained by $\phi_\theta^* g^{\mathbb{M}\text{-DBW}}$. Besides, as constant scaling does not affect WFM, the WFM under $(\theta, \mathbb{M})$-DBWM is the same as the one under $\phi_\theta^* g^{\mathbb{M}\text{-DBW}}$. The latter can be calculated by the properties of isometries presented in Sec. C.2. Therefore, by the properties of isometry in Sec. C.2 and Eqs. (27) and (28), we can obtain all the Riemannian operators.

# E  EXPERIMENTAL DETAILS

## E.1  IMPLEMENTATION DETAILS OF SPD NEURAL NETWORKS

### E.1.1  BACKBONE NETWORKS

SPDNet (Huang et al., 2017) consists of three basic building blocks:

$$\text{BiMap: } S^k = W^k S^{k-1} W^k,$$
$$\text{ReEig: } S^k = U^{k-1} \max(\Sigma^{k-1}, \epsilon I_n) U^{k-1\top}, \tag{29}$$
$$\text{LogEig: } S^k = \mathrm{mlog}(S^{k-1}),$$

where $S^{k-1} = U^{k-1} \Sigma^{k-1} U^{k-1\top}$ is the eigendecomposition, and $W^k$ is column-wisely orthogonal. BiMap (Bilinear Mapping) and ReEig (Eigenvalue Rectification) are the counterparts of linear and

ReLu nonlinear activation functions in Euclidean networks. LogEig layer projects SPD matrices into the tangent space at the identity matrix for classification. However, LogEig might distort the innate geometry of SPD features. Recently, Nguyen and Yang (2023); Chen et al. (2024a;c) generalized the Euclidean MLR to SPD manifolds for intrinsic classification.

GyroSPD (Nguyen and Yang, 2023) substitutes the BiMap layer in SPDNet with gyrotranslation: $S^k = W^k \oplus S^{k-1}$, where $W^k$ is an SPD matrix parameter. Following Nguyen and Yang (2023), we use the AIM-based gyrotranslation:

$$P \oplus Q = P^{\frac{1}{2}} Q P^{\frac{1}{2}}, \quad \forall P, Q \in \mathcal{S}_{++}^n. \tag{30}$$

RResNet (Katsman et al., 2024) on the SPD manifold follows the same architecture as SPDNet, except that it inserts an SPD residual block before the LogEig layer.

### E.1.2 DATASETS AND PREPROCESSING

The Radar[1] (Brooks et al., 2019) dataset consists of 3,000 synthetic radar signals. Following the protocol in Brooks et al. (2019), each signal is split into windows of length 20, resulting in 3,000 SPD covariance matrices of $20 \times 20$ equally distributed in 3 classes.

The HDM05[2] (Müller et al., 2007) dataset contains 2,273 skeleton-based motion capture sequences executed by various actors. Each frame consists of 3D coordinates of 31 joints of the subjects, and each sequence can be modeled by a $93 \times 93$ covariance matrix. Following Brooks et al. (2019), we trim the dataset down to 2086 sequences scattered throughout 117 classes by removing some under-represented classes.

The FPHA[3] (Garcia-Hernando et al., 2018) includes 1,175 skeleton-based first-person hand gesture videos of 45 different categories with 600 clips for training and 575 for testing. Each frame contains the 3D coordinates of 21 hand joints.

### E.1.3 DETAILS OF THE EXPERIMENTS ON SPD MLR

We follow the official PyTorch code[4] of SPD MLR (Chen et al., 2024a) for implementation. Due to the lack of an official code, the GyroSPD backbone is carefully reimplemented in PyTorch following the original paper (Nguyen and Yang, 2023). For simplicity, we set $\mathbb{M}$ in $(\theta, \mathbb{M})$-BWCM as the identity matrix.

**SPDNet** (Huang and Van Gool, 2017). We replace the vanilla tangent classifier (LogEig + FC + Softmax) in SPDNet with SPD MLRs induced by AIM, LEM, LCM, $\theta$-PCM, and $\theta$-BWCM. We use a Riemannian AMSGrad (Bécigneul and Ganea, 2019) with a learning rate of $1e^{-2}$, a batch size of 30, and a maximum of 200 epochs. The architectures are denoted as $[d_0, d_1, \ldots, d_L]$, where $d_i$ is the output dimension of the $i$-th BiMap layer. Following prior work (Huang and Van Gool, 2017; Brooks et al., 2019; Chen et al., 2024b), we adopt $[20, 16, 8]$ on Radar, $[63, 33]$ on FPHA, and $[93, 30]$, $[93, 70, 30]$, $[93, 70, 50, 30]$ for 1-, 2-, and 3-block variants on HDM05. As suggested by Chen et al. (2024b), matrix power improves the performance of Cholesky-based metric on FPHA. We therefore apply a power of $-0.25$ before SPD MLR layers under LCM, $\theta$-PCM, and $\theta$-BWCM. For $\theta$-PCM on FPHA, we further adopt a weight decay of $1e^{-4}$. The deformation factor $\theta$ is reported in Tab. 8.

**GyroSPD** (Nguyen and Yang, 2023). Following Nguyen and Yang (2023), the backbone consists of one gyrotranslation layer followed by an SPD MLR. We compare LEM-, LCM-, AIM-, and our metrics under the same settings: Riemannian AMSGrad with a learning rate of $1e^{-2}$ and a batch size of 30. The training epochs are up to 100, 100, and 50, respectively. Similarly, a matrix power of 0.25 is applied on FPHA for LCM and our Cholesky-based metrics. The deformation factor $\theta$ is reported in Tab. 8.

**SPD input of SPDNet.** Following Chen et al. (2024c, App. G. 1.2), we model each sequence into a global covariance of $20 \times 20$, $93 \times 93$, and $63 \times 63$.

---

[1] https://www.dropbox.com/s/dfnlx2bnyh3kjwy/data.zip?dl=0
[2] https://resources.mpi-inf.mpg.de/HDM05/
[3] https://github.com/guiggh/hand_pose_action
[4] https://github.com/GitZH-Chen/SPDMLR

Table 8: Hyperparameter $\theta$ in $\theta$-PCM and $\theta$-BWCM. It is selected from Tabs. 10 and 11.

| Backbone | Metric | Radar | HDM05 | FPHA |
|---|---|---|---|---|
| SPDNet | $\theta$-PCM | -1.5 | -0.5 | 0.75 |
| | $\theta$-BWCM | -0.75 | -1.5 | 1 |
| GyroSPD | $\theta$-PCM | -0.75 | -0.75 | -0.75 |
| | $\theta$-BWCM | -1.5 | -1.5 | -0.5 |

**SPD input of GyroSPD.** For Radar, the input is the same as SPDNet. For HDM05 and FPHA, we follow Nguyen and Yang (2023) to model each sample into a multi-channel covariance tensor $[c, n, n]$. Specifically, we first identify the closest left (right) neighbor of every joint based on their distance to the hip (wrist) joint, and then combine the 3D coordinates of each joint and those of its left (right) neighbor to create a feature vector for the joint. For a given frame $t$, we compute its Gaussian embedding (Lovrić et al., 2000):

$$Y_t = (\det \Sigma_t)^{-\frac{1}{n+1}} \begin{bmatrix} \Sigma_t + \mu_t (\mu_t)^T & \mu_t \\ (\mu_t)^T & 1 \end{bmatrix}, \tag{31}$$

where $\mu_t$ and $\Sigma_t$ are the mean vector and covariance matrix computed from the set of feature vectors within the frame. The lower part of the matrix $\log(Y_t)$ is flattened to obtain a vector $\tilde{v}_t$. All vectors $\tilde{v}_t$ within a time window $[t, t+c-1]$, where $c$ is determined from a temporal pyramid representation of the sequence (the number of temporal pyramids is set to 2 in our experiments), are used to compute a covariance matrix as

$$\widetilde{\Sigma}_t = \frac{1}{c} \sum_{i=t}^{t+c-1} (\tilde{v}_i - \overline{v}_t)(\tilde{v}_i - \overline{v}_t)^T, \tag{32}$$

where $\overline{v}_t = \frac{1}{c} \sum_{i=t}^{t+c-1} \tilde{v}_i$. The resulting $\{\widetilde{\Sigma}_t\}$ are the covariance matrices that we need. On FPHA, we generate the covariance based on three sets of neighbors: left, right, and vertical (bottom) neighbors. After preprocessing, the input correlation matrices are $[3, 28, 28]$, and $[9, 28, 28]$ on HDM05 and FPHA, respectively.

### E.1.4 Details of the experiments on RResNet

**Implementation.** The backbone architectures of RResNet (Katsman et al., 2024) are similar to those of SPDNet: $[20, 16, 8]$ on Radar, $[93, 30]$ on HDM05, and $[63, 33]$ on FPHA, respectively. The only architectural difference lies in the head: SPDNet directly uses an classification layer, whereas RResNet attaches a residual block before the classification head. Following RResNet, we adopt $\mathrm{Log}_I$ + FC + Softmax for classification under each metric. We use the official code[5] to re-implement the AIM- and LEM-based RResNet. For the RResNets based on LCM and our $\theta$-PCM, we adopt the following settings: a learning rate of $1e^{-2}$, batch size of 30, and a maximum of 200 epochs. We use the standard cross-entropy loss as the training objective and optimize the parameters with the Riemannian AMSGrad optimizer (Bécigneul and Ganea, 2019). The Cholesky diagonal power is set to $-1$, $0.5$, and $-0.5$ on the Radar, HDM05, and FPHA datasets, respectively. Additionally, on the FPHA dataset, a matrix power of $-0.25$ is applied before the residual blocks to activate the latent geometry for both LCM and our metric.

**SPD input.** The input SPD matrices are the same as those used in SPDNet.

### E.2 Tensor interpolation

As shown by Arsigny et al. (2005; 2007), geodesic interpolation of SPD matrices is important in diffusion tensor imaging. This subsection illustrates the geodesic interpolation under different SPD metrics.

---

[5] https://github.com/CUAI/Riemannian-Residual-Neural-Networks

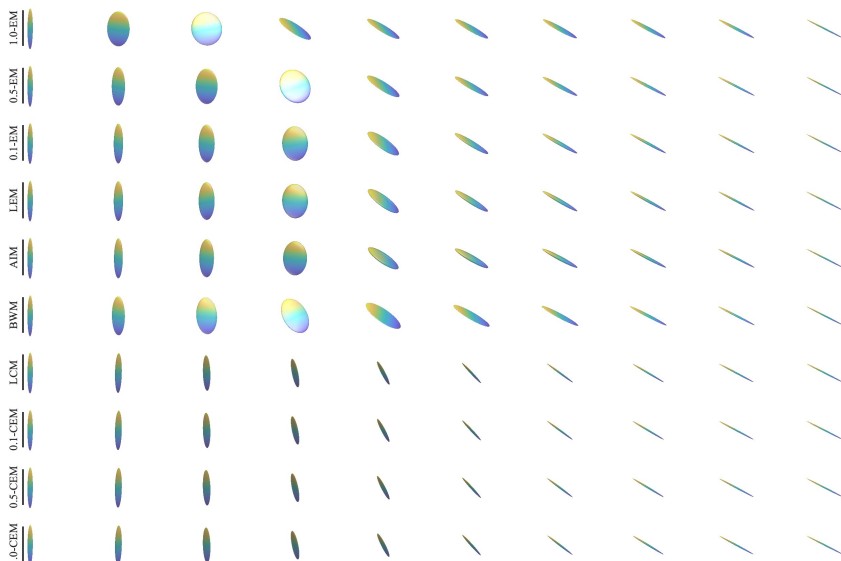

Figure 2: Geodesic interpolation of SPD matrices under different Riemannian metrics. Each $3 \times 3$ SPD matrix can be visualized as an ellipse (Arsigny et al., 2005). The two endpoints are fixed across all metrics.

Table 9: Swelling effects of geodesic SPD interpolations. Deeper greens indicate greater swelling.

| Metric | The determinant of the $i$-th interpolation | | | | | | | | | |
|---|---|---|---|---|---|---|---|---|---|---|
| | 0 | 1 | 2 | 3 | 4 | 5 | 6 | 7 | 8 | 9 |
| 1.0-EM | 3.07 | 104.86 | 182.09 | 234.38 | 261.35 | 262.64 | 237.86 | 186.64 | 108.61 | 3.38 |
| 0.5-EM | 3.07 | 18.67 | 39.93 | 59.53 | 71.96 | 73.79 | 64.14 | 45.01 | 21.73 | 3.38 |
| 0.1-EM | 3.07 | 4.25 | 5.42 | 6.38 | 6.96 | 7.05 | 6.62 | 5.75 | 4.6 | 3.38 |
| LEM | 3.07 | 3.1 | 3.14 | 3.17 | 3.2 | 3.24 | 3.27 | 3.31 | 3.34 | 3.38 |
| AIM | 3.07 | 3.1 | 3.14 | 3.17 | 3.2 | 3.24 | 3.27 | 3.31 | 3.34 | 3.38 |
| BWM | 3.07 | 15.32 | 32.04 | 48.14 | 59.22 | 62.09 | 55.33 | 39.93 | 19.98 | 3.38 |
| LCM | 3.07 | 3.1 | 3.14 | 3.17 | 3.2 | 3.24 | 3.27 | 3.31 | 3.34 | 3.38 |
| 0.1-PCM | 3.07 | 3.15 | 3.23 | 3.29 | 3.34 | 3.37 | 3.39 | 3.4 | 3.4 | 3.38 |
| 0.5-PCM | 3.07 | 3.35 | 3.59 | 3.79 | 3.91 | 3.97 | 3.94 | 3.83 | 3.64 | 3.38 |
| 1.0-PCM | 3.07 | 3.6 | 4.07 | 4.46 | 4.72 | 4.83 | 4.76 | 4.49 | 4.03 | 3.38 |

Suppose $P = LL^\top$ and $Q = KK^\top$ as the Cholesky decomposition of $P, Q \in \mathcal{S}_{++}^n$, the geodesics that connect $P$ and $Q$ under $\theta$-PCM and $(\theta, \mathbb{M})$-BWCM are

$$\theta\text{-PCM: } \text{Chol}^{-1}\left[\lfloor L \rfloor + t(\lfloor K \rfloor - \lfloor L \rfloor) + \left(\mathbb{L}^\theta + t(\mathbb{K}^\theta - \mathbb{L}^\theta)\right)^{\frac{1}{\theta}}\right], \tag{33}$$

$$(\theta, \mathbb{M})\text{-BWCM: } \text{Chol}^{-1}\left[\lfloor L \rfloor + t(\lfloor K \rfloor - \lfloor L \rfloor) + \left(\mathbb{L}^{\frac{\theta}{2}} + t(\mathbb{K}^{\frac{\theta}{2}} - \mathbb{L}^{\frac{\theta}{2}})\right)^{\frac{\theta}{2}}\right]. \tag{34}$$

As the geodesics under $\theta$-PCM and $(\theta, \mathbb{M})$-BWCM have a similar expression, we focus on $\theta$-PCM. Fig. 2 visualizes the geodesic interpolations on $\mathcal{S}_{++}^3$ under different metrics, including $\theta$-EM, LEM, AIM, BWM, LCM, and $\theta$-PCM. Tab. 9 presents the associated determinant of each interpolated SPD matrix. We can make the following observation.

1. The standard Euclidean metric (1-EM) exhibits a significant swelling effect, where the maximal determinant of interpretation is extremely larger than the determinants of the starting and end points. Although matrix power can mitigate the swelling effect, $\theta$-EM still suffers from swelling.
2. We find that BWM also demonstrates a clear swelling effect. In contrast, LCM, AIM, and LEM show no swelling effect.

Table 10: Results of SPD MLR on the SPDNet backbone under $\theta$-PCM and $\theta$-BWCM with different deformation factors $\theta$.

| Dataset | Metric | -2 | -1.5 | -1 | -0.75 | -0.5 | -0.25 | 0.25 | 0.5 | 0.75 | 1 | 1.5 |
|---------|--------|----|------|----|-------|------|-------|------|-----|------|---|-----|
| Radar | $\theta$-PCM | 95.71 ± 0.57 | **95.79 ± 0.38** | 94.64 ± 0.34 | 95.36 ± 1.32 | 94.11 ± 1.05 | 94.51 ± 0.75 | 94.35 ± 0.71 | 93.55 ± 0.59 | 94.16 ± 0.96 | 93.87 ± 0.40 | 92.75 ± 0.68 |
| | $\theta$-BWCM | 91.89 ± 0.31 | 92.91 ± 1.05 | 92.16 ± 1.10 | **93.93 ± 0.79** | 92.13 ± 0.60 | 92.16 ± 1.30 | 92.40 ± 0.95 | 92.08 ± 0.85 | 92.48 ± 0.93 | 92.77 ± 0.66 | 91.81 ± 0.52 |
| HDM05 | $\theta$-PCM | 46.63 ± 1.63 | 49.86 ± 1.41 | 58.81 ± 1.41 | 65.18 ± 2.89 | **65.75 ± 2.86** | 65.65 ± 2.03 | 64.00 ± 3.20 | 63.73 ± 2.90 | 64.59 ± 3.22 | 64.58 ± 3.14 | 64.62 ± 1.44 |
| | $\theta$-BWCM | 65.20 ± 2.63 | **67.40 ± 0.90** | 65.42 ± 1.88 | 65.29 ± 2.32 | 64.33 ± 2.71 | 64.07 ± 2.65 | 63.89 ± 2.50 | 64.67 ± 1.38 | 65.15 ± 0.75 | 63.33 ± 2.18 | 65.00 ± 1.48 |
| FPHA | $\theta$-PCM | 88.20 ± 0.29 | 88.20 ± 0.32 | 88.30 ± 0.24 | 88.33 ± 0.21 | 88.47 ± 0.29 | 88.70 ± 0.39 | 89.03 ± 0.22 | 88.97 ± 0.16 | **89.40 ± 0.13** | 89.03 ± 0.16 | 89.23 ± 0.33 |
| | $\theta$-BWCM | 85.87 ± 0.66 | 85.87 ± 0.66 | 85.87 ± 0.66 | 85.93 ± 0.67 | 85.93 ± 0.67 | 85.93 ± 0.67 | 85.93 ± 0.67 | 86.00 ± 0.70 | 86.03 ± 0.75 | **86.27 ± 0.60** | 86.13 ± 0.73 |

Table 11: Results of SPD MLR on the GyroSPD backbone under $\theta$-PCM and $\theta$-BWCM with different deformation factors $\theta$.

| Dataset | Metric | -2 | -1.5 | -1 | -0.75 | -0.5 | -0.25 | 0.25 | 0.5 | 0.75 | 1 | 1.5 |
|---------|--------|----|------|----|-------|------|-------|------|-----|------|---|-----|
| Radar | $\theta$-PCM | 96.88 ± 0.36 | 96.75 ± 0.26 | 96.27 ± 0.36 | **97.04 ± 0.64** | 96.45 ± 0.88 | 97.01 ± 0.66 | 96.77 ± 0.64 | 96.51 ± 0.65 | 96.56 ± 0.86 | 96.53 ± 0.79 | 96.32 ± 0.68 |
| | $\theta$-BWCM | 96.16 ± 0.90 | **96.21 ± 0.25** | 95.25 ± 1.19 | 96.16 ± 0.57 | 96.08 ± 0.48 | 95.84 ± 0.69 | 94.96 ± 0.94 | 95.60 ± 0.65 | 95.04 ± 0.33 | 94.77 ± 0.96 | 95.55 ± 0.91 |
| HDM05 | $\theta$-PCM | 50.18 ± 0.99 | 32.49 ± 25.81 | 66.43 ± 1.22 | **71.93 ± 1.21** | 70.94 ± 1.17 | 69.64 ± 0.68 | 67.68 ± 0.92 | 68.24 ± 0.28 | 68.02 ± 0.40 | 67.80 ± 1.07 | 67.86 ± 1.69 |
| | $\theta$-BWCM | 72.64 ± 1.15 | **72.74 ± 0.43** | 69.58 ± 1.34 | 69.24 ± 1.14 | 68.84 ± 0.74 | 68.10 ± 1.02 | 67.72 ± 1.19 | 68.49 ± 0.64 | 68.08 ± 0.93 | 67.72 ± 0.58 | 67.45 ± 0.45 |
| FPHA | $\theta$-PCM | 90.60 ± 0.58 | 91.17 ± 0.32 | 91.10 ± 0.29 | **91.17 ± 0.30** | 91.00 ± 0.30 | 90.87 ± 0.12 | 90.97 ± 0.16 | 90.97 ± 0.12 | 90.97 ± 0.07 | 90.90 ± 0.08 | 90.97 ± 0.22 |
| | $\theta$-BWCM | 90.87 ± 0.16 | 90.80 ± 0.07 | 90.83 ± 0.11 | 90.97 ± 0.07 | **91.00 ± 0.11** | **91.00 ± 0.11** | 90.97 ± 0.12 | 90.97 ± 0.12 | 90.93 ± 0.08 | 90.90 ± 0.08 | 90.87 ± 0.12 |

3. The trivial PCM ($\theta = 1$) considerably mitigates the swelling effect compared to the Euclidean metric, but it still exhibits some level of swelling. However, by introducing Cholesky power deformation, $\theta$-PCM effectively reduces the swelling effect. Notably, the swelling effect of $\theta$-PCM is significantly weaker than that of $\theta$-EM under the same $\theta$.

4. Our PCM shows a visually similar interpolation as LCM. This suggests that our metric retains some practical potential of LCM but with better numerical stability (as demonstrated in Sec. 6.2).

### E.3 ABLATIONS ON CHOLESKY POWER DEFORMATION

To better understand the Cholesky power deformation factor $\theta$, we sweep $\theta$ from $-2$ to $1.5$ for both $\theta$-PCM and $\theta$-BWCM. We focus on the MLR experiments. For SPDNet, we reuse the architectures described in Sec. E.1.3 and adopt the 3-block variant on HDM05. For GyroSPD, we follow the backbone in Sec. E.1.3 with one gyrotranslation layer and an SPD MLR. All other training hyperparameters are kept fixed.

Tabs. 10 and 11 summarize the 5-fold results on SPDNet and GyroSPD backbones, respectively. We make the following observations.

- **Effectiveness.** On both backbones, most tested values of $\theta$ already match or surpass the AIM, LEM, and LCM baselines, and suitably chosen $\theta$ consistently improves accuracy over the default $\theta = 1$, confirming the effectiveness of our deformation factor. Besides, the magnitude of this improvement is dataset dependent. The gains are modest on Radar and FPHA, whereas HDM05 benefits much more from tuning $\theta$. Sec. E.4 relates this behaviour to the statistics of Cholesky diagonal entries of the input SPD matrices on each dataset.
- **Relative stability of BWCM.** On HDM05, when $\theta$ takes small negative values (for example, $\theta = -1.5$ or $\theta = -2$), the accuracy of PCM deteriorates, whereas BWCM remains comparatively stable. This behaviour is consistent with Thm. 5.1. Since the SPD MLRs under PCM and BWCM differ mainly in the diagonal powers $(\cdot)^{\theta}$ versus $(\cdot)^{\theta/2}$, BWCM experiences a milder deformation for the same $\theta$.

### E.4 CHOLESKY POWER DEFORMATION AND DIAGONAL STATISTICS

According to Thm. 5.1, the deformation factor $\theta$ influences the SPD MLR only through the diagonal powers of the Cholesky factors. Therefore, the effect of $\theta$ is inherently tied to the distribution of the Cholesky diagonal entries. To further interpret the dataset-dependent sensitivity to $\theta$ and, in particular, the preference for negative values on HDM05, we analyse the diagonal distribution in the Cholesky domain. For each dataset, we compute the Cholesky factors of the input and output SPD matrices in SPDNet and collect their diagonal entries. For HDM05, we focus on the 3-block architecture. Fig. 3 reports the resulting distributions, from which we make the following observations.

- **Balanced diagonals on Radar and FPHA.** On Radar and FPHA, both the input and output Cholesky diagonals are relatively balanced, with most entries lying in a moderate range. In

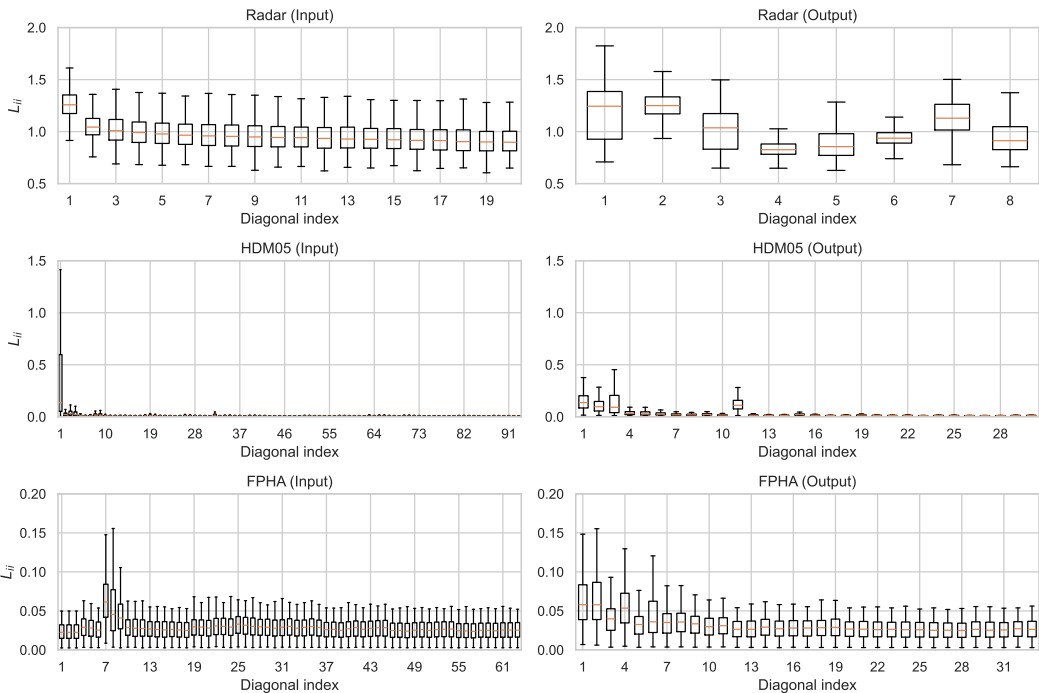

Figure 3: Distributions of the Cholesky diagonal entries of the SPDNet input and output covariances on Radar, HDM05, and FPHA.

this case, applying a power deformation with different $\theta$ mainly rescales the diagonals without drastically changing their relative order. This explains the results in Tabs. 10 and 11, where the performance is relatively stable across a wide range of $\theta$ on these two datasets.

- **Imbalanced diagonals on HDM05.** In contrast, HDM05 exhibits a highly skewed distribution for both input and output covariances, where most Cholesky diagonal entries are very small and concentrated close to zero, while only a few entries are much larger. Therefore, varying $\theta$ induces a much more pronounced reweighting. Moreover, a negative $\theta$ effectively acts as an activation: taking a negative power turns these small diagonal entries into larger values while relatively shrinking those larger than 1. As a result, many previously under-represented directions in the Cholesky spectrum are emphasised, which accounts for the noticeably larger gains of negative $\theta$ in Tabs. 10 and 11.

### E.5 SCALABILITY

In this subsection, we investigate the scalability of our PCM and BWCM, in comparison with five existing SPD metrics, namely AIM, LEM, LCM, PEM, and BWM. We use SPD MLR as a representative application. We first analyze the asymptotical complexity of each SPD MLR. We then complement this analysis with synthetic experiments that measure the actual runtime of a single SPD MLR training step across different matrix dimensions.

#### E.5.1 ASYMPTOTIC COMPLEXITY

As shown by Chen et al. (2024c, Thm. 4.2) and our Thm. 5.1, the $C$-class SPD MLRs under different metrics for the input $S \in \mathcal{S}_{++}^n$ are

$$\text{AIM:} \quad p(y = k \mid S) \propto \exp\left[\left\langle \log\left(P_k^{-\frac{1}{2}} S P_k^{-\frac{1}{2}}\right), A_k\right\rangle\right], \tag{35}$$

$$\text{LEM:} \quad p(y = k \mid S) \propto \exp\left[\langle \log(S) - \log(P_k), A_k\rangle\right], \tag{36}$$

$$\text{LCM:} \quad p(y = k \mid S) \propto \exp\left[\left\langle \lfloor K \rfloor - \lfloor L_k \rfloor + \log(\mathbb{K}) - \log(\mathbb{L}_k), \lfloor A_k \rfloor + \frac{1}{2}\mathbb{A}_k\right\rangle\right], \tag{37}$$

Table 12: Number of matrix functions required per sample for a $C$-class SPD MLR. Spectral matrix functions include matrix logarithm, matrix power, and the Lyapunov operator.

| Metric | Num. spectral matrix functions | Num. Cholesky decomposition |
|--------|--------------------------------|------------------------------|
| AIM | $1 + 2C$ | 0 |
| LEM | $1 + C$ | 0 |
| LCM | 0 | $1 + C$ |
| PEM | $1 + C$ | 0 |
| BWM | $1 + 3C$ | $C$ |
| PCM | 0 | $1 + C$ |
| BWCM | 0 | $1 + C$ |

Table 13: Asymptotic per-sample complexity of a $C$-class SPD MLR for an $n \times n$ input SPD matrix.

| Metric | Asymptotic complexity |
|--------|------------------------|
| AIM | $O\big(9(1 + 2C)n^3\big)$ |
| LEM | $O\big(9(1 + C)n^3\big)$ |
| LCM | $O\big(\frac{1+C}{3}n^3\big)$ |
| PEM | $O\big(9(1 + C)n^3\big)$ |
| BWM | $O\big((9(1 + 3C) + \frac{C}{3})n^3\big)$ |
| PCM | $O\big(\frac{1+C}{3}n^3\big)$ |
| BWCM | $O\big(\frac{1+C}{3}n^3\big)$ |

$$\text{PEM:} \quad p(y = k \mid S) \propto \exp\left[\frac{1}{\theta}\left\langle S^\theta - P_k^\theta, A_k\right\rangle\right], \tag{38}$$

$$\text{BWM:} \quad p(y = k \mid S) \propto \exp\left[\frac{1}{2}\left\langle (P_k S)^{\frac{1}{2}} + (SP_k)^{\frac{1}{2}} - 2P_k, \mathcal{L}_{P_k}(L_k A_k L_k^\top)\right\rangle\right], \tag{39}$$

$$\theta\text{-PCM} : p(y = k \mid S) \propto \exp\left[\langle \lfloor K \rfloor - \lfloor L_k \rfloor, \lfloor A_k \rfloor\rangle + \frac{1}{2\theta}\left\langle \mathbb{K}^\theta - \mathbb{L}_k^\theta, \mathbb{A}_k\right\rangle\right], \tag{40}$$

$$(\theta, \mathbb{M})\text{-BWCM} : p(y = k \mid S) \propto \exp\left[\langle \lfloor K \rfloor - \lfloor L_k \rfloor, \lfloor A_k \rfloor\rangle + \frac{1}{4\theta}\left\langle \mathbb{K}^{\frac{\theta}{2}} - \mathbb{L}_k^{\frac{\theta}{2}}, \mathbb{M}^{-1}\mathbb{A}_k\right\rangle\right], \tag{41}$$

where $P_k \in \mathcal{S}_{++}^n$ and $A_k \in \mathcal{S}^n$ are MLR weights, $\log(\cdot)$ is the matrix logarithm, and $\mathcal{L}_P(V)$ is the solution to the matrix linear system $\mathcal{L}_P[V]P + P\mathcal{L}_P[V] = V$, known as the Lyapunov operator.

**Analysis.** Tab. 12 summarizes the number of spectral and Cholesky matrix functions required by each SPD MLR. Cholesky decomposition requires $O(1/3n^3)$ flops, while eigendecomposition costs $O(9n^3)$ flops (Golub and Van Loan, 2013, Algs. 4.2.3 and 8.3.3). Combining these counts, Tab. 13 reports the resulting asymptotic per-sample complexity for each metric. Cholesky-based metrics (LCM, PCM, BWCM) are asymptotically more efficient than the eigen-based metrics (LEM, PEM, AIM, and BWM), with AIM and BWM being the slowest among the considered methods. In addition, PCM and BWCM can be practically more efficient than LCM, since diagonal powers are cheaper to compute than diagonal logarithms.

### E.5.2 EMPIRICAL VALIDATION

**Setup.** To validate the asymptotic complexity in Tab. 13, we measure the average wall-clock time of a single forward–backward training step of an SPD MLR classifier as the matrix dimension increases. The model consists of a single SPD MLR layer with 50 output classes followed by a cross-entropy loss. For each dimension $n \in \{32, 64, 128, 256, 512\}$, we randomly generate a batch of 30 $n \times n$ SPD matrices. In each run, we perform one forward and one backward pass and record the total runtime of this step. For PEM, we set the matrix power to $0.5$.

**Results.** As reported in Tab. 14, our PCM and BWCM are the fastest metrics across all tested dimensions, and the gap becomes particularly pronounced in the high-dimensional case. For small and medium scales (32 and 64), LCM, PCM, and BWCM have very similar runtimes and all are

Table 14: Average runtime (in seconds) of one SPD MLR training step under different dimensions.

| Dim | AIM | LEM | LCM | PEM | BWM | PCM | BWCM |
|---|---|---|---|---|---|---|---|
| 32 | 0.2380 | 0.0077 | 0.0046 | 0.0076 | 0.2377 | 0.0040 | 0.0040 |
| 64 | 1.0139 | 0.0395 | 0.0303 | 0.0473 | 1.1205 | 0.0251 | 0.0225 |
| 128 | 3.6256 | 0.1832 | 0.1490 | 0.1844 | 4.0674 | 0.1013 | 0.1019 |
| 256 | 14.5142 | 0.7793 | 0.5833 | 0.7853 | 16.5918 | 0.3848 | 0.4077 |
| 512 | 60.1918 | 3.2948 | 2.5030 | 3.4357 | 70.8647 | 1.7553 | 1.7526 |

Table 15: Ablation on the diagonal matrix $\mathbb{M}$ in BWCM. We compare BWCM with a variant that learns the diagonal of $\mathbb{M}$ ($\mathbb{M}$-BWCM) on SPDNet and GyroSPD.

| Backbone | | SPDNet | | | GyroSPD | |
|---|---|---|---|---|---|---|
| Dataset | Radar | HDM05 | FPHA | Radar | HDM05 | FPHA |
| $\mathbb{M}$-BWCM | **95.15 ± 0.61** | 66.71 ± 0.96 | 86.27 ± 0.63 | **96.93 ± 0.40** | 72.62 ± 0.43 | 91.00 ± 0.11 |
| BWCM | 93.93 ± 0.79 | **67.40 ± 0.90** | 86.27 ± 0.60 | 96.21 ± 0.25 | **72.74 ± 0.43** | 91.00 ± 0.11 |

clearly faster than AIM, LEM, PEM, and BWM. When the dimension increases to 512, AIM and BWM require about 60 and 70 seconds per training step, whereas PCM and BWCM remain within roughly 1.7 seconds. In this setting, PCM and BWCM are even faster than LCM. This behaviour can be explained from three perspectives.

### E.6 EFFECT OF $\mathbb{M}$ IN BWCM

**Setup.** To examine the effect of the diagonal matrix $\mathbb{M}$ in our $(\theta, \mathbb{M})$-BWCM, we compare BWCM with $\mathbb{M} = I$ against a variant that learns $\mathbb{M} \in \mathbb{D}_{++}^n$ ($\mathbb{M}$-BWCM). We follow the SPD MLR configurations on both SPDNet and GyroSPD backbones. The value of $\theta$ is the same for both BWCM and $\mathbb{M}$-BWCM and is set as in Tab. 8. In $\mathbb{M}$-BWCM, we optimize an unconstrained diagonal vector $v$ and set $\mathbb{M} = \mathrm{diag}(\exp(v))$ so that the diagonal of $\mathbb{M}$ remains positive. We initialize $v$ to zero, corresponding to $\mathbb{M} = I$ at the beginning of training. The size of $v$ matches the input dimension of the SPD MLR, which is 8, 30, and 33 on SPDNet and 20, $3 \times 28$, and $9 \times 28$ on GyroSPD for Radar, HDM05, and FPHA, where 3 and 9 are channel dimensions.

**Results.** Tab. 15 reports the five-fold results. On both backbones, $\mathbb{M}$-BWCM slightly improves accuracy on Radar, is marginally worse on HDM05, and matches BWCM on FPHA. Since the additional parameters introduced by $\mathbb{M}$ are relatively small, its impact on the overall model is marginal. These observations suggest that we can simply fix $\mathbb{M} = I$ in BWCM in practice, which is the setting in our main paper.

### E.7 HARDWARE

The experiments on the GyroSPD backbone are executed on a single NVIDIA A6000 GPU. Other experiments require SVD and Cholesky decompositions on relatively large SPD matrices, which are more efficiently performed on CPUs. Therefore, these experiments are conducted on an Intel Core i9-7960X CPU with 32GB RAM.

## F PROOFS

### F.1 PROOF OF THM. 3.2

*Proof.* As $\theta$-DPM is the product metric of $\{\mathcal{SL}^n, g^{\mathrm{E}}\}$ and $n$ copies of $\{\mathbb{R}_{++}, g^{\theta\text{-E}}\}$. We first show the Riemannian operators on $\{\mathbb{R}_{++}, g^{\theta\text{-E}}\}$, and then we can readily obtain the Riemannian operators on $\{\mathcal{L}_{++}^n, g^{\theta\text{-DE}}\}$ by the principles of product metrics.

As shown by Thanwerdas and Pennec (2022), $g^{\theta\text{-E}}$ is the pullback metric of $g^{\mathrm{E}}$ by power function $\mathrm{Pow}_\theta(\cdot)$ and scaled by $\frac{1}{\theta^2}$, expressed as $g^{\theta\text{-E}} = \frac{1}{\theta^2} \mathrm{Pow}_\theta^* g^{\mathrm{E}}$. Besides, as constant scaling does

not change the Christoffel symbols, the geodesic, Riemannian logarithm & exponentiation, and parallel transportation along a geodesic remain the same under $g^{\theta\text{-E}}$ and $\mathrm{Pow}_\theta^* g^{\mathrm{E}}$. These Riemannian operators under $\mathrm{Pow}_\theta^* g^{\mathrm{E}}$ can be obtained by the properties of Riemannian isometries (Sec. C.2). Specifically, given $p, q \in \mathbb{R}_{++}$ and $w, v \in T_p \mathbb{R}_{++}$, we have the following:

$$\mathrm{Pow}_{\theta*,p}(v) = \theta p^{\theta-1} v, \tag{42}$$

$$g_p^{\theta\text{-E}}(v, w) = \frac{1}{\theta^2} g^{\mathrm{E}}(\mathrm{Pow}_{\theta*,p}(v), \mathrm{Pow}_{\theta*,p}(w)) = \langle p^{\theta-1} v, p^{\theta-1} w \rangle = p^{2(\theta-1)} vw, \tag{43}$$

$$\begin{aligned}
\gamma_{(p,v)}(t) &= \mathrm{Pow}_\theta^{-1}\left(\mathrm{Pow}_\theta(p) + t\,\mathrm{Pow}_{\theta*,p}(v)\right) \\
&= (p^\theta + t\theta p^{\theta-1} v)^{\frac{1}{\theta}} \\
&= p(1 + t\theta p^{-1} v)^{\frac{1}{\theta}}, \text{ with } t \in \{t \in \mathbb{R} \mid 1 + t\theta p^{-1} v \in \mathbb{R}_{++}\},
\end{aligned} \tag{44}$$

$$\begin{aligned}
\mathrm{Log}_p(q) &= \mathrm{Pow}_{\theta*,p}^{-1}\left(\mathrm{Pow}_\theta(q) - \mathrm{Pow}_\theta(p)\right) \\
&= \frac{1}{\theta} p^{1-\theta}(q^\theta - p^\theta) = \frac{1}{\theta} p\left(\left(\frac{q}{p}\right)^\theta - 1\right),
\end{aligned} \tag{45}$$

$$\mathrm{PT}_{p \to q}(v) = \mathrm{Pow}_{\theta*,q}^{-1}(\mathrm{Pow}_{\theta*,p}(v)) = \left(\frac{q}{p}\right)^{1-\theta} v. \tag{46}$$

The geodesic distance between $p$ and $q$ under $g^{\theta\text{-E}}$ is given as

$$\mathrm{d}^2(p, q) = g_p^{\theta\text{-E}}(\mathrm{Log}_p(q), \mathrm{Log}_p(q)) = \frac{1}{\theta^2}(q^\theta - p^\theta)^2. \tag{47}$$

The weighted Fréchet mean of $\{p_i \in \mathbb{R}_{++}\}_{i=1}^N$ with weights $\{w_i\}_{i=1}^N$ satisfying $w_i > 0$ for all $i$ and $\sum_i w_i = 1$ under $g^{\theta\text{-E}}$ is defined as

$$\mathrm{WFM}(\{p_i\}, \{w_i\}) = \operatorname*{argmin}_{p \in \mathbb{R}_{++}} \sum_{i=1}^N w_i\, \mathrm{d}^2(p, p_i). \tag{48}$$

Obviously, the WFM of $\{p_i\}$ under $g^{\theta\text{-E}}$ is the same as the one under $\mathrm{Pow}_\theta^* g^{\mathrm{E}}$. Due to the isometry of $\mathrm{Pow}_\theta^* g^{\mathrm{E}}$ to $g^{\mathrm{E}}$, the WFM of $\{p_i\}$ under $g^{\theta\text{-E}}$ can be calculated as

$$\mathrm{WFM}(\{p_i\}, \{w_i\}) = \mathrm{Pow}_\theta^{-1}\left(\mathrm{WFM}^{\mathrm{E}}(\{\mathrm{Pow}_\theta(p_i)\}, \{w_i\})\right) = \left(\sum_{i=1}^N w_i p_i^\theta\right)^{\frac{1}{\theta}}, \tag{49}$$

where $\mathrm{WFM}^{\mathrm{E}}$ in Eq. (49) is the Euclidean WFM, which is the familiar weighted average.

So far, we have obtained all the necessary Riemannian operators on $\{\mathbb{R}_{++}, g^{\theta\text{-E}}\}$. Combined with the Euclidean space $\mathcal{SL}^n$, one can obtain the results in the theorem. $\qquad\square$

### F.2 PROOF OF THM. 3.3

*Proof.* Similar to the proof of Thm. 3.2, we only need to show the Riemannian operators of $\{\mathbb{R}_{++}, g^{m\text{-BW}}\}$ with $m \in \mathbb{R}_{++}$. The expressions of Riemannian operators under GBWM can be found in Han et al. (2023). Here, we further simplify the associated expressions for the 1-dimensional case. Specifically, given $p, q \in \mathbb{R}_{++}$ and $w, v \in T_p \mathbb{R}_{++}$, we have the following:

$$\mathcal{L}_{p,m}(v) = \frac{v}{2mp}, \tag{50}$$

$$g_p^{m\text{-BW}}(v, w) = \frac{1}{2}\langle \mathcal{L}_{p,m}(v), w \rangle = \frac{vw}{4mp}, \tag{51}$$

$$\gamma_{(p,v)}(t) = p + tv + \mathcal{L}_{p,m}(tv)^2 m^2 p = p + tv + \frac{(tv)^2}{4p} = p\left(1 + \frac{t}{2}\frac{v}{p}\right)^2, \tag{52}$$

$$\mathrm{Log}_p(q) = 2\left(m(m^{-2}pq)^{\frac{1}{2}} - p\right) = 2\left((pq)^{\frac{1}{2}} - p\right) = 2p\left(\left(\frac{q}{p}\right)^{\frac{1}{2}} - 1\right), \tag{53}$$

$$\mathrm{d}^2(p,q) = g_p^{m\text{-BW}}(\mathrm{Log}_p(q), \mathrm{Log}_p(q)) = \frac{1}{4mp} 4 \left( (pq)^{\frac{1}{2}} - p \right)^2$$

$$= \frac{1}{mp} \left( pq - 2p^{\frac{3}{2}} q^{\frac{1}{2}} + p^2 \right)$$

$$= m^{-1} \left( q - 2p^{\frac{1}{2}} q^{\frac{1}{2}} + p \right) \tag{54}$$

$$= \left( m^{-\frac{1}{2}} \left( q^{\frac{1}{2}} - p^{\frac{1}{2}} \right) \right)^2.$$

As shown by Han et al. (2023), GBWM on $\mathcal{S}_{++}^n$ is the pullback metric of BWM by $\pi(S) = M^{-\frac{1}{2}} S M^{-\frac{1}{2}}$ for all $S \in \mathcal{S}_{++}^n$ with $M \in \mathcal{S}_{++}^n$. For the specific $\mathbb{R}_{++} \cong \mathcal{S}_{++}^1$, the isometry is simplified as

$$\pi(p) = m^{-1} p, \forall p \in \mathbb{R}_{++} \text{ with } m \in \mathbb{R}_{++}. \tag{55}$$

The geodesic $\widetilde{\gamma}_{(p,v)}(t)$ under BWM over $\mathbb{R}_{++}$ exists in the interval $\{t \in \mathbb{R} | 1 + t\mathcal{L}_p(v) \in \mathbb{R}_{++}\}$ (Malagò et al., 2018), which can be simplified as $\left\{ t \in \mathbb{R} | 1 + t\frac{v}{2p} \in \mathbb{R}_{++} \right\}$. Therefore, the geodesic $\gamma_{(p,v)}(t)$ under $g^{m\text{-BW}}$ exists in the interval:

$$\left\{ t \in \mathbb{R} | 1 + t\frac{\pi_{*,p}(v)}{2\pi(p)} \in \mathbb{R}_{++} \right\} = \left\{ t \in \mathbb{R} | 1 + t\frac{v}{2p} \in \mathbb{R}_{++} \right\} \tag{56}$$

According to Thanwerdas and Pennec (2023, Tab. 6), the parallel transportation on $\{\mathbb{R}_{++}, g^{\text{BW}}\}$ is

$$\widetilde{\mathrm{PT}}_{p \to q}(v) = \left( \frac{q}{p} \right)^{\frac{1}{2}} v \tag{57}$$

Therefore, the parallel transportation on $\{\mathbb{R}_{++}, g^{m\text{-BW}}\}$ is

$$\mathrm{PT}_{p \to q}(v) = \pi_{*,q}^{-1} \left( \widetilde{\mathrm{PT}}_{\pi(p) \to \pi(q)}(\pi_{*,p}(v)) \right) = \pi_{*,q}^{-1} \left( \left( \frac{\pi(q)}{\pi(p)} \right)^{\frac{1}{2}} \pi_{*,p}(v) \right)$$

$$= \left( \frac{q}{p} \right)^{\frac{1}{2}} v. \tag{58}$$

Lastly, we show the WFM on $\{\mathbb{R}_{++}, g^{m\text{-BW}}\}$. Given $\{p_i \in \mathbb{R}_{++}\}_{i=1}^N$ with weights $\{w_i\}_{i=1}^N$ satisfying $w_i > 0$ for all $i$ and $\sum_{i=1}^N w_i = 1$, the WFM on $\{\mathbb{R}_{++}, g^{m\text{-BW}}\}$ is

$$\mathrm{WFM}(\{p_i\}, \{w_i\}) = \underset{p \in \mathbb{R}_{++}}{\mathrm{argmin}} \sum_{i=1}^N w_i \, \mathrm{d}^2(p, p_i)$$

$$= \underset{p \in \mathbb{R}_{++}}{\mathrm{argmin}} \sum_{i=1}^N w_i m^{-1} \left( p^{\frac{1}{2}} - p_i^{\frac{1}{2}} \right)^2$$

$$= \underset{p \in \mathbb{R}_{++}}{\mathrm{argmin}} \sum_{i=1}^N w_i \left( p^{\frac{1}{2}} - p_i^{\frac{1}{2}} \right)^2 \tag{59}$$

$$= \underset{p \in \mathbb{R}_{++}}{\mathrm{argmin}} \sum_{i=1}^N w_i \left( p - 2p^{\frac{1}{2}} p_i^{\frac{1}{2}} \right)$$

$$= \underset{p \in \mathbb{R}_{++}}{\mathrm{argmin}} \quad p - 2p^{\frac{1}{2}} \sum_{i=1}^N w_i p_i^{\frac{1}{2}}.$$

Let $f(p) = p - 2p^{\frac{1}{2}} \sum_{i=1}^N w_i p_i^{\frac{1}{2}}$. Then, the 1st and 2nd order derivatives are

$$\frac{\mathrm{d} f}{\mathrm{d} p} = 1 - p^{-\frac{1}{2}} \sum_i w_i p_i^{\frac{1}{2}}, \tag{60}$$

$$\frac{\mathrm{d}^2 f}{\mathrm{d}\,p^2} = \frac{1}{2}p^{-\frac{3}{2}}\sum_i w_i p_i^{\frac{1}{2}} > 0, \quad \forall p \in \mathbb{R}_{++}. \tag{61}$$

Therefore, the optimal solution can be obtained by setting Eq. (60) equal to 0:

$$\frac{\mathrm{d}\,f}{\mathrm{d}\,p} = 0 \Rightarrow p^* = \left(\sum_i w_i p_i^{\frac{1}{2}}\right)^2. \tag{62}$$

Combining the above results with the Euclidean geometry on $\mathcal{SL}^n$, one can readily obtain the results. $\qquad\square$

### F.3 PROOF OF LEM. 3.5

*Proof.* The differential of $\mathrm{DPow}_\theta$ at $\mathbb{L} \in \mathbb{D}_{++}^n$ is

$$\mathrm{DPow}_{\theta*,\mathbb{L}}(\mathbb{V}) = \theta\mathbb{L}^{\theta-1}\mathbb{V}, \quad \forall\mathbb{V} \in T_\mathbb{L}\mathcal{L}_{++}^n. \tag{63}$$

Putting Eq. (63) into Def. 3.4, one can easily get the results. $\qquad\square$

### F.4 PROOF OF LEM. 3.6

We first present a useful lemma.

**Lemma F.1.** *The Riemannian exponentiation, logarithm, and parallel transportation along the geodesic are the same under $(\theta, \mathbb{M})$-DBWM and $\theta/2$-DPM.*

*Proof.* This can be directly obtained by Thms. 3.2 and 3.3. $\qquad\square$

Now we begin to prove Lem. 3.6.

*Proof.* According to Lem. F.1, we only need to show the expression of $\theta/2$-DPM. We omit the superscript $\mathcal{C}$ for simplicity.

For $X \in T_L\mathcal{L}_{++}^n$, we have the following:

$$\mathrm{Log}_I(L) = \lfloor L \rfloor + \frac{1}{\theta}\left(\mathbb{L}^\theta - I\right), \tag{64}$$

$$\mathrm{PT}_{I \to L}(X) = \lfloor X \rfloor + \mathbb{L}^{1-\theta}\mathbb{X}, \tag{65}$$

$$\mathrm{Exp}_I X = \lfloor X \rfloor + (I + \theta\mathbb{X})^{\frac{1}{\theta}}. \tag{66}$$

For the binary operation, putting Eqs. (64) and (65) into Eq. (7), we have

$$\begin{aligned} L \oplus K &= \mathrm{Exp}_L\left(\mathrm{PT}_{I \to L}\left(\mathrm{Log}_I(K)\right)\right) \\ &= \mathrm{Exp}_L\left(\mathrm{PT}_{I \to L}\left(\lfloor K \rfloor + \frac{1}{\theta}\left(\mathbb{K}^\theta - I\right)\right)\right) \\ &= \mathrm{Exp}_L\left(\lfloor K \rfloor + \frac{1}{\theta}\mathbb{L}^{1-\theta}\left(\mathbb{K}^\theta - I\right)\right) \\ &= \lfloor L \rfloor + \lfloor K \rfloor + \mathbb{L}\left(I + \theta\mathbb{L}^{-1}\left(\frac{1}{\theta}\mathbb{L}^{1-\theta}\left(\mathbb{K}^\theta - I\right)\right)\right)^{\frac{1}{\theta}} \\ &= \lfloor L \rfloor + \lfloor K \rfloor + \left(L^\theta + K^\theta - I\right)^{\frac{1}{\theta}}. \end{aligned} \tag{67}$$

In the 3rd row of Eq. (67), the well-definedness of exponential map requires

$$\mathbb{L} + \theta\left[\frac{1}{\theta}\mathbb{L}^{1-\theta}\left(\mathbb{K}^\theta - I\right)\right] \in \mathbb{D}_{++}^n \Leftrightarrow L^\theta + K^\theta - I \in \mathbb{D}_{++}^n. \tag{68}$$

For the gyromultiplication, injecting Eqs. (64) and (66) into Eq. (8), we have

$$
\begin{aligned}
t \odot L &= \operatorname{Exp}_I(t \operatorname{Log}_I(L)) \\
&= \operatorname{Exp}_I \left( t \lfloor L \rfloor + \frac{t}{\theta} \left( \mathbb{L}^\theta - I \right) \right) \\
&= t \lfloor L \rfloor + \left( I + t \left( \mathbb{L}^\theta - I \right) \right)^{\frac{1}{\theta}} \\
&= t \lfloor L \rfloor + \left( t \mathbb{L}^\theta + (1 - t)I \right)^{\frac{1}{\theta}}.
\end{aligned}
\tag{69}
$$

In the 2nd row of Eq. (69), the exponential map requires

$$
I + \theta \left( \frac{t}{\theta} \left( \mathbb{L}^\theta - I \right) \right) \in \mathbb{D}_{++}^n \Leftrightarrow t\mathbb{L}^\theta + (1 - t)I \in \mathbb{D}_{++}^n.
\tag{70}
$$

$\square$

## F.5 PROOF OF THM. 3.7

*Proof.* In this proof, we assume $L, K, J \in \mathcal{L}_{++}^n$ and $s, t \in \mathbb{R}$. Note that as stated in the main paper, we assume all the gyro operations satisfy the required assumption presented in Lem. 3.6. As indicated by Lem. F.1, we only need to prove the case of $\theta$-DPM. For simplicity, we omit the superscript $\mathcal{C}$.

**Axiom (G1):** Eq. (7) implies that the identity element is the identity matrix.

**Axiom (G2):** We define the inverse element of $L$ as

$$
\ominus L = -1 \odot L = -\lfloor L \rfloor + \left( 2I - \mathbb{L}^\theta \right)^{\frac{1}{\theta}}.
\tag{71}
$$

Simple computations show that $\ominus L \oplus L = I$.

**Axiom (G3):** Gyroaddition in Lem. 3.6 indicates that

$$
L \oplus (K \oplus J) = (L \oplus K) \oplus J = \lfloor L \rfloor + \lfloor K \rfloor + \lfloor J \rfloor + \left( \mathbb{L}^\theta + \mathbb{K}^\theta + \mathbb{J}^\theta - 2I \right)^{\frac{1}{\theta}}.
\tag{72}
$$

Therefore the gyroautomorphism is the identity map, *i.e.* $\operatorname{gyr}[L, K] = \operatorname{id}$.

**Axiom (G4):** This is a direct corollary of (G3).

**Gyrocommutative law:** Gyroaddition in Lem. 3.6 indicates that

$$
L \oplus K = K \oplus L.
\tag{73}
$$

**Axiom (V1):** This can be obtained by gyromultiplication in Lem. 3.6 and Eq. (71).

**Axiom (V2):**

$$
\begin{aligned}
(s + t) \odot L &= (s + t)\lfloor L \rfloor + \left( (s + t)\mathbb{L}^\theta + (1 - (s + t))I \right)^{\frac{1}{\theta}} \\
&= s\lfloor L \rfloor + t\lfloor L \rfloor + \left( s\mathbb{L}^\theta + (1 - s)I + t\mathbb{L}^\theta + (1 - t)I - I \right)^{\frac{1}{\theta}} \\
&= (s \odot L) \oplus (t \odot L).
\end{aligned}
\tag{74}
$$

**Axiom (V3):**

$$
\begin{aligned}
(st) \odot L &= (st)\lfloor L \rfloor + \left( st\mathbb{L}^\theta + (1 - st)I \right)^{\frac{1}{\theta}} \\
&= (st)\lfloor L \rfloor + \left( s \left( t\mathbb{L}^\theta + (1 - t)I \right) + (1 - s)I \right)^{\frac{1}{\theta}} \\
&= s \odot \left[ t\lfloor L \rfloor + \left( t\mathbb{L}^\theta + (1 - t)I \right)^{\frac{1}{\theta}} \right] \\
&= s \odot (t \odot L).
\end{aligned}
\tag{75}
$$

**Axioms (V4) and (V5):** These two axioms can be directly obtained, as gyroautomorphisms are all identity maps. $\square$

### F.6 PROOF OF THM. 4.1

*Proof.* According to Nguyen and Yang (2023, Thm. 2.4), the gyro vectors can be preserved by Riemannian isometry. Besides, the Cholesky decomposition is the Riemannian isometry:

$$\mathrm{Chol} : \{\mathcal{S}_{++}^n, g^{\mathcal{S}}\} \to \{\mathcal{L}_{++}^n, g^{\mathcal{C}}\}. \tag{76}$$

□

### F.7 PROOF OF THM. 5.1

*Proof.* Putting the associated operators in Sec. 4 into Eq. (19) in Lem. H.1 by Chen et al. (2024c), one can directly obtain the results. □

