# OpenReview forum: "Fast and Stable Riemannian Metrics on SPD Manifolds via Cholesky Product Geometry"
_ICLR.cc/2026/Conference — ICLR 2026 Poster_

### Official Review · Reviewer_QVyo · 2025-10-25

**Soundness:** 3
**Presentation:** 3
**Contribution:** 3
**Rating:** 6
**Confidence:** 3

**Summary:**

The paper revisits the Cholesky geometry of SPD matrices and exposes a product structure: a Euclidean metric on the strictly lower-triangular part plus (n) copies of a 1-D positive-reals metric on the diagonal. Leveraging this, the authors define two new Cholesky-side metrics -- Diagonal Power Metric (θ-DPM) and Diagonal Bures-Wasserstein Metric ((θ,M)-DBWM)--and pull them back through Cholesky to SPD metrics: Power-Cholesky Metric ($\theta$-PCM) and Bures-Wasserstein-Cholesky Metric (($\theta$,M)-BWCM). These metrics admit closed-form operators (geodesic, log/exp, parallel transport, Frechet mean) and gyrovector operations, while replacing diagonal log/exp by diagonal powers for improved numerical stability. Empirically, they integrate the metrics into Riemannian MLR classifiers and Riemannian residual blocks, showing accuracy/runtime advantages over AIM/LEM/LCM on Radar, HDM05, and FPHA; and they present large-scale stability tests (failure rates under tiny diagonal entries) favoring the proposed metrics.

**Strengths:**

* Originality: Product-geometry perspective on Cholesky leading to $\theta$-PCM/($\theta$,M)-BWCM with closed-form operators and gyro-structures.
* Quality: Mathematical development is careful; operator summaries are comprehensive; empirical evaluations show consistent accuracy/runtime gains and robustness under tiny eigenvalues.
* Clarity: Pipeline overview and tabulated operators (Table 1) facilitate adoption.
* Significance: Practical drop-ins for SPD MLR and residual blocks; mitigates numerical brittleness of diagonal log/exp while retaining LCM-like utility.

**Weaknesses:**

1. Baseline breadth. Experiments primarily compare to AIM/LEM/LCM; PEM and (G)BWM baselines as end-to-end learners are not presented, leaving open how much improvement stems from closed-form convenience vs. intrinsic geometry.
2. Assumption transparency. Gyro-structures and some exp/log formulas are locally defined and require ($L^\beta$) feasibility (e.g., ($L^\beta+K^\beta-I\in D_{++}$)); these constraints aren’t prominently quantified for training dynamics.
3. Reproducibility details. While mean$\pm$std are reported, seed counts, episode splits (for cross-dataset parity), and tuning parity across metrics could be clearer; code is not yet available (this is highly preferred).
4. Invariance discussion. The paper could explicitly contrast affine-invariance and other desirable properties of existing metrics vs. the proposed ones, and discuss trade-offs.

**Questions:**

1. Invariance & trade-offs. How do $\theta$-PCM/($\theta$,M)-BWCM compare to AIM’s affine-invariance in practice? Are there scenarios where losing/altering invariance harms performance, and can deformation ($\theta$) mitigate this?
2. Baseline expansion. Can you include PEM and BWM/GBWM (even via numerical solvers) as end-to-end baselines on at least one dataset to calibrate geometry choice vs. closed-form convenience?
3. Domain of validity. Please quantify the practical parameter/step-size ranges ensuring (L^\beta) feasibility during training; do you observe violations, and how are they handled (clamping, retraction)?
4. $\theta$ sensitivity. Provide systematic sweeps of ($\theta$) (and (M)) across tasks; when does the limit ($\theta\rightarrow 0$) (LCM-like) help/hurt in practice?
5. Scaling & large-scale tasks. Any results with high-dimensional SPD in modern deep nets (e.g., covariance pooling on ImageNet-scale features) to validate computational/storage gains?
6. Code & seeds. Please clarify #seeds, hyperparameter parity, and release a minimal reference implementation for log/exp/transport under your metrics to facilitate adoption.

---

> ### Author Response · Authors · 2025-11-27
> **Responses to Reviewer QVyo (1/4)**
>
> We thank Reviewer $\textcolor{purple}{QVyo}$ for the encouraging feedback and the constructive comments!
> ***
>
>
> # 1. Comparison with Bures–Wasserstein Metric (BWM) and Power-Euclidean Metric (PEM).
>
> Tab. A: Comparison of our metrics with AIM, LEM, LCM, PEM, and BWM on SPD MLR under the SPDNet backbone.
> | Metric | Radar | Radar | HDM05 | HDM05 | FPHA | FPHA |
> |:------:|:-----:|:-----:|:-----:|:-----:|:----:|:----:|
> |  | Acc | Time | Acc | Time | Acc | Time |
> | AIM | 94.53 ± 0.95 | 0.80 | 61.14 ± 0.94 | 19.23 | 85.57 ± 0.50 | 7.14 |
> | LEM | 93.55 ± 1.21 | 0.76 | 60.28 ± 0.91 | 3.50 | 85.90 ± 0.47 | 0.98 |
> | LCM | 93.49 ± 1.25 | 0.72 | 62.33 ± 2.15 | 2.90 | 86.37 ± 0.59 | 0.74 |
> | PEM | 94.21 ± 1.03 | 0.74 | **70.22 ± 0.81** | 2.87 | 77.80 ± 0.41 | 0.66 |
> | BWM | 92.22 ± 0.83 | 0.93 | 70.20 ± 0.91 | 20.56 | 74.07 ± 0.31 | 7.36 |
> | PCM | **95.79 ± 0.38** | 0.72 | 65.75 ± 2.86 | 2.76 | **89.40 ± 0.13** | 0.69 |
> | BWCM | 93.93 ± 0.79 | 0.71 | 67.40 ± 0.90 | 2.87 | 86.27 ± 0.60 | 0.70 |
>
> Tab. B: Comparison of our metrics with AIM, LEM, LCM, PEM, and BWM on SPD MLR under the GyroSPD backbone.
> | Metric | Radar | Radar | HDM05 | HDM05 | FPHA | FPHA |
> |:------:|:-----:|:-----:|:-----:|:-----:|:----:|:----:|
> |  | Acc | Time | Acc | Time | Acc | Time |
> | AIM | 96.80 ± 0.59 | 1.23 | 66.05 ± 1.80 | 21.65 | 85.77 ± 0.52 | 11.48 |
> | LEM | 96.58 ± 0.27 | 1.18 | 66.42 ± 0.47 | 2.02 | 85.87 ± 0.79 | 1.22 |
> | LCM | 96.29 ± 0.53 | 1.12 | 68.37 ± 0.66 | 1.66 | 89.83 ± 0.28 | 0.98 |
> | PEM | 95.79 ± 0.69 | 1.12 | 70.23 ± 0.75 | 2.11 | 84.53 ± 1.58 | 1.24 |
> | BWM | 96.24 ± 0.75 | 1.63 | 71.63 ± 1.36 | 24.35 | 84.43 ± 0.50 | 13.08 |
> | PCM | **97.04 ± 0.64** | 1.18 | 71.93 ± 1.21 | 1.51 | **91.17 ± 0.30** | 1.00 |
> | BWCM | 96.21 ± 0.25 | 1.05 | **72.74 ± 0.43** | 1.58 | 91.00 ± 0.11 | 0.96 |
>
> **Setup.** We compare our metrics with PEM [a] and BWM [b] in the SPD MLR setting on Radar, HDM05, and FPHA, using both SPDNet and GyroSPD backbones. The PEM- and BWM-based MLRs are implemented following [d, Thm. 4.2], and all other settings follow the main experiments.
>
> **Better effectiveness and efficiency.** Tabs. A and B report the 5-fold accuracy and per-epoch runtime. Across all settings except SPDNet on HDM05, PCM/BWCM achieve higher accuracy than all other metrics. The gain over PEM and BWM is especially large on FPHA with the SPDNet backbone, where accuracy improves from 77.80% for PEM and 74.07% for BWM to 89.40% for PCM. In terms of efficiency, PCM/BWCM have runtime faster than PEM while providing substantial speedups over BWM. For example, on GyroSPD with HDM05, the runtime drops from 24.35s for BWM to about 1.5s for PCM/BWCM, which is nearly a 16× improvement.
>
> # 2. Locality and numerical remedy
>
> - **Locality.** Among our Riemannian operators, the only one that is not globally defined is the Riemannian exponential map. The Riemannian logarithm, parallel transport, geodesic distance, and Fréchet mean are globally well-defined. Moreover, since the gyro operators in Eqs. (7–8) involve the exponential map, they inherit the locality. This pattern also appears in other popular geometries. The Bures–Wasserstein metric has locally defined exponential map [b, Prop. 5]. On Grassmann manifolds, logarithmic maps and gyro-structures are defined only away from the cut locus [c, Sec. 5; d, Sec. 3.2].
>
> - **Numerical remedy.** Since the locality arises only through diagonal terms, a simple scalar regularization on the Cholesky diagonals is sufficient. Taking $ L ^{\beta} + K ^{\beta} - I \in D _n ^{++}$ as an example, we can apply an elementwise clipping on each diagonal entry,
> $$
> d _i \leftarrow \max(d _i, \varepsilon),
> $$
> with a small $ \varepsilon$ (e.g., $10 ^{-6}$), which guarantees positive diagonals ($ D _n ^{++}$). This cheap scalar operation introduces negligible overhead. We have added the above strategy explicitly in Rmk. 3.9 of the revised manuscript.
>
> - **Effects on training.** The SPD residual blocks (see Eq. (16)) involve the exponential map, which may trigger the above locality issues during training. However, in the original Riemannian ResNet implementation [g], the tangent vector is normalized by the Frobenius norm before applying the exponential map to avoid numerical instability of AIM. Under this normalization, the diagonal clipping described above acts only as a rare safeguard. Tab. 4 shows that the residual block under our metrics consistently outperforms that under LEM/LCM/AIM, indicating that locality constraints do not hinder effective training.

---

> > ### Author Response · Authors · 2025-11-27
> > **Responses to Reviewer QVyo (2/4)**
> >
> > # 3. Reproducibility details and minimal implementation
> >
> > App. E.1 provides the dataset preprocessing and training configurations. For every experiment we use a fixed random seed of 42.
> > - For SPD MLR on SPDNet, the baseline results on Radar and HDM05 are taken from the original paper [e, Tabs. 3–4], and the remaining results are reproduced using the official code [e].
> > - For SPD MLR on GyroSPD, the backbone code is not publicly available. We reimplemented the GyroSPD architecture following the specifications in [f].
> > - For Riemannian residual networks, we follow the official code in [g].
> >
> > Across all the above experiments, all metrics share exactly the same training protocol, including backbone architecture, optimizer, learning rate, batch size, number of epochs, and data splits, to ensure fair comparison.
> >
> > A minimal implementation of our metrics can be found at the anonymous GitHub repo: https://anonymous.4open.science/r/PCM_BWCM_ICLR161. Other code will be released upon acceptance.
> >
> > # 4. Scalability
> >
> > The covariance pooling [h] typically relies on fast iterative approximations for matrix functions, such as Newton–Schulz iterations for matrix square roots. Adapting our Cholesky-based metrics to this setting would require designing and analysing new approximate algorithms for the Cholesky decomposition, which is beyond the scope of the current work. Instead, we conduct numerical experiments to validate the scalability of our metrics.
> >
> > We measure the efficiency of different metrics as the matrix dimension increases. We use an SPD MLR with 50 output classes followed by a cross-entropy loss. For each dimension $n$, we randomly generate a batch of 30 $ n \times n$ SPD matrices, and for PEM we fix the matrix power to $0.5$. Tab. C reports the per-batch runtimes of a single forward and backward training step. PCM and BWCM achieve the best efficiency across all dimensions and significantly reduce runtime compared with BWM and AIM.
> >
> > Tab. C: Per-batch runtime (seconds) of SPD MLRs under different metrics.
> >
> > | Dim | AIM | LEM | LCM | PEM | BWM | PCM | BWCM |
> > |:---:|:---:|:---:|:---:|:---:|:---:|:---:|:----:|
> > | 32 | 0.2380 | 0.0077 | 0.0046 | 0.0076 | 0.2377 | **0.0040** | **0.0040** |
> > | 64 | 1.0139 | 0.0395 | 0.0303 | 0.0473 | 1.1205 | **0.0251** | **0.0225** |
> > | 128 | 3.6256 | 0.1832 | 0.1490 | 0.1844 | 4.0674 | **0.1013** | **0.1019** |
> > | 256 | 14.5142 | 0.7793 | 0.5833 | 0.7853 | 16.5918 | **0.3848** | **0.4077** |
> > | 512 | 60.1918 | 3.2948 | 2.5030 | 3.4357 | 70.8647 | **1.7553** | **1.7526** |
> >
> > # 5. Invariance and trade-offs
> >
> > **1. Invariance**
> > Although our metrics are not affine-invariant, they admit gyro bi-invariance. For all feasible SPD matrices $P,Q,S$, we have
> > $$
> > d(S \oplus P, S \oplus Q)= d(P \oplus S, Q \oplus S)=d(P,Q),
> > $$
> > where $\oplus$ denotes gyroaddition (Eq. (11)) and $d$ is the geodesic distance. This is the natural non-Euclidean analogue of Euclidean translation-invariance:
> > $$
> > \lVert \left(x+a\right)-\left(y+a\right) \rVert = \lVert x-y \rVert, \quad \forall x, y, a \in \mathbb{R} ^{n}.
> > $$
> >
> > **2. Trade-off**
> > - Although AIM enjoys affine-invariance, its Riemannian computations are more complex. AIM requires eigendecomposition with complexity $O\left(9n ^{3}\right)$ [i], while our metrics largely rely on the Cholesky decomposition with complexity $O\left(\tfrac{1}{3}n ^{3}\right)$ [i], which is significantly cheaper. Take the SPD MLR as an example. Tab. D summarizes the per-sample complexity of a $C$-class SPD MLR, where our metrics significantly reduce computational cost. Consistent with this analysis, Tabs. A, B, and C show that PCM and BWCM train faster than AIM and also achieve higher accuracy.
> > - In the next response, we also conduct ablation studies on the deformation $\theta$, ranging from $-2$ to $1.5$. The results show that most of the tested values outperform AIM.
> >
> > The above analysis demonstrates that our metrics serve as an efficient and effective alternative to AIM in SPD learning scenarios.
> >
> > Tab. D: Asymptotic per-sample complexity of a $C$-class SPD MLR for an $n \times n$ input SPD matrix.
> > | Metric | Asymptotic complexity |
> > |:------:|:----------------------|
> > | AIM | $O\left(9(1 + 2C)n ^{3}\right)$ |
> > | PCM | $O\left(\tfrac{1 + C}{3}n ^{3}\right)$ |
> > | BWCM | $O\left(\tfrac{1 + C}{3}n ^{3}\right)$ |

---

> ### Author Response · Authors · 2025-11-27
> **Responses to Reviewer QVyo (3/4)**
>
> # 6. Ablations on $\theta$ and $\mathbb{M}$
>
> **1. Ablations on $\theta$.**
>
> Tab. E: Results on SPD MLR of PCM and BWCM on the SPDNet backbone for different values of $ \theta$. The best result in each row is shown in **bold**.
> | Dataset | Metric | -2 | -1.5 | -1 | -0.75 | -0.5 | -0.25 | 0.25 | 0.5 | 0.75 | 1 | 1.5 |
> |:-:|:-:|:--:|:----:|:--:|:---:|:--:|:--:|:----:|:---:|:----:|:-:|:---:|
> | Radar | PCM | 95.71 ± 0.57 | **95.79 ± 0.38** | 94.64 ± 0.34 | 95.36 ± 1.32 | 94.11 ± 1.05 | 94.51 ± 0.75 | 94.35 ± 0.71 | 93.55 ± 0.59 | 94.16 ± 0.96 | 93.87 ± 0.40 | 92.75 ± 0.68 |
> |  | BWCM | 91.89 ± 0.31 | 92.91 ± 1.05 | 92.16 ± 1.10 | **93.93 ± 0.79** | 92.13 ± 0.60 | 92.16 ± 1.30 | 92.40 ± 0.95 | 92.08 ± 0.85 | 92.48 ± 0.93 | 92.77 ± 0.66 | 91.81 ± 0.52 |
> | HDM05 | PCM | 46.63 ± 1.63 | 49.86 ± 1.41 | 58.81 ± 1.41 | 65.18 ± 2.89 | **65.75 ± 2.86** | 65.65 ± 2.03 | 64.00 ± 3.20 | 63.73 ± 2.90 | 64.59 ± 3.22 | 64.58 ± 3.14 | 64.62 ± 1.44 |
> |  | BWCM | 65.20 ± 2.63 | **67.40 ± 0.90** | 65.42 ± 1.88 | 65.29 ± 2.32 | 64.33 ± 2.71 | 64.07 ± 2.65 | 63.89 ± 2.50 | 64.67 ± 1.38 | 65.15 ± 0.75 | 63.33 ± 2.18 | 65.00 ± 1.48 |
> | FPHA | PCM | 88.20 ± 0.29 | 88.20 ± 0.32 | 88.30 ± 0.24 | 88.33 ± 0.21 | 88.47 ± 0.29 | 88.70 ± 0.39 | 89.03 ± 0.22 | 88.97 ± 0.16 | **89.40 ± 0.13** | 89.03 ± 0.16 | 89.23 ± 0.33 |
> |  | BWCM | 85.87 ± 0.66 | 85.87 ± 0.66 | 85.87 ± 0.66 | 85.93 ± 0.67 | 85.93 ± 0.67 | 85.93 ± 0.67 | 85.93 ± 0.67 | 86.00 ± 0.70 | 86.03 ± 0.75 | **86.27 ± 0.60** | 86.13 ± 0.73 |
>
> Tab. F: Results on SPD MLR of PCM and BWCM on the GyroSPD backbone for different values of $ \theta$. The best result in each row is shown in **bold**.
> | Dataset | Metric | -2 | -1.5 | -1 | -0.75 | -0.5 | -0.25 | 0.25 | 0.5 | 0.75 | 1 | 1.5 |
> |:-------:|:------:|:--:|:----:|:--:|:-----:|:----:|:-----:|:----:|:---:|:----:|:-:|:---:|
> | Radar | PCM | 96.88 ± 0.36 | 96.75 ± 0.26 | 96.27 ± 0.36 | **97.04 ± 0.64** | 96.45 ± 0.88 | 97.01 ± 0.66 | 96.77 ± 0.64 | 96.51 ± 0.65 | 96.56 ± 0.86 | 96.53 ± 0.79 | 96.32 ± 0.68 |
> |  | BWCM | 96.16 ± 0.90 | **96.21 ± 0.25** | 95.25 ± 1.19 | 96.16 ± 0.57 | 96.08 ± 0.48 | 95.84 ± 0.69 | 94.96 ± 0.94 | 95.60 ± 0.65 | 95.04 ± 0.33 | 94.77 ± 0.96 | 95.55 ± 0.91 |
> | HDM05 | PCM | 50.18 ± 0.99 | 32.49 ± 25.81 | 66.43 ± 1.22 | **71.93 ± 1.21** | 70.94 ± 1.17 | 69.64 ± 0.68 | 67.68 ± 0.92 | 68.24 ± 0.28 | 68.02 ± 0.40 | 67.80 ± 1.07 | 67.86 ± 1.69 |
> |  | BWCM | 72.64 ± 1.15 | **72.74 ± 0.43** | 69.58 ± 1.34 | 69.24 ± 1.14 | 68.84 ± 0.74 | 68.10 ± 1.02 | 67.72 ± 1.19 | 68.49 ± 0.64 | 68.08 ± 0.93 | 67.72 ± 0.58 | 67.45 ± 0.45 |
> | FPHA | PCM | 90.60 ± 0.58 | 91.17 ± 0.32 | 91.10 ± 0.29 | **91.17 ± 0.30** | 91.00 ± 0.30 | 90.87 ± 0.12 | 90.97 ± 0.16 | 90.97 ± 0.12 | 90.97 ± 0.07 | 90.90 ± 0.08 | 90.97 ± 0.22 |
> |  | BWCM | 90.87 ± 0.16 | 90.80 ± 0.07 | 90.83 ± 0.11 | 90.97 ± 0.07 | **91.00 ± 0.11** | **91.00 ± 0.11** | 90.97 ± 0.12 | 90.97 ± 0.12 | 90.93 ± 0.08 | 90.90 ± 0.08 | 90.87 ± 0.12 |
>
> Tab. G: Results of the LCM SPD MLR on the SPDNet and GyroSPD backbones, which is copied from Tabs. 2-3 from the manuscript.
> | SPDNet | SPDNet | SPDNet | GyroSPD | GyroSPD | GyroSPD |
> |:------:|:------:|:------:|:-------:|:-------:|:-------:|
> | Radar | HDM05 | FPHA | Radar | HDM05 | FPHA |
> | 93.49 ± 1.25 | 62.33 ± 2.15 | 86.37 ± 0.59 | 96.29 ± 0.53 | 68.37 ± 0.66 | 89.83 ± 0.28 |
>
> **Setup.** We study $\theta$ for PCM and BWCM in the SPD MLR on both SPDNet and GyroSPD backbones. We vary $\theta$ between $-2$ and $1.5$ while keeping all other settings as the main experiments.
>
> **Analysis.** Tabs. E and F report the 5-fold results on the SPDNet and GyroSPD backbones. We have the following observations.
> - **Effectiveness.** We can observe that tuning $ \theta$ yields modest improvements on Radar and FPHA, but stronger gains on HDM05. In the SPD MLR, $\theta$ is applied to the Cholesky diagonal entries (Thm. 5.1). Its effect therefore depends on how these diagonals are distributed. App. E.4 visualizes the Cholesky diagonals across the three datasets. Radar and FPHA have relatively balanced diagonals, whereas on HDM05 (especially at the SPDNet output) almost all diagonal entries lie below $0.5$ and most are close to zero. For such diagonals, a negative $ \theta$ amplifies the entries and activates them, providing more discriminative information.
> - **Comparison with LCM.** Tab. G reviews the results of LCM MLR on both backbones. We could observe that both PCM and BWCM outperform LCM in most of the tested $\theta$ values.
> - **Relative stability of BWCM.** On HDM05, when $ \theta$ takes small negative values (for example, $ \theta = -1.5$ or $ \theta = -2$), the accuracy of PCM deteriorates, whereas BWCM remains comparatively stable. This behaviour aligns with the formulation of their SPD MLRs (Thm. 5.1). Since PCM and BWCM differ mainly in the diagonal powers $(\cdot) ^{\theta}$ versus $(\cdot) ^{\theta/2}$, BWCM experiences a milder deformation for the same $ \theta$.
>
> **Note.** The above has been added as App. E.3 in the revised manuscript.

---

> > ### Author Response · Authors · 2025-11-27
> > **Responses to Reviewer QVyo (4/4)**
> >
> > **2. Ablations on $\mathbb{M}$.**
> >
> > Tab. C: Results of BWCM with or without learnable $\mathbb{M}$ for the SPD MLR on the SPDNet and GyroSPD backbones.
> > | Backbone | SPDNet | SPDNet | SPDNet | GyroSPD | GyroSPD | GyroSPD |
> > |:--------:|:------:|:------:|:------:|:-------:|:-------:|:-------:|
> > | Dataset | Radar | HDM05 | FPHA | Radar | HDM05 | FPHA |
> > | $ \mathbb{M}$-BWCM | **95.15 ± 0.61** | 66.71 ± 0.96 | 86.27 ± 0.63 | **96.93 ± 0.40** | 72.62 ± 0.43 | 91.00 ± 0.11 |
> > | BWCM | 93.93 ± 0.79 | **67.40 ± 0.90** | 86.27 ± 0.60 | 96.21 ± 0.25 | **72.74 ± 0.43** | 91.00 ± 0.11 |
> >
> > **Setup.** To examine the effect of the diagonal matrix $\mathbb{M}$ in our BWCM, we compare BWCM with fixed $ \mathbb{M} = I$ against a variant that learns a positive diagonal matrix $\mathbb{M}$ ($ \mathbb{M}$-BWCM). We follow the SPD MLR configurations on both SPDNet and GyroSPD backbones and use the same value of $ \theta$ for BWCM and $ \mathbb{M}$-BWCM, which is determined in $ \theta$-ablation experiments. In $ \mathbb{M}$-BWCM, we optimize an unconstrained diagonal vector $ v$ and set $ \mathbb{M} = \operatorname{diag}(\exp(v))$ so that the diagonal of $ \mathbb{M}$ remains positive. We initialize $ v$ to zero, corresponding to $ \mathbb{M} = I$ at the beginning of training. The size of $ v$ matches the input dimension of the SPD MLR: $8$, $30$, and $33$ on SPDNet and $20$, $3 \times 28$, and $9 \times 28$ on GyroSPD for Radar, HDM05, and FPHA (where $3$ and $9$ are channel dimensions).
> >
> > **Results.** Tab. C reports the 5-fold results. On both backbones, $ \mathbb{M}$-BWCM slightly improves accuracy on Radar, is marginally worse on HDM05, and matches BWCM on FPHA. Since the additional parameters introduced by $ \mathbb{M}$ are relatively small, their impact on the overall model is marginal. These results suggest that we can simply fix $ \mathbb{M} = I$ in BWCM, which is the setting used in our main experiments.
> >
> > **Note.** The above has been added as App. E.6 in the revised manuscript.
> >
> >
> > ***
> >
> > # References
> > > [a] Power Euclidean metrics for covariance matrices with application to diffusion tensor imaging
> > >
> > > [b] Wasserstein Riemannian geometry of Gaussian densities
> > >
> > > [c] A Grassmann Manifold Handbook: Basic Geometry and Computational Aspects
> > >
> > > [d] The Gyro-Structure of Some Matrix Manifolds
> > >
> > > [e] RMLR: Extending Multinomial Logistic Regression into General Geometries
> > >
> > > [f] Building Neural Networks on Matrix Manifolds: A Gyrovector Space Approach
> > >
> > > [g] Riemannian Residual Neural Networks
> > >
> > > [h] Global second-order pooling convolutional networks
> > >
> > > [i] Matrix Computations (4th edition)

---

> > > ### Comment · Reviewer_QVyo · 2025-11-27
> > > **Thanks for your thoughtful rebuttal**
> > >
> > > I thank all the authors for their great efforts in preparing this rebuttal.  I can sense they have taken this process very seriously, from making clarification (e.g. "Among our Riemannian operators, the only one that is not globally defined is the Riemannian exponential map. The Riemannian logarithm, parallel transport, geodesic distance, and Fréchet mean are globally well-defined."   I might have overlooked this in the first round review.) to addressing experimental issues with a full set of new design of experiments.   As I mentioned in the first round, the paper was already above the acceptance bar.  Given the new information provided in their rebuttals (including those to other reviewers), I would like to further raise my rating score.   Thank you for your excellent work.

---

> > > > ### Author Response · Authors · 2025-11-27
> > > >
> > > > Thank you very much for your quick and positive response. We appreciate your recognition of our rebuttal and are grateful for your thoughtful feedback, which helped improve the clarity and overall quality of our work.
> > > >
> > > > Thank you again for your constructive input and for engaging with our submission!

---

### Official Review · Reviewer_cJuQ · 2025-10-27

**Soundness:** 3
**Presentation:** 3
**Contribution:** 3
**Rating:** 8
**Confidence:** 3

**Summary:**

This paper revisits the geometry of the Cholesky manifold and identifies a product structure that decomposes the Cholesky factor into a Euclidean component. Leveraging this insight, the authors introduce two new Riemannian metrics on SPD manifold: PCM and BWCM. The authors apply the proposed metrics to build Riemannian MLR classifiers and residual blocks. The authors evaluate the two metrics and applications with 3 datasets: Radar, HDM05 and FPHA.

**Strengths:**

Replacing logarithms with power transforms to address numerical instability in LCM is well-motivated and empirically validated. The proposed metrics maintain closed-form. This paper is in general sound, the usability of the two proposed metrics is critical for deep learning integration.

**Weaknesses:**

This paper is in general well-wriiten, but there are still some concerns:
1. Several Riemannian and gyro-operators used in this work are only local defined. In particular, certain expressions are well-defined only under positivity constraints such as $L^{\beta} + K^{\beta} - I \in D_{n}^{++}$
2. 𝜃 power is a main contribution, however, the authors don't present how to choose 𝜃
3. Recent SPD learning works with GBWM-based classifiers are not experimentally compared
4. The stability of the DPM and DBWM are not empirically validated

**Questions:**

1. Can the authors provide guidance on how to choose the $\theta$ values
2. The authors presents the stability of the proposed metrics, but I didn't see how it is validated with experiments
3. As this paper is on Bures–Wasserstein–Cholesky metric, I think the authors should compare with othe works Bures–Wasserstein geometry, e.g.,
         1. Learning to Normalize on the SPD Manifold under Bures‑Wasserstein Geometry (Wang 2025)
         2. Learning with Symmetric Positive Definite Matrices via Generalized Bures‑Wasserstein Geometry (Han 2021)

---

> ### Author Response · Authors · 2025-11-27
> **Responses to Reviewer cJuQ (1/4)**
>
> We thank Reviewer $\textcolor{blue}{cJuQ}$ for the constructive suggestions and positive feedback!
> ***
>
> # 1. Locality and numerical remedy
>
> - **Locality.** Among our Riemannian operators, the only one that is not globally defined is the Riemannian exponential map. The Riemannian logarithm, parallel transport, geodesic distance, and Fréchet mean are globally well-defined. Moreover, since the gyro operators in Eqs. (7–8) involve the exponential map, they inherit the locality. This pattern also appears in other popular geometries. The Bures–Wasserstein metric has locally defined exponential map [a, Prop. 5]. On Grassmann manifolds, logarithmic maps and gyro-structures are defined only away from the cut locus [b, Sec. 5; c, Sec. 3.2].
>
> - **Numerical remedy.** Since the locality arises only through diagonal terms, a simple scalar regularization on the Cholesky diagonals is sufficient. Taking $ L ^{\beta} + K ^{\beta} - I \in D _n ^{++}$ as an example, we can apply an elementwise clipping on each diagonal entry,
> $$
> d _i \leftarrow \max(d _i, \varepsilon),
> $$
> with a small $ \varepsilon$ (e.g., $10 ^{-6}$), which guarantees positive diagonals ($ D _n ^{++}$). This cheap scalar operation introduces negligible overhead. We have added the above strategy explicitly in Rmk. 3.9 of the revised manuscript.

---

> ### Author Response · Authors · 2025-11-27
> **Responses to Reviewer cJuQ (2/4)**
>
> # 2. On tuning $\theta$.
> We first give the general rule of thumb for choosing $ \theta$, followed by detailed experimental results.
>
> **1. General rule.**
> Since our metrics converge to LCM when $ \theta \to 0$ (Lem. 3.5), it is natural to use moderate values around zero, such as $ \theta = \pm 0.5$, as default settings. The choice of $ \theta$ can then be refined in a dataset-dependent way. For datasets whose Cholesky diagonals are relatively balanced (e.g., Radar and FPHA), these default values already work well. For imbalanced datasets such as HDM05, tuning $ \theta$ brings larger gains. Besides, when most diagonal entries are close to zero, a negative $ \theta$ is particularly effective because it activates these entries and provides more discriminative information.
>
>
> Tab. A: Results on SPD MLR of PCM and BWCM on the SPDNet backbone for different values of $ \theta$. The best result in each row is shown in **bold**.
> | Dataset | Metric | -2 | -1.5 | -1 | -0.75 | -0.5 | -0.25 | 0.25 | 0.5 | 0.75 | 1 | 1.5 |
> |:-------:|:------:|:--:|:----:|:--:|:-----:|:----:|:-----:|:----:|:---:|:----:|:-:|:---:|
> | Radar | PCM | 95.71 ± 0.57 | **95.79 ± 0.38** | 94.64 ± 0.34 | 95.36 ± 1.32 | 94.11 ± 1.05 | 94.51 ± 0.75 | 94.35 ± 0.71 | 93.55 ± 0.59 | 94.16 ± 0.96 | 93.87 ± 0.40 | 92.75 ± 0.68 |
> |  | BWCM | 91.89 ± 0.31 | 92.91 ± 1.05 | 92.16 ± 1.10 | **93.93 ± 0.79** | 92.13 ± 0.60 | 92.16 ± 1.30 | 92.40 ± 0.95 | 92.08 ± 0.85 | 92.48 ± 0.93 | 92.77 ± 0.66 | 91.81 ± 0.52 |
> | HDM05 | PCM | 46.63 ± 1.63 | 49.86 ± 1.41 | 58.81 ± 1.41 | 65.18 ± 2.89 | **65.75 ± 2.86** | 65.65 ± 2.03 | 64.00 ± 3.20 | 63.73 ± 2.90 | 64.59 ± 3.22 | 64.58 ± 3.14 | 64.62 ± 1.44 |
> |  | BWCM | 65.20 ± 2.63 | **67.40 ± 0.90** | 65.42 ± 1.88 | 65.29 ± 2.32 | 64.33 ± 2.71 | 64.07 ± 2.65 | 63.89 ± 2.50 | 64.67 ± 1.38 | 65.15 ± 0.75 | 63.33 ± 2.18 | 65.00 ± 1.48 |
> | FPHA | PCM | 88.20 ± 0.29 | 88.20 ± 0.32 | 88.30 ± 0.24 | 88.33 ± 0.21 | 88.47 ± 0.29 | 88.70 ± 0.39 | 89.03 ± 0.22 | 88.97 ± 0.16 | **89.40 ± 0.13** | 89.03 ± 0.16 | 89.23 ± 0.33 |
> |  | BWCM | 85.87 ± 0.66 | 85.87 ± 0.66 | 85.87 ± 0.66 | 85.93 ± 0.67 | 85.93 ± 0.67 | 85.93 ± 0.67 | 85.93 ± 0.67 | 86.00 ± 0.70 | 86.03 ± 0.75 | **86.27 ± 0.60** | 86.13 ± 0.73 |
>
> Tab. B: Results on SPD MLR of PCM and BWCM on the GyroSPD backbone for different values of $ \theta$. The best result in each row is shown in **bold**.
> | Dataset | Metric | -2 | -1.5 | -1 | -0.75 | -0.5 | -0.25 | 0.25 | 0.5 | 0.75 | 1 | 1.5 |
> |:-------:|:------:|:--:|:----:|:--:|:-----:|:----:|:-----:|:----:|:---:|:----:|:-:|:---:|
> | Radar | PCM | 96.88 ± 0.36 | 96.75 ± 0.26 | 96.27 ± 0.36 | **97.04 ± 0.64** | 96.45 ± 0.88 | 97.01 ± 0.66 | 96.77 ± 0.64 | 96.51 ± 0.65 | 96.56 ± 0.86 | 96.53 ± 0.79 | 96.32 ± 0.68 |
> |  | BWCM | 96.16 ± 0.90 | **96.21 ± 0.25** | 95.25 ± 1.19 | 96.16 ± 0.57 | 96.08 ± 0.48 | 95.84 ± 0.69 | 94.96 ± 0.94 | 95.60 ± 0.65 | 95.04 ± 0.33 | 94.77 ± 0.96 | 95.55 ± 0.91 |
> | HDM05 | PCM | 50.18 ± 0.99 | 32.49 ± 25.81 | 66.43 ± 1.22 | **71.93 ± 1.21** | 70.94 ± 1.17 | 69.64 ± 0.68 | 67.68 ± 0.92 | 68.24 ± 0.28 | 68.02 ± 0.40 | 67.80 ± 1.07 | 67.86 ± 1.69 |
> |  | BWCM | 72.64 ± 1.15 | **72.74 ± 0.43** | 69.58 ± 1.34 | 69.24 ± 1.14 | 68.84 ± 0.74 | 68.10 ± 1.02 | 67.72 ± 1.19 | 68.49 ± 0.64 | 68.08 ± 0.93 | 67.72 ± 0.58 | 67.45 ± 0.45 |
> | FPHA | PCM | 90.60 ± 0.58 | 91.17 ± 0.32 | 91.10 ± 0.29 | **91.17 ± 0.30** | 91.00 ± 0.30 | 90.87 ± 0.12 | 90.97 ± 0.16 | 90.97 ± 0.12 | 90.97 ± 0.07 | 90.90 ± 0.08 | 90.97 ± 0.22 |
> |  | BWCM | 90.87 ± 0.16 | 90.80 ± 0.07 | 90.83 ± 0.11 | 90.97 ± 0.07 | **91.00 ± 0.11** | **91.00 ± 0.11** | 90.97 ± 0.12 | 90.97 ± 0.12 | 90.93 ± 0.08 | 90.90 ± 0.08 | 90.87 ± 0.12 |
>
> **2. Experiments.**
>
> **Setup.** We study $\theta$ for PCM and BWCM in the SPD MLR on both SPDNet and GyroSPD backbones. We vary $\theta$ between $-2$ and $1.5$ while keeping all other settings as the main experiments.
>
> **Analysis.** Tabs. A and B report the 5-fold results on the SPDNet and GyroSPD backbones. We can observe that tuning $ \theta$ yields modest improvements on Radar and FPHA, but stronger gains on HDM05. In the SPD MLR, $\theta$ is applied to the Cholesky diagonal entries (Thm. 5.1). Its effect therefore depends on how these diagonals are distributed. App. E.4 visualizes the Cholesky diagonals across the three datasets. Radar and FPHA have relatively balanced diagonals, whereas on HDM05 (especially at the SPDNet output) almost all diagonal entries lie below $0.5$ and most are close to zero. For such diagonals, a negative $ \theta$ amplifies the entries and activates them, providing more discriminative information.

---

> > ### Author Response · Authors · 2025-11-27
> > **Responses to Reviewer cJuQ (3/4)**
> >
> > # 3. Comparison with Bures–Wasserstein Metric (BWM).
> >
> > Tab. C: Comparison of our metrics with AIM, LEM, LCM, and BWM on SPD MLR under the SPDNet backbone.
> > | Metric | Radar | Radar | HDM05 | HDM05 | FPHA | FPHA |
> > |:------:|:-----:|:-----:|:-----:|:-----:|:----:|:----:|
> > |  | Acc | Time | Acc | Time | Acc | Time |
> > | AIM | 94.53 ± 0.95 | 0.80 | 61.14 ± 0.94 | 19.23 | 85.57 ± 0.50 | 7.14 |
> > | LEM | 93.55 ± 1.21 | 0.76 | 60.28 ± 0.91 | 3.50 | 85.90 ± 0.47 | 0.98 |
> > | LCM | 93.49 ± 1.25 | 0.72 | 62.33 ± 2.15 | 2.90 | 86.37 ± 0.59 | 0.74 |
> > | BWM | 92.22 ± 0.83 | 0.93 | **70.20 ± 0.91** | 20.56 | 74.07 ± 0.31 | 7.36 |
> > | PCM | **95.79 ± 0.38** | 0.72 | 65.75 ± 2.86 | 2.76 | **89.40 ± 0.13** | 0.69 |
> > | BWCM | 93.93 ± 0.79 | 0.71 | 67.40 ± 0.90 | 2.87 | 86.27 ± 0.60 | 0.70 |
> >
> > Tab. D: Comparison of our metrics with AIM, LEM, LCM, and BWM on SPD MLR under the GyroSPD backbone.
> > | Metric | Radar | Radar | HDM05 | HDM05 | FPHA | FPHA |
> > |:------:|:-----:|:-----:|:-----:|:-----:|:----:|:----:|
> > |  | Acc | Time | Acc | Time | Acc | Time |
> > | AIM | 96.80 ± 0.59 | 1.23 | 66.05 ± 1.80 | 21.65 | 85.77 ± 0.52 | 11.48 |
> > | LEM | 96.58 ± 0.27 | 1.18 | 66.42 ± 0.47 | 2.02 | 85.87 ± 0.79 | 1.22 |
> > | LCM | 96.29 ± 0.53 | 1.12 | 68.37 ± 0.66 | 1.66 | 89.83 ± 0.28 | 0.98 |
> > | BWM | 96.24 ± 0.75 | 1.63 | 71.63 ± 1.36 | 24.35 | 84.43 ± 0.50 | 13.08 |
> > | PCM | **97.04 ± 0.64** | 1.18 | 71.93 ± 1.21 | 1.51 | **91.17 ± 0.30** | 1.00 |
> > | BWCM | 96.21 ± 0.25 | 1.05 | **72.74 ± 0.43** | 1.58 | 91.00 ± 0.11 | 0.96 |
> >
> > **Setup.** We compare our metrics with BWM [a] in the SPD MLR setting on Radar, HDM05, and FPHA, using both SPDNet and GyroSPD backbones. The BWM-based MLR is implemented following [d, Thm. 4.2], and all other settings follow the main experiments.
> >
> > **Better effectiveness and efficiency.** Tabs. C and D report the 5-fold accuracy and per-epoch runtime. Across all settings except SPDNet on HDM05, PCM/BWCM achieve higher accuracy than BWM. The gain is especially large on FPHA with the SPDNet backbone, where accuracy improves from 74.07% for BWM to 89.40% for PCM. In terms of efficiency, PCM/BWCM also provide substantial speedups over BWM. In particular, on GyroSPD with HDM05 the per-epoch runtime drops from 24.35s for BWM to about 1.5s for PCM/BWCM, which is nearly a 16× improvement.

---

> ### Author Response · Authors · 2025-11-27
> **Responses to Reviewer cJuQ (4/4)**
>
> # 4. Empirical evidence for the stability of our metrics.
>
> Tab. E: Failure probabilities (%) of geodesics under different metrics with small diagonal values in $L$. An output matrix containing any Inf or NaN is considered a failure. DLM denotes the diagonal log metric on the Cholesky manifold, corresponding to the LCM on the SPD manifold. DPM/DBWM on the Cholesky manifold corresponds to the PCM and BWCM on the SPD manifold.
>
> | $ \epsilon$ | DLM (3×3) | DPM 1.5 (3×3) | DBWM 1.5 (3×3) | DPM 0.5 (3×3) | DBWM 0.5 (3×3) | DPM 0.15 (3×3) | DBWM 0.15 (3×3) | DLM (256×256) | DPM 1.5 (256×256) | DBWM 1.5 (256×256) | DPM 0.5 (256×256) | DBWM 0.5 (256×256) | DPM 0.15 (256×256) | DBWM 0.15 (256×256) |
> |:-----------:|:---------:|:-------------:|:---------------:|:-------------:|:---------------:|:---------------:|:----------------:|:-------------:|:-----------------:|:-------------------:|:-----------------:|:-------------------:|:-------------------:|:--------------------:|
> | $1e^{-1}$ | 0.62 | 0 | 0 | 0 | 0 | 0 | 0 | 14.29 | 0 | 0 | 0 | 0 | 0 | 0 |
> | $1e^{-2}$ | 5.70 | 0 | 0 | 0 | 0 | 0 | 0 | 18.48 | 0 | 0 | 0 | 0 | 0 | 0 |
> | $1e^{-3}$ | 51.32 | 0 | 0 | 0 | 0 | 0 | 0 | 58.35 | 0 | 0 | 0 | 0 | 0 | 0 |
> | $1e^{-4}$ | 94.34 | 0 | 0 | 0 | 0 | 0 | 0 | 95.02 | 0 | 0 | 0 | 0 | 0 | 0 |
> | $1e^{-5}$ | 99.39 | 0 | 0 | 0 | 0 | 0 | 0 | 99.47 | 0 | 0 | 0 | 0 | 0 | 0 |
> | $1e^{-10}$ | 100 | 0 | 0 | 0 | 0 | 0 | 0 | 100 | 0 | 0 | 0 | 0 | 0 | 0 |
> | $1e^{-15}$ | 100 | 0 | 0 | 0 | 0 | 0 | 0 | 100 | 0 | 0 | 0 | 0 | 0 | 0 |
> | $1e^{-20}$ | 100 | 0 | 0 | 0 | 0 | 0 | 0.002 | 100 | 0 | 0 | 0 | 0 | 0 | 0.02 |
> | $1e^{-21}$ | 100 | 0 | 0 | 0 | 0 | 0 | 0.03 | 100 | 0 | 0 | 0 | 0 | 0 | 0.01 |
> | $1e^{-22}$ | 100 | 0 | 0 | 0 | 0 | 0 | 0.25 | 100 | 0 | 0 | 0 | 0 | 0 | 0.23 |
> | $1e^{-23}$ | 100 | 0 | 0 | 0 | 0 | 0 | 2.26 | 100 | 0 | 0 | 0 | 0 | 0 | 2.42 |
> | $1e^{-24}$ | 100 | 0 | 0 | 0 | 0 | 0 | 22.98 | 100 | 0 | 0 | 0 | 0 | 0 | 23.13 |
> | $1e^{-25}$ | 100 | 0 | 0 | 0 | 0 | 0 | 86.34 | 100 | 0 | 0 | 0 | 0 | 0 | 86.58 |
> | $1e^{-30}$ | 100 | 0 | 0 | 0 | 0 | 0 | 100 | 100 | 0 | 0 | 0 | 0 | 0 | 100 |
>
> Sec. 6.2 evaluates the numerical stability of our metrics. Prior work [e] has shown that LCM is more stable than eigendecomposition-based metrics such as AIM and LEM, due to the stable Cholesky decomposition. Therefore, it is sufficient to compare our metrics against LCM for stability. The following is summarized from Sec. 6.2.
>
> **Setup.** Like LCM, our SPD metrics (PCM and BWCM) are induced from Cholesky metrics via the Cholesky decomposition. The SPD geodesic under these metrics has the form
> $$
> \gamma _{(P,V)} ^{\mathcal{S}}(t)
> = \operatorname{Chol} ^{-1} \left(\gamma _{(L,\widetilde{V})}(t)\right),
> $$
> where $ L = \operatorname{Chol}(P)$ is the Cholesky decomposition, $ \widetilde{V} = \operatorname{Chol} _{*,P}(V)$ is the differential map at $ P$, and $ \gamma$ is the geodesic in the Cholesky space. Therefore, it is enough to study stability in the Cholesky space.
>
> **Experiments.** We consider both small (3×3) and large (256×256) Cholesky matrices. For each size, we control the smallest Cholesky diagonal $ \epsilon$, vary it over several orders of magnitude, and measure the failure rate of geodesic computations, where a failure means that the result contains Inf or NaN. Tab. E presents the results. Across all tested $ \epsilon$, our metrics show significantly better stability than the existing DLM (corresponding to LCM), especially in the large 256×256 case.
> ***
>
> # References
> > [a] Wasserstein Riemannian geometry of Gaussian densities
> >
> > [b] A Grassmann Manifold Handbook: Basic Geometry and  Computational Aspects
> >
> > [c] The Gyro-Structure of Some Matrix Manifolds
> >
> > [d] RMLR: Extending Multinomial Logistic Regression into General Geometries
> >
> > [e] Riemannian Geometry of Symmetric Positive Definite Matrices via Cholesky Decomposition

---

### Official Review · Reviewer_SYJp · 2025-10-31

**Soundness:** 3
**Presentation:** 3
**Contribution:** 2
**Rating:** 2
**Confidence:** 4

**Summary:**

This paper proposes two new SPD metrics called Power–Cholesky Metric (PCM) and Bures–Wasserstein–Cholesky Metric (BWCM). The proposed metrics are used to develop Riemannian Multinomial Logistic Regression (MLR) classifiers and residual blocks for SPD neural networks. The authors conducted experiments on radar signal classification, action recognition, and tensor interpolation to show the effectiveness of the proposed metrics.

**Strengths:**

- This paper aims at improving SPD neural networks which have applications in many fields.
- In general, the exposition is clear and the paper is easy to follow.

**Weaknesses:**

- Contribution is marginal.

**Questions:**

This paper is heavily based on the works in [Nguyen and Yang, ICML 2023; Nguyen et al., ICLR 2024; Chen et al., NeurIPS 2024]. The proposed Riemannian metrics are formed by introducing small changes to existing ones.

For the experiments, while some improvements are shown on the chosen datasets, I do not think it is due to a better design of the proposed metrics compared to the well-known ones such as AIM, LEM, and LCM. This is because the performance of the proposed metrics depends on additional parameters (e.g., $\theta$) and one can easily figure out good settings for these parameters to get enhanced performance on some specific datasets. The main advantage of the proposed Riemannian metrics w.r.t. AIM and LEM is computation time, but as can be seen from the experiments, they have similar computation times as LCM.

Overall, I think the paper presents marginal contributions to the literature of SPD neural networks which are not good enough for publication in ICLR 2026.

Question:

1. In terms of technical contributions, what is the novelty of this work w.r.t. those in [Nguyen and Yang, ICML 2023; Nguyen et al., ICLR 2024; Chen et al., NeurIPS 2024] ?

---

> ### Author Response · Authors · 2025-11-27
> **Responses to Reviewer SYJp (1/2)**
>
> We thank Reviewer $\textcolor{green}{SYJp}$ for the valuable comments.
> ***
>
>
> # 1. Difference from [a-d].
> **Layer level vs. metric level.** Prior works [a–d] operate at the layer level. Given an existing SPD Riemannian metric such as AIM, LEM, or LCM, they derive network layers (e.g., MLR, FC) using the associated Riemannian exponential, logarithmic map, and parallel transport. In contrast, we work at the metric level by introducing new SPD metrics (PCM/BWCM) and deriving closed‑form expressions for their exponential/logarithmic maps, geodesics, parallel transport, and gyro-structure. Sec. 3.2 and Lem. 3.5 show that LCM is a special case of this framework, and Sec. 5 illustrates how these metrics immediately induce new MLR and residual layers for SPD neural networks. The following table summarizes the above comparison.
>
> Tab. A: summary of comparison between [a–d] and our work.
> | Research content | [a–d] | Our work |
> |------------------|---------------------------|-------------|
> | Research level | layer-level (based on existing metrics, derive layers) | metric-level (new metrics) |
> | New geometric structure | ✗ none | ✓ Cholesky product geometry; PCM & BWCM on SPD manifold |
> | New closed-form Riemannian operators | ✗ none | ✓ all closed-form |
> | Stability (LCM limitations) | ✗ unresolved | ✓ diagonal powers replace diagonal log/exp |
> | Induced MLR / residual blocks | Based on existing metrics | new metric ⇒ new layers |
> | Experimental behavior | | PCM/BWCM outperform existing metrics |
>
> # 2. Contributions.
>
> **1. Geometry level.**
> - **Cholesky product geometry and new SPD geometry.** We identify a product geometry underlying LCM and use it to define two new SPD Riemannian metrics, PCM and BWCM. For these metrics we derive closed‑form expressions for all core Riemannian operators. Lem. 3.5 further shows that our PCM/BWCM recover LCM as the limit $ \theta \to 0$. Therefore, PCM/BWCM form a family of metrics that contains LCM as a special case.
> - **Regularized view of LCM with improved stability.** The main numerical issue of LCM comes from diagonal logarithms and exponentials, which can drastically change the magnitudes of the Cholesky diagonals and hurt learning stability. PCM/BWCM modify LCM by replacing diagonal log/exp with diagonal powers. This yields a controlled regularization of LCM: when $ \theta$ is near zero, our metrics remain close to LCM while mitigating the excessive amplification of ill‑conditioned diagonals. Sec. 6.2 demonstrates that PCM/BWCM have better numerical stability than LCM, especially in high dimensions.
>
> **2. Network level.**
> - **Novel SPD MLR and residual blocks.** Using the closed‑form Riemannian operators of PCM/BWCM, we instantiate new SPD MLR layers and Riemannian residual blocks (Sec. 5). Sec. 6.1 shows that these PCM/BWCM‑based MLR and residual blocks achieve better performance than the ones based on AIM, LEM, and LCM.
>
> # 3. PCM and BWCM are faster than LCM on large dimensions.
>
> **Complexity analysis.** PCM, BWCM, and LCM are based on Cholesky decomposition, which is cheaper than the eigenvalue decompositions required by AIM, LEM, PEM, and BWM. Within this Cholesky family, the extra cost of LCM comes from diagonal logarithms and exponentials, while PCM/BWCM use diagonal powers instead. Since power operations are cheaper than log/exp, PCM/BWCM are expected to become faster than LCM as the dimension grows.
>
> **Empirical validation.** We measure the average wall-clock time of a single forward and backward training step of an SPD MLR as the matrix dimension increases. The model uses one SPD MLR with 50 output classes followed by a cross-entropy loss. For each dimension $n$, we randomly generate a batch of 30 $ n \times n$ SPD matrices, and for PEM we fix the matrix power to $0.5$. Tab. B reports the per-batch runtimes. The results align with the analysis above: PCM and BWCM have runtime comparable to LCM at small dimensions and become the fastest metrics as the matrix size grows.
>
> Tab. B: Per-batch runtime (seconds) of SPD MLRs under different metrics.
>
> | Dim | AIM | LEM | LCM | PEM | BWM | PCM | BWCM |
> |:---:|:---:|:---:|:---:|:---:|:---:|:---:|:----:|
> | 32 | 0.2380 | 0.0077 | 0.0046 | 0.0076 | 0.2377 | **0.0040** | **0.0040** |
> | 64 | 1.0139 | 0.0395 | 0.0303 | 0.0473 | 1.1205 | **0.0251** | **0.0225** |
> | 128 | 3.6256 | 0.1832 | 0.1490 | 0.1844 | 4.0674 | **0.1013** | **0.1019** |
> | 256 | 14.5142 | 0.7793 | 0.5833 | 0.7853 | 16.5918 | **0.3848** | **0.4077** |
> | 512 | 60.1918 | 3.2948 | 2.5030 | 3.4357 | 70.8647 | **1.7553** | **1.7526** |

---

> ### Author Response · Authors · 2025-11-27
> **Responses to Reviewer SYJp (2/2)**
>
> # 4. Evidence that our gains are not solely due to tuning $ \theta$.
>
>
> Tab. C: Results of SPD MLR of AIM, LEM, and LCM, which are copied from Tabs 2-3 in the main paper.
> | Backbone | SPDNet | SPDNet | SPDNet | GyroSPD | GyroSPD | GyroSPD |
> |:--------:|:------:|:------:|:------:|:-------:|:-------:|:-------:|
> | Dataset | Radar | HDM05 | FPHA | Radar | HDM05 | FPHA |
> | AIM | 94.53 ± 0.95 | 61.14 ± 0.94 | 85.57 ± 0.50 | 96.80 ± 0.59 | 66.05 ± 1.80 | 85.77 ± 0.52 |
> | LEM | 93.55 ± 1.21 | 60.28 ± 0.91 | 85.90 ± 0.47 | 96.58 ± 0.27 | 66.42 ± 0.47 | 85.87 ± 0.79 |
> | LCM | 93.49 ± 1.25 | 62.33 ± 2.15 | 86.37 ± 0.59 | 96.29 ± 0.53 | 68.37 ± 0.66 | 89.83 ± 0.28 |
>
> Tab. D: Results on SPD MLR of PCM and BWCM on the SPDNet backbone for different values of $ \theta$. The best result in each row is shown in **bold**.
> | Dataset | Metric | -2 | -1.5 | -1 | -0.75 | -0.5 | -0.25 | 0.25 | 0.5 | 0.75 | 1 | 1.5 |
> |:-------:|:------:|:--:|:----:|:--:|:-----:|:----:|:-----:|:----:|:---:|:----:|:-:|:---:|
> | Radar | PCM | 95.71 ± 0.57 | **95.79 ± 0.38** | 94.64 ± 0.34 | 95.36 ± 1.32 | 94.11 ± 1.05 | 94.51 ± 0.75 | 94.35 ± 0.71 | 93.55 ± 0.59 | 94.16 ± 0.96 | 93.87 ± 0.40 | 92.75 ± 0.68 |
> |  | BWCM | 91.89 ± 0.31 | 92.91 ± 1.05 | 92.16 ± 1.10 | **93.93 ± 0.79** | 92.13 ± 0.60 | 92.16 ± 1.30 | 92.40 ± 0.95 | 92.08 ± 0.85 | 92.48 ± 0.93 | 92.77 ± 0.66 | 91.81 ± 0.52 |
> | HDM05 | PCM | 46.63 ± 1.63 | 49.86 ± 1.41 | 58.81 ± 1.41 | 65.18 ± 2.89 | **65.75 ± 2.86** | 65.65 ± 2.03 | 64.00 ± 3.20 | 63.73 ± 2.90 | 64.59 ± 3.22 | 64.58 ± 3.14 | 64.62 ± 1.44 |
> |  | BWCM | 65.20 ± 2.63 | **67.40 ± 0.90** | 65.42 ± 1.88 | 65.29 ± 2.32 | 64.33 ± 2.71 | 64.07 ± 2.65 | 63.89 ± 2.50 | 64.67 ± 1.38 | 65.15 ± 0.75 | 63.33 ± 2.18 | 65.00 ± 1.48 |
> | FPHA | PCM | 88.20 ± 0.29 | 88.20 ± 0.32 | 88.30 ± 0.24 | 88.33 ± 0.21 | 88.47 ± 0.29 | 88.70 ± 0.39 | 89.03 ± 0.22 | 88.97 ± 0.16 | **89.40 ± 0.13** | 89.03 ± 0.16 | 89.23 ± 0.33 |
> |  | BWCM | 85.87 ± 0.66 | 85.87 ± 0.66 | 85.87 ± 0.66 | 85.93 ± 0.67 | 85.93 ± 0.67 | 85.93 ± 0.67 | 85.93 ± 0.67 | 86.00 ± 0.70 | 86.03 ± 0.75 | **86.27 ± 0.60** | 86.13 ± 0.73 |
>
> Tab. E: Results on SPD MLR of PCM and BWCM on the GyroSPD backbone for different values of $ \theta$. The best result in each row is shown in **bold**.
> | Dataset | Metric | -2 | -1.5 | -1 | -0.75 | -0.5 | -0.25 | 0.25 | 0.5 | 0.75 | 1 | 1.5 |
> |:-------:|:------:|:--:|:----:|:--:|:-----:|:----:|:-----:|:----:|:---:|:----:|:-:|:---:|
> | Radar | PCM | 96.88 ± 0.36 | 96.75 ± 0.26 | 96.27 ± 0.36 | **97.04 ± 0.64** | 96.45 ± 0.88 | 97.01 ± 0.66 | 96.77 ± 0.64 | 96.51 ± 0.65 | 96.56 ± 0.86 | 96.53 ± 0.79 | 96.32 ± 0.68 |
> |  | BWCM | 96.16 ± 0.90 | **96.21 ± 0.25** | 95.25 ± 1.19 | 96.16 ± 0.57 | 96.08 ± 0.48 | 95.84 ± 0.69 | 94.96 ± 0.94 | 95.60 ± 0.65 | 95.04 ± 0.33 | 94.77 ± 0.96 | 95.55 ± 0.91 |
> | HDM05 | PCM | 50.18 ± 0.99 | 32.49 ± 25.81 | 66.43 ± 1.22 | **71.93 ± 1.21** | 70.94 ± 1.17 | 69.64 ± 0.68 | 67.68 ± 0.92 | 68.24 ± 0.28 | 68.02 ± 0.40 | 67.80 ± 1.07 | 67.86 ± 1.69 |
> |  | BWCM | 72.64 ± 1.15 | **72.74 ± 0.43** | 69.58 ± 1.34 | 69.24 ± 1.14 | 68.84 ± 0.74 | 68.10 ± 1.02 | 67.72 ± 1.19 | 68.49 ± 0.64 | 68.08 ± 0.93 | 67.72 ± 0.58 | 67.45 ± 0.45 |
> | FPHA | PCM | 90.60 ± 0.58 | 91.17 ± 0.32 | 91.10 ± 0.29 | **91.17 ± 0.30** | 91.00 ± 0.30 | 90.87 ± 0.12 | 90.97 ± 0.16 | 90.97 ± 0.12 | 90.97 ± 0.07 | 90.90 ± 0.08 | 90.97 ± 0.22 |
> |  | BWCM | 90.87 ± 0.16 | 90.80 ± 0.07 | 90.83 ± 0.11 | 90.97 ± 0.07 | **91.00 ± 0.11** | **91.00 ± 0.11** | 90.97 ± 0.12 | 90.97 ± 0.12 | 90.93 ± 0.08 | 90.90 ± 0.08 | 90.87 ± 0.12 |
>
> **Ablations on $\theta$.** We study $\theta$ for PCM and BWCM in the SPD MLR on both SPDNet and GyroSPD backbones. We vary $\theta$ between $-2$ and $1.5$ while keeping all other settings as the main experiments. The tables above collect these results, including baselines under AIM, LEM, and LCM. Across many tested $\theta$ values, PCM and BWCM already match or outperform these existing metrics, demonstrating the effectiveness of our deformation $\theta$. For more details, please refer to App. E.3 in the revised manuscript.
>
> **Deformation $\theta$ and dataset characteristics.** In the SPD MLR, $\theta$ acts directly on Cholesky diagonals. App. E.4 further analyzes how $\theta$ interacts with the datasets by visualizing the distributions of Cholesky diagonals. On Radar and FPHA, the diagonals are relatively balanced, so changing $\theta$ only brings moderate improvements. On HDM05, the diagonals are highly imbalanced: most entries are small and close to zero, while a few are much larger. In this regime, tuning $\theta$ is more effective, and using a negative $\theta$ specifically activates the many small diagonal values, thus capturing more discriminative information.
>
>
> ***
> # References
> > [a] RMLR: Extending Multinomial Logistic Regression into General Geometries
> >
> > [b] Matrix Manifold Neural Networks++
> >
> > [c] Building Neural Networks on Matrix Manifolds: A Gyrovector Space Approach
> >
> > [d] Riemannian Residual Neural Networks

---

### Official Review · Reviewer_V5WE · 2025-11-01

**Soundness:** 3
**Presentation:** 2
**Contribution:** 2
**Rating:** 4
**Confidence:** 3

**Summary:**

The authors proposed two new metrics for the Cholesky manifold inspired by the
Log-Cholesky Metric (LCM) [Lin, 2019] and Bures–Wasserstein Metric (BWM) [Bhatia et al., 2019]. They show that the Cholesky manifold admits a simple decomposition into a linear space of strictly lower triangular matrices (SL) with the Euclidean metric and the product of positive numbers for the diagonal part.
The specific choices of metrics for the later results in a novel
$\theta$-Diagonal Power Metric ($\theta$-DPM) and $\mathbb{M}$-Diagonal
Bures-Wasserstein Metric ($\mathbb{M}$-BWM). The authors  develop closed formulas for geodesics, Riemannian logarithm, vector transport and weighted Frechet mean, which allows for
deriving gyro-structures on the Cholesky manifold. Finally, the authors suggest
using their metrics for SPD manifold as pullback metrics via Cholesky decomposition, which was proved to be a diffeomorphism in [Lin, 2019].

The authors claim that new metrics (both on the Cholesky manifold and SPD as a pullback) are demonstrating better numerical properties since
they are based on linear operations and inversions instead of exponential functions. The authors have conducted two experiments for the Multinomial Logistics Regression (MLR) classifiers on Radar dataset [Brooks et al., 2019] (signal classification),
HDM05 [Muller et al., 2007] and FPHA [Garcia-Hernando et al., 2018] datasets (human actions classification). The proposed approach is appears to be more reliable in the SPD interpolation task.

**Strengths:**

- Rigorous analysis of proposed Riemannian metrics for the Cholesky manifold: from geodesics to pullback metrics for SPD manifold.
 - Stability experiments: empirical demonstration of failure probability for the derived metrics.

**Weaknesses:**

- I am not sure if the composition of logarithm and exponentiation from LCE is not possible to be done using some standard numerical tricks like logsumexp or smth in this direction. Could you give the exact formula for the unstable part of LCE and reason why there is no any standard workaround? I will be ok with raising the score if this part is properly addressed, because it seems to be one of the key advantages of choosing your method.
- Given that $\theta$ is a central contribution of this work, an accurate ablation study is important to isolate its effect. Similarly regarding the matrix $\mathbb{M}$ across the $\theta$-DPM, $\mathbb{M}$-BWM, and $(\theta, \mathbb{M})$-BWM models.

**Questions:**

- Have you conducted experiments for failure probabilities estimation for pullback metrics on the SPD manifold?

 - How can I estimate $\theta$ parameter for a dataset in advance? Why is it negative in Tab. $8$ (lines $918$-$924$)? Is there any analysis for negative $\theta$?

 - What does the resulting asymptotical complexity of the proposed metrics equal to?

---

> ### Author Response · Authors · 2025-11-27
> **Responses to Reviewer V5WE (1/5)**
>
> We thank Reviewer $\textcolor{red}{V5WE}$ for the constructive suggestions and insightful comments!
> ***
>
>
> # 1. Numerical behavior: LCM vs. PCM \& BWCM
>
> We thank the reviewer for this important question and clarify below the stability advantages of PCM/BWCM over LCM from three perspectives. For brevity, we refer to $ \theta$-PCM and $(\theta, \mathbb{M})$-BWCM simply as PCM and BWCM.
>
> **1. Metric‑level comparison via pullback maps.**
> All three metrics (LCM, PCM, and BWCM) are induced through the Cholesky decomposition. Their key difference lies in the diagonal operator applied to the Cholesky factor.
>
> - LCM is induced by the diagonal logarithm $\log(\mathbb{L})$, whose logarithmic and exponential terms significantly amplify magnitude when diagonal values are small or large:
>   $$
>   \mathcal{S} _{++} ^n \ni P \mapsto \lfloor L \rfloor + \log(\mathbb{L}),
>   $$
> where $P = L L^{\top}$ is the Cholesky decomposition with diagonal $\mathbb{L}$ and strictly lower part $\lfloor L \rfloor$.
> - PCM and BWCM are induced by diagonal power, which changes more gradually and is therefore more numerically stable:
>   $$
>   \mathcal{S} _{++} ^n \ni P \mapsto \lfloor L \rfloor + \mathbb{L} ^{\theta}.
>   $$
>
> - Lem. 3.5 also shows that PCM/BWCM converge to LCM as $\theta \to 0$, making them stable generalizations.
>
> **2. Operator-level comparison via geodesics and distances.**
> Eq. (11) provides the geodesic and geodesic distance under these metrics:
> $$
> \gamma _{(P,V)} ^{\mathcal{S}}(t)
> = \operatorname{Chol} ^{-1}\left(\gamma _{(L,\widetilde{V})}(t)\right),
> \qquad
> d ^{\mathcal{S}}(P,Q)
> = d(L,K),
> $$
> where
> - $\mathrm{Chol}(P) = L$ and $\mathrm{Chol}(Q) = K$ are the Cholesky decomposition,
> - $V$ is a tangent vector at $P$,
> - $\widetilde{V}=\mathrm{Chol}_{*,P}(V)$ with $\mathrm{Chol}_{*,P}$ the differential of the Cholesky map at $P$, and
> - $\gamma$ and $d$ are the geodesic and distance in the Cholesky space.
>
> To analyze numerical behavior, it suffices to examine the Cholesky geometry. For Cholesky matrix $L$ and tangent vector $X$, Tab. 1 in the main paper provides the explicit forms.
>
> - Corresponding to LCM (diagonal log metric):
> $$
> \begin{aligned}
> \gamma(L,X)(t) &= \lfloor L \rfloor + t\lfloor X \rfloor + L \exp\left(t \mathbb{L} ^{-1} \mathbb{X}\right), \newline
> d ^{2}(P,Q) &= \lVert\lfloor K\rfloor-\lfloor L\rfloor\rVert _{\mathrm{F}} ^2+\lVert\log (\mathbb{K})-\log (\mathbb{L})\rVert _{\mathrm{F}} ^2,
> \end{aligned}
> $$
>
> - Corresponding to PCM ($\theta$-DPM):
> $$
> \begin{aligned}
> \gamma(L,X)(t) &= \lfloor L\rfloor + t\lfloor X\rfloor + \mathbb{L} \left(I + t\theta \mathbb{L} ^{-1}\mathbb{X}\right) ^{1/\theta}, \newline
> d ^{2}(P,Q)
> &= \lVert\lfloor K\rfloor-\lfloor L\rfloor\rVert _{\mathrm{F}} ^2+\frac{1}{\theta ^2}\left\lVert \left(\mathbb{K} ^{\theta}-\mathbb{L} ^{\theta}\right)\right\rVert _{\mathrm{F}} ^2.
> \end{aligned}
> $$
>
> - Corresponding to BWCM (($\theta,\mathbb{M}$)-DBWM):
> $$
> \begin{aligned}
> \gamma(L,X)(t) &= \lfloor L\rfloor + t\lfloor X\rfloor + \mathbb{L} \left(I + t\frac{\theta}{2}\mathbb{L} ^{-1}\mathbb{X}\right) ^{2/\theta}, \newline
> d ^{2}(P,Q)
> &= \lVert\lfloor K\rfloor-\lfloor L\rfloor\rVert _{\mathrm{F}} ^2+\frac{1}{\theta ^2}\left\lVert\mathbb{M} ^{-\frac{1}{2}}\left(\mathbb{K} ^{\frac{\theta}{2}}-\mathbb{L} ^{\frac{\theta}{2}}\right)\right\rVert _{\mathrm{F}} ^2.
> \end{aligned}
> $$
>
> From the above expressions, we highlight the following key numerical behavior.
> - **Geodesic.** For LCM, the exponential term $L \exp(t L ^{-1}X)$ is unstable when $L$ has small diagonal entries. In contrast, PCM and BWCM use the power map $(\cdot) ^{\theta}$, which mediates this instability.
> - **Distance.** For LCM, the logarithmic term $\|\log (\mathbb{K})-\log (\mathbb{L})\| _{\mathrm{F}} ^2$ can drastically change the magnitude of the diagonal entries. In contrast, PCM and BWCM use the power, which varies more gradually.
>
> **Empirical evidence.** Tab. 5 evaluates the numerical stability of geodesics for all three metrics on both small (3×3) and large (256×256) matrices by varying the smallest Cholesky diagonal $\epsilon$. PCM/BWCM maintain near-zero failure rates even when $\epsilon$ decreases to $1e^{-20}$, while LCM deteriorates rapidly as the diagonal entries shrink. The gap becomes especially pronounced in the high-dimensional setting, confirming the numerical advantage of our metrics.

---

> ### Author Response · Authors · 2025-11-27
> **Responses to Reviewer V5WE (2/5)**
>
> **3. Network‑level comparison via MLR and RResNet.**
> The operator-level behaviors propagate directly into downstream models. We illustrate this by the SPD MLR in Thm. 5.1 and the SPD residual blocks in Eq. (16).
>
> For SPD MLRs, let
> - $S$ be the input SPD feature,
> - $P _{k} \in \mathcal{S} ^n _{++}$ and $A _k \in \mathcal{S} ^n$ the MLR weights,
> - $K=\mathrm{Chol}(S)$ and $L _k=\mathrm{Chol}(P _{k})$,
> - $\mathbb{K}$, $\mathbb{L} _k$, $\mathbb{A} _k$ their diagonal parts,
> - $\lfloor K\rfloor$ and $\lfloor L _k\rfloor$ their strictly lower triangular parts.
>
> The probabilities take the following forms.
>
> - LCM [a, Thm. 4.2]:
>   $$
>   p(y=k\mid S)
>   \propto
>   \exp\left(
>     \left\langle \lfloor  K \rfloor - \lfloor L _k \rfloor, \lfloor A  _{k} \rfloor \right\rangle + \frac{1}{2}\left\langle\log(\mathbb{K}) - \log(\mathbb{L} _k),  \mathbb{A} _k \right\rangle
>   \right).
>   $$
>
> - PCM:
>   $$
>   \log p(y=k\mid S)
>   \propto
>   \exp \left[\left\langle\lfloor K\rfloor-\left\lfloor L _k  \right\rfloor,\left\lfloor A _k\right\rfloor\right\rangle + \frac{1}{2 \theta}\left\langle\mathbb{K} ^\theta-\mathbb{L} _k ^\theta, \mathbb{A} _k\right\rangle\right]
>   $$
>
> - BWCM:
>   $$
>   \log p(y=k\mid S)
>   \propto
>   \exp \left[\left\langle\lfloor K\rfloor-\left\lfloor L _k\right\rfloor,\left\lfloor A _k\right\rfloor\right\rangle+\frac{1}{4 \theta}\left\langle\mathbb{K} ^{\frac{\theta}{2}}-\mathbb{L} _k ^{\frac{\theta}{2}}, \mathbb{M} ^{-1} \mathbb{A} _k\right\rangle\right]
>   $$
>
> LCM MLR applies the diagonal logarithm $\log(\mathbb{K}) - \log(\mathbb{L} _k)$, which can drastically amplify differences in feature magnitudes. In contrast, PCM/BWCM use diagonal powers $\mathbb{K} ^{\theta} - \mathbb{L} _k ^{\theta}$, which moderate these changes.
>
> A similar effect appears in Riemannian ResNets, where the residual blocks rely on the Riemannian exponential. Under LCM, this involves a diagonal exponential term, whereas under our metrics it is replaced by a diagonal power map, leading to more gradual and numerically stable updates in practice.
>
> **Empirical evidence.** Experiments in Sec. 6 demonstrate that SPD MLRs and RResNets under PCM/BWCM outperform their LCM counterparts.
>
> # 2. Why LogSumExp is not applicable to LCM.
> The unstable part of LCM arises from elementwise diagonal logarithms and exponentials, different from LogSumExp $\log\sum _i \exp(z _i)$. A straightforward workaround is to clip the diagonal values before applying log or exp, which indeed bounds the feature range but may also restrict the representation power and degrade performance. In contrast, PCM/BWCM replace them with diagonal power maps that moderate these changes and lead to more gradual and numerically stable behavior.

---

> ### Author Response · Authors · 2025-11-27
> **Responses to Reviewer V5WE (3/5)**
>
> # 3. Ablations on $\theta$ and $\mathbb{M}$.
>
> **1. Ablations on the deformation factor $\theta$.**
>
> Tab. A: Results on SPD MLR of PCM and BWCM on the SPDNet backbone for different values of $ \theta$. The best result in each row is shown in **bold**.
> | Dataset | Metric | -2 | -1.5 | -1 | -0.75 | -0.5 | -0.25 | 0.25 | 0.5 | 0.75 | 1 | 1.5 |
> |:-------:|:------:|:--:|:----:|:--:|:-----:|:----:|:-----:|:----:|:---:|:----:|:-:|:---:|
> | Radar | PCM | 95.71 ± 0.57 | **95.79 ± 0.38** | 94.64 ± 0.34 | 95.36 ± 1.32 | 94.11 ± 1.05 | 94.51 ± 0.75 | 94.35 ± 0.71 | 93.55 ± 0.59 | 94.16 ± 0.96 | 93.87 ± 0.40 | 92.75 ± 0.68 |
> |  | BWCM | 91.89 ± 0.31 | 92.91 ± 1.05 | 92.16 ± 1.10 | **93.93 ± 0.79** | 92.13 ± 0.60 | 92.16 ± 1.30 | 92.40 ± 0.95 | 92.08 ± 0.85 | 92.48 ± 0.93 | 92.77 ± 0.66 | 91.81 ± 0.52 |
> | HDM05 | PCM | 46.63 ± 1.63 | 49.86 ± 1.41 | 58.81 ± 1.41 | 65.18 ± 2.89 | **65.75 ± 2.86** | 65.65 ± 2.03 | 64.00 ± 3.20 | 63.73 ± 2.90 | 64.59 ± 3.22 | 64.58 ± 3.14 | 64.62 ± 1.44 |
> |  | BWCM | 65.20 ± 2.63 | **67.40 ± 0.90** | 65.42 ± 1.88 | 65.29 ± 2.32 | 64.33 ± 2.71 | 64.07 ± 2.65 | 63.89 ± 2.50 | 64.67 ± 1.38 | 65.15 ± 0.75 | 63.33 ± 2.18 | 65.00 ± 1.48 |
> | FPHA | PCM | 88.20 ± 0.29 | 88.20 ± 0.32 | 88.30 ± 0.24 | 88.33 ± 0.21 | 88.47 ± 0.29 | 88.70 ± 0.39 | 89.03 ± 0.22 | 88.97 ± 0.16 | **89.40 ± 0.13** | 89.03 ± 0.16 | 89.23 ± 0.33 |
> |  | BWCM | 85.87 ± 0.66 | 85.87 ± 0.66 | 85.87 ± 0.66 | 85.93 ± 0.67 | 85.93 ± 0.67 | 85.93 ± 0.67 | 85.93 ± 0.67 | 86.00 ± 0.70 | 86.03 ± 0.75 | **86.27 ± 0.60** | 86.13 ± 0.73 |
>
> Tab. B: Results on SPD MLR of PCM and BWCM on the GyroSPD backbone for different values of $ \theta$. The best result in each row is shown in **bold**.
> | Dataset | Metric | -2 | -1.5 | -1 | -0.75 | -0.5 | -0.25 | 0.25 | 0.5 | 0.75 | 1 | 1.5 |
> |:-------:|:------:|:--:|:----:|:--:|:-----:|:----:|:-----:|:----:|:---:|:----:|:-:|:---:|
> | Radar | PCM | 96.88 ± 0.36 | 96.75 ± 0.26 | 96.27 ± 0.36 | **97.04 ± 0.64** | 96.45 ± 0.88 | 97.01 ± 0.66 | 96.77 ± 0.64 | 96.51 ± 0.65 | 96.56 ± 0.86 | 96.53 ± 0.79 | 96.32 ± 0.68 |
> |  | BWCM | 96.16 ± 0.90 | **96.21 ± 0.25** | 95.25 ± 1.19 | 96.16 ± 0.57 | 96.08 ± 0.48 | 95.84 ± 0.69 | 94.96 ± 0.94 | 95.60 ± 0.65 | 95.04 ± 0.33 | 94.77 ± 0.96 | 95.55 ± 0.91 |
> | HDM05 | PCM | 50.18 ± 0.99 | 32.49 ± 25.81 | 66.43 ± 1.22 | **71.93 ± 1.21** | 70.94 ± 1.17 | 69.64 ± 0.68 | 67.68 ± 0.92 | 68.24 ± 0.28 | 68.02 ± 0.40 | 67.80 ± 1.07 | 67.86 ± 1.69 |
> |  | BWCM | 72.64 ± 1.15 | **72.74 ± 0.43** | 69.58 ± 1.34 | 69.24 ± 1.14 | 68.84 ± 0.74 | 68.10 ± 1.02 | 67.72 ± 1.19 | 68.49 ± 0.64 | 68.08 ± 0.93 | 67.72 ± 0.58 | 67.45 ± 0.45 |
> | FPHA | PCM | 90.60 ± 0.58 | 91.17 ± 0.32 | 91.10 ± 0.29 | **91.17 ± 0.30** | 91.00 ± 0.30 | 90.87 ± 0.12 | 90.97 ± 0.16 | 90.97 ± 0.12 | 90.97 ± 0.07 | 90.90 ± 0.08 | 90.97 ± 0.22 |
> |  | BWCM | 90.87 ± 0.16 | 90.80 ± 0.07 | 90.83 ± 0.11 | 90.97 ± 0.07 | **91.00 ± 0.11** | **91.00 ± 0.11** | 90.97 ± 0.12 | 90.97 ± 0.12 | 90.93 ± 0.08 | 90.90 ± 0.08 | 90.87 ± 0.12 |
>
>
> **Setup.** We study $\theta$ for PCM and BWCM in the SPD MLR on both SPDNet and GyroSPD backbones. We vary $\theta$ between $-2$ and $1.5$ while keeping all other settings as the main experiments.
>
> **Results.** Tabs. A and B report the 5-fold results on the SPDNet and GyroSPD backbones. We make the following observations.
>
> - **Effectiveness.** On both backbones, most tested values of $\theta$ already match or surpass the AIM, LEM, and LCM baselines, and suitably chosen $\theta$ consistently improves accuracy over the default $\theta=1$, confirming the effectiveness of our deformation factor. Besides, the magnitude of this improvement is dataset dependent. On Radar and FPHA, tuning $ \theta$ brings modest improvements, whereas on HDM05 $ \theta$ yields much larger gains. This pattern is consistent with the analysis of Cholesky diagonal distributions discussed in the fourth response, as well as App. E.4.
> - **Relative stability of BWCM.** On HDM05, when $ \theta$ takes small negative values (for example, $ \theta = -1.5$ or $ \theta = -2$), the accuracy of PCM deteriorates, whereas BWCM remains comparatively stable. This behaviour aligns with the formulation of their SPD MLRs (Thm. 5.1). Since PCM and BWCM differ mainly in the diagonal powers $(\cdot) ^{\theta}$ versus $(\cdot) ^{\theta/2}$, BWCM experiences a milder deformation for the same $ \theta$.
>
> **Note.** The above has been added as App. E.3 in the revised manuscript.

---

> > ### Author Response · Authors · 2025-11-27
> > **Responses to Reviewer V5WE (4/5)**
> >
> > **2. Ablations on $\mathbb{M}$.**
> >
> > Tab. C: Results of BWCM with or without learnable $\mathbb{M}$ for the SPD MLR on the SPDNet and GyroSPD backbones.
> > | Backbone |      SPDNet      |      SPDNet      |    SPDNet    |      GyroSPD     |      GyroSPD     |    GyroSPD   |
> > |:--------:|:----------------:|:----------------:|:------------:|:----------------:|:----------------:|:------------:|
> > |  Dataset  |       Radar      |       HDM05      |     FPHA     |       Radar      |       HDM05      |     FPHA     |
> > |  $ \mathbb{M}$-BWCM  | **95.15 ± 0.61** |   66.71 ± 0.96   | 86.27 ± 0.63 | **96.93 ± 0.40** |   72.62 ± 0.43   | 91.00 ± 0.11 |
> > |   BWCM   |   93.93 ± 0.79   | **67.40 ± 0.90** | 86.27 ± 0.60 |   96.21 ± 0.25   | **72.74 ± 0.43** | 91.00 ± 0.11 |
> >
> > **Setup.** To examine the effect of the diagonal matrix $\mathbb{M}$ in our BWCM, we compare BWCM with fixed $ \mathbb{M} = I$ against a variant that learns a positive diagonal matrix $\mathbb{M}$ ($ \mathbb{M}$-BWCM). We follow the SPD MLR configurations on both SPDNet and GyroSPD backbones and use the same value of $ \theta$ for BWCM and $ \mathbb{M}$-BWCM, which is determined in $ \theta$-ablation experiments. In $ \mathbb{M}$-BWCM, we optimize an unconstrained diagonal vector $ v$ and set $ \mathbb{M} = \operatorname{diag}(\exp(v))$ so that the diagonal of $ \mathbb{M}$ remains positive. We initialize $ v$ to zero, corresponding to $ \mathbb{M} = I$ at the beginning of training. The size of $ v$ matches the input dimension of the SPD MLR: $8$, $30$, and $33$ on SPDNet and $20$, $3 \times 28$, and $9 \times 28$ on GyroSPD for Radar, HDM05, and FPHA (where $3$ and $9$ are channel dimensions).
> >
> > **Results.** Tab. C reports the 5-fold results. On both backbones, $ \mathbb{M}$-BWCM slightly improves accuracy on Radar, is marginally worse on HDM05, and matches BWCM on FPHA. Since the additional parameters introduced by $ \mathbb{M}$ are relatively small, their impact on the overall model is marginal. These results suggest that we can simply fix $ \mathbb{M} = I$ in BWCM, which is the setting used in our main experiments.
> >
> > **Note.** The above has been added as App. E.6 in the revised manuscript.
> >
> > # 4. On tuning $\theta$ and negative $\theta$.
> >
> > - **Analysis.** Tabs. A and B show that tuning $ \theta$ yields modest improvements on Radar and FPHA, but stronger gains on HDM05. In the SPD MLR, $\theta$ is applied to the Cholesky diagonal entries (Thm. 5.1). Its effect therefore depends on how these diagonals are distributed. App. E.4 visualizes the Cholesky diagonals across the three datasets. Radar and FPHA have relatively balanced diagonals, whereas on HDM05 (especially at the SPDNet output) almost all diagonal entries lie below $0.5$ and most are close to zero. For such diagonals, a negative $ \theta$ amplifies the entries and activates them, providing more discriminative information.
> >
> > - **Conclusions.** Since our metrics converge to LCM when $ \theta \to 0$ (Lem. 3.5), it is natural to use moderate values around zero, such as $ \theta = \pm 0.5$, as default settings. The choice of $ \theta$ can then be refined in a dataset-dependent way. For datasets whose Cholesky diagonals are relatively balanced (e.g., Radar and FPHA), these default values already work well. For imbalanced datasets such as HDM05, tuning $ \theta$ brings larger gains. Besides, when most diagonal entries are close to zero, a negative $ \theta$ is particularly effective because it activates these entries and provides more discriminative information.

---

> > > ### Author Response · Authors · 2025-11-27
> > > **Responses to Reviewer V5WE (5/5)**
> > >
> > > # 5. Asymptotic complexity and empirical validation.
> > >
> > > We analyse the computational efficiency of different metrics using SPD MLR as a representative task, from two perspectives: (1) asymptotic per-sample complexity; and (2) empirical running time.
> > >
> > > **Asymptotic complexity.** The dominant cost comes from metric-specific matrix functions. AIM, LEM, PEM, and BWM use eigenvalue-based matrix functions, whereas LCM, PCM, and BWCM rely on Cholesky decomposition. As shown in Algs. 4.2.3 and 8.3.3 of [e], eigenvalue and Cholesky decompositions require $ O(9 n ^{3})$ and $ O(\tfrac{1}{3} n ^{3})$ flops, respectively. Tab. D summarizes the resulting per-sample complexity for MLRs under different metrics. In particular, LCM, PCM, and BWCM achieve the lowest asymptotic complexity among these metrics, and PCM/BWCM are expected to be even faster in practice than LCM because diagonal power is cheaper than diagonal logarithm.
> > >
> > > Tab. D: Asymptotic per-sample complexity of a $ C$-class SPD MLR for an $ n \times n$ input SPD matrix.
> > > | Metric | Asymptotic complexity |
> > > |:------:|:----------------------|
> > > | AIM | $ O\bigl(9(1 + 2C)n ^{3}\bigr)$ |
> > > | LEM | $ O\bigl(9(1 + C)n ^{3}\bigr)$ |
> > > | LCM | $ O\bigl(\tfrac{1 + C}{3}n ^{3}\bigr)$ |
> > > | PEM | $ O\bigl(9(1 + C)n ^{3}\bigr)$ |
> > > | BWM | $ O\bigl((9(1 + 3C) + \tfrac{C}{3})n ^{3}\bigr)$ |
> > > | PCM | $ O\bigl(\tfrac{1 + C}{3}n ^{3}\bigr)$ |
> > > | BWCM | $ O\bigl(\tfrac{1 + C}{3}n ^{3}\bigr)$ |
> > >
> > > **Empirical validation.** We measure the average wall-clock time of a single forward and backward training step of an SPD MLR as the matrix dimension increases. The model uses one SPD MLR with 50 output classes followed by a cross-entropy loss. For each dimension $n$, we randomly generate a batch of 30 $ n \times n$ SPD matrices, and for PEM we fix the matrix power to $0.5$. Tab. E reports the per-batch runtimes. The results align with the asymptotic analysis: PCM and BWCM are among the fastest metrics across all dimensions and are significantly faster than eigenvalue-based metrics such as AIM, LEM, PEM, and BWM.
> > >
> > > Tab. E: Per-batch runtime (seconds) of SPD MLRs under different metrics.
> > >
> > > | Dim | AIM | LEM | LCM | PEM | BWM | PCM | BWCM |
> > > |:---:|:---:|:---:|:---:|:---:|:---:|:---:|:----:|
> > > | 32 | 0.2380 | 0.0077 | 0.0046 | 0.0076 | 0.2377 | **0.0040** | **0.0040** |
> > > | 64 | 1.0139 | 0.0395 | 0.0303 | 0.0473 | 1.1205 | **0.0251** | **0.0225** |
> > > | 128 | 3.6256 | 0.1832 | 0.1490 | 0.1844 | 4.0674 | **0.1013** | **0.1019** |
> > > | 256 | 14.5142 | 0.7793 | 0.5833 | 0.7853 | 16.5918 | **0.3848** | **0.4077** |
> > > | 512 | 60.1918 | 3.2948 | 2.5030 | 3.4357 | 70.8647 | **1.7553** | **1.7526** |
> > >
> > > **Note.** The above has been added as App. E.5 in the revised manuscript.
> > > ***
> > >
> > > # References
> > > > [a] RMLR: Extending Multinomial Logistic Regression into General Geometries
> > > >
> > > > [b] Deep CNNs Meet Global Covariance Pooling: Better Representation and Generalization
> > > >
> > > > [c] Global second-order pooling convolutional networks
> > > >
> > > > [d] Fast Differentiable Matrix Square Root and Inverse Square Root
> > > >
> > > > [e] Matrix computations (4th edition).

---

### Author Response · Authors · 2025-11-27
**Summary of Revisions in the Updated Manuscript**

We thank all reviewers for their constructive comments, which helped us improve the clarity and presentation of our work. In the revised manuscript, we added several new analyses and experiments (highlighted in $\textcolor{blue}{\text{blue}}$) that directly address the raised concerns. Below we summarize the main additions.

- **Ablations on the deformation factor $\theta$ in App. E.3.** We conduct ablations of the deformation factor $\theta$ ranging in $[-2, 1.5]$ on both SPDNet and GyroSPD backbones. Results show that most $\theta$ values already match or outperform existing metrics, confirming the effectiveness of our deformation.

- **How to choose $\theta$ via Cholesky-diagonal analysis in App. E.4.** We visualize the distributions of Cholesky diagonals across datasets and explain how $\theta$ interacts with datasets. When the diagonals are relatively balanced (e.g., Radar and FPHA), moderate values around zero (such as $\theta = \pm 0.5$) already work well and tuning $\theta$ only brings modest improvements. For datasets with highly imbalanced diagonals, especially when many diagonal entries are close to zero (e.g., HDM05), tuning $\theta$ has a much larger effect, and negative $\theta$ values are particularly effective because they activate these small entries and make them more discriminative.

- **Asymptotic complexity and empirical efficiency in App. E.5.** We compare all metrics in terms of asymptotic complexity and add a runtime experiment for SPD matrices under different dimensions. Our metrics are among the fastest across all tested dimensions and are especially efficient at high matrix dimensions.

- **Ablation on the diagonal matrix $\mathbb{M}$ in BWCM in App. E.6.** We evaluate learnable $\mathbb{M}$ versus fixed $\mathbb{M}=I$ and find that the improvement from learning $\mathbb{M}$ is marginal. This indicates that BWCM with $\mathbb{M} = I$ is already near-saturated, which is the configuration used in our main experiments.

We hope these additions address the reviewers’ concerns and further clarify the advantages and practicality of our proposed metrics.

---

### Meta-Review · Area_Chair_5nYd · 2025-12-24

**Summary:**

This submission proposes new Riemannian metrics for SPD learning by exploiting a product structure on the Cholesky manifold, leading to PCM/BWCM and giving closed-form Riemannian operators that are intended to be more numerically stable than log/exp-based alternatives. Reviewers broadly agreed the geometric development is solid, and multiple reviewers found the “replace diagonal log/exp by diagonal power transforms” motivation convincing and practically useful, especially given the closed-form toolkit and stability focus. The main concern was novelty and significance over prior SPD-network papers. After the rebuttal, the authors added additional ablations and efficiency evidence and clarified novelty against related work. The rebuttal effectively addresses the main concerns of the reviewers and I thus recommend acceptance.

**Reviewer Concerns:**

Across reviews, the core concerns were: (i) novelty vs prior SPD-layer papers, (ii) whether gains are primarily from tunable hyperparameters, (iii) adequacy of baselines and the breadth of evaluation, and (iv) whether existing numerical tricks could mitigate the stability issues. The main concerns on novelty, hyperparameter tuning and adequacy of baselines are addressed. The concern on whether existing numerical tricks could address the stability issue may remain valid.

**Reviewer Scores:**

Reviewer cJuQ would stay positive with a score of 8. Reviewer QVyo has confirmed his increase of score from 6 to 8. Reviewer V5WE may increase the score from 4 to 6 given their main concerns are addressed. Reviewer SYJp may slightly increase to 4 but still remain negative towards the paper due to their strong negative comments on novelty and significance.

---

### Decision · Program_Chairs · 2026-01-26

Accept (Poster)